# THEORY ON MIXTURE-OF-EXPERTS IN CONTINUAL LEARNING

**Hongbo Li[1,3], Sen Lin[2], Lingjie Duan[1], Yingbin Liang[3], Ness B. Shroff[3,4]**

[1]Engineering Systems and Design Pillar, Singapore University of Technology and Design,
`hongbo_li@mymail.sutd.edu.sg, lingjie_duan@sutd.edu.sg`
[2]Department of Computer Science, University of Houston, `slin50@central.uh.edu`
[3]Department of ECE, The Ohio State University, `{li.15242, liang.889, shroff.11}@osu.edu`
[4]Department of CSE, The Ohio State University

## ABSTRACT

Continual learning (CL) has garnered significant attention because of its ability to adapt to new tasks that arrive over time. Catastrophic forgetting (of old tasks) has been identified as a major issue in CL, as the model adapts to new tasks. The Mixture-of-Experts (MoE) model has recently been shown to effectively mitigate catastrophic forgetting in CL, by employing a gating network to sparsify and distribute diverse tasks among multiple experts. However, there is a lack of theoretical analysis of MoE and its impact on the learning performance in CL. This paper provides the first theoretical results to characterize the impact of MoE in CL via the lens of overparameterized linear regression tasks. We establish the benefit of MoE over a single expert by proving that the MoE model can diversify its experts to specialize in different tasks, while its router learns to select the right expert for each task and balance the loads across all experts. Our study further suggests an intriguing fact that the MoE in CL needs to terminate the update of the gating network after sufficient training rounds to attain system convergence, which is not needed in the existing MoE studies that do not consider the continual task arrival. Furthermore, we provide explicit expressions for the expected forgetting and overall generalization error to characterize the benefit of MoE in the learning performance in CL. Interestingly, adding more experts requires additional rounds before convergence, which may not enhance the learning performance. Finally, we conduct experiments on both synthetic and real datasets to extend these insights from linear models to deep neural networks (DNNs), which also shed light on the practical algorithm design for MoE in CL.

## 1 INTRODUCTION

Continual Learning (CL) has emerged as an important paradigm in machine learning (Parisi et al. (2019); Wang et al. (2024)), in which an expert aims to learn a sequence of tasks one by one over time. The expert is anticipated to leverage the knowledge gained from old tasks to facilitate learning new tasks, while simultaneously enhancing the performance of old tasks via the knowledge obtained from new ones. Given the dynamic nature of CL, one major challenge herein is known as *catastrophic forgetting* (McCloskey & Cohen (1989); Kirkpatrick et al. (2017)), where the expert can perform poorly on (i.e., easily forget) the previous tasks when learning new tasks if data distributions change largely across tasks. This becomes a more serious issue if a single expert continues to serve an increasing number of tasks.

Recently, the sparsely-gated Mixture-of-Experts (MoE) model has achieved astonishing successes in deep learning, especially in the development of large language models (LLMs) (e.g., Du et al. (2022); Li et al. (2024); Lin et al. (2024); Xue et al. (2024)). By adaptively routing different input data to one of the multiple experts through a gating network (Eigen et al. (2013); Shazeer et al. (2016)), different experts in the MoE model will be specialized to grasp different knowledge in the data. Thus inspired, there have emerged several attempts to leverage MoE in mitigating the forgetting issue in CL (e.g., Hihn & Braun (2021); Wang et al. (2022); Doan et al. (2023); Rypeść et al. (2023); Yu et al. (2024)) by training each expert to handle a particular set of tasks. However, these studies have primarily

focused on experimental investigations, whereas the theoretical understanding of MoE and its impact on the learning performance in CL is still lacking. In this paper, we aim to fill this gap by providing the first explicit theoretical results to comprehensively understand MoE in CL.

To this end, we study the sparsely-gated MoE model with $M$ experts in CL through the lens of overparameterized linear regression tasks (Viele & Tong (2002); Anandkumar et al. (2014); Zhong et al. (2016)). In our setting of CL, one learning task arrives in each round and its dataset is generated with the ground truth randomly drawn from a shared pool encompassing $N$ unknown linear models. Subsequently, the data is fed into a parameterized gating network, guided by which a softmax-based router will route the task to one of the $M$ experts for model training. The trained MoE model then further updates the gating network by gradient descent (GD). After that, a new task arrives and the above process repeats until the end of CL. It is worth noting that analyzing linear models is an important first step towards understanding the performance of deep neural networks (DNNs), as shown in many recent studies (e.g., Evron et al. (2022); Lin et al. (2023); Chen et al. (2022)).

Our main contributions are summarized as follows.

We provide *the first theoretical analysis to understand the behavior of MoE in CL*, through the lens of overparameterized linear regression tasks. By updating the gating network with a carefully designed loss function, we show that after sufficient training rounds (on the order of $\mathcal{O}(M)$) in CL for expert exploration and router learning, the MoE model will diversify and move into a balanced system state: Each expert will specialize either in a specific task (if $M > N$) or in a cluster of similar tasks (if $M < N$), and the router will consistently select the right expert for each task. Another interesting finding is that, unlike existing studies in MoE (e.g., Fedus et al. (2022); Chen et al. (2022); Li et al. (2024)), it is necessary to terminate the update of the gating network for MoE due to the dynamics of task arrival in CL. This will ensure that the learning system eventually converges to a stable state with balanced loads among all experts.

We provide *explicit expressions of the expected forgetting and generalization error to characterize the benefit of MoE on the performance of CL*. 1) Compared to the single expert case ($M = 1$), where tasks are learned by a single diverged model, the MoE model with diversified experts significantly enhances the learning performance, especially with large changes in data distributions across tasks. 2) Regardless of whether there are more experts ($M > N$) or fewer experts ($M < N$), both forgetting and generalization error converge to a small constant. This occurs because the router consistently selects the right expert for each task after the MoE model converges, efficiently minimizing model errors caused by switching tasks. 3) In MoE, initially adding more experts requires additional exploration rounds before convergence, which does not necessarily improve learning performance.

Finally, we conduct extensive experiments to verify our theoretical results. Specifically, our experimental results on synthetic data with linear models not only support our above-mentioned theoretical findings, but also show that load balancing reduces the average generalization error. This effectively improves the capacity of the MoE model compared to the unbalanced case. More importantly, the experiments on real datasets suggest that our theoretical findings can be further carried over beyond linear models to DNNs, which also provides insights on practical algorithm design for MoE in CL.

## 2 RELATED WORK

**Continual learning.** In the past decade, various empirical approaches have been proposed to tackle catastrophic forgetting in CL, which generally fall into three categories: 1) Regularization-based approaches (e.g., Kirkpatrick et al. (2017); Ritter et al. (2018); Gou et al. (2021); Liu & Liu (2021)), which introduce explicit regularization terms on key model parameters trained by previous tasks to balance old and new tasks. 2) Parameter-isolation-based approaches (e.g., Chaudhry et al. (2018); Serra et al. (2018); Jerfel et al. (2019); Yoon et al. (2019); Konishi et al. (2023)), which isolate parameters associated with different tasks to prevent interference between parameters. 3) Memory-based approaches (e.g., Farajtabar et al. (2020); Jin et al. (2021); Lin et al. (2021); Saha et al. (2020); Tang et al. (2021); Gao & Liu (2023)), which store data or gradient information from old tasks and replay them during training of new tasks.

On the other hand, theoretical studies on CL are very limited. Among them, Doan et al. (2021) and Bennani et al. (2020) introduce NTK overlap matrix to measure the task similarity and propose variants of the orthogonal gradient descent approach to address catastrophic forgetting. Lee et al.

(2021) consider a teacher-student framework to examine the impact of task similarity on learning performance. Peng et al. (2023) propose an ideal CL framework that can achieve no forgetting by assuming i.i.d. data distributions for all tasks. Evron et al. (2022) provide forgetting bounds in overparameterized linear models on different task orders. Further, Lin et al. (2023) provide explicit forms of forgetting and overall generalization error based on the testing error. These works collectively suggest that learning performance with a *single* expert tends to deteriorate when subsequent tasks exhibit significant diversification. In contrast, our work is the first to conduct theoretical analysis to understand the benefit of *multiple* experts on CL.

**Mixture-of-Experts model.** The MoE model has been extensively studied over the years for enhancing model capacity in deep learning (e.g., Eigen et al. (2013); Riquelme et al. (2021); Wang & Van Hoof (2022); Zhou et al. (2022); Chi et al. (2022); Zadouri et al. (2023)). Recently, it has found widespread applications in emerging fields such as LLMs (e.g., Du et al. (2022); Li et al. (2024); Lin et al. (2024); Xue et al. (2024)). To improve training stability and simplify the MoE structure, Shazeer et al. (2016) propose to sparsify the output of the gating network. Subsequently, Fedus et al. (2022) suggest routing each data sample to a single expert instead of multiple experts. For theoretical studies, Nguyen & Chamroukhi (2018) propose a maximum quasi-likelihood method for estimating MoE parameters, while Chen et al. (2022) analyze MoE mechanisms in deep learning for single-task classification. Unlike these works, which do not address sequential task training in CL, our study focuses on MoE in CL, introducing distinct training phases for the gating network. Additionally, we derive explicit expressions for forgetting and generalization errors.

**MoE in CL.** Recently, the MoE model has been applied to reducing catastrophic forgetting in CL (Lee et al. (2020); Hihn & Braun (2021); Wang et al. (2022); Doan et al. (2023); Rypeść et al. (2023); Yu et al. (2024)). For example, Lee et al. (2020) expand the number of experts using the Bayesian nonparametric framework to address task-free CL. Rypeść et al. (2023) propose to diverse experts by routing data with minimal distribution overlap to each expert and then combine experts' knowledge during task predictions to enhance learning stability. Additionally, Yu et al. (2024) apply MoE to expand the capacity of vision-language models, alleviating forgetting in CL. However, these works solely focus on empirical methods, lacking theoretical analysis of how the MoE performs in CL.

## 3 PROBLEM SETTING AND MoE MODEL DESIGN

**Notations.** For a vector $\boldsymbol{w}$, let $\|\boldsymbol{w}\|_2$ and $\|\boldsymbol{w}\|_\infty$ denote its $\ell$-2 and $\ell$-$\infty$ norms, respectively. For some positive constant $c_1$ and $c_2$, we define $x = \Omega(y)$ if $x > c_2|y|$, $x = \Theta(y)$ if $c_1|y| < x < c_2|y|$, and $x = \mathcal{O}(y)$ if $x < c_1|y|$. We also denote by $x = o(y)$ if $x/y \to 0$.

### 3.1 CL IN LINEAR MODELS

**General setting.** We consider the CL setting with $T$ training rounds. In each round $t \in [T]$, one out of $N$ tasks randomly arrives to be learned by the MoE model with $M$ experts. For each task, we follow most theoretical work on CL by fitting a linear model $f(\mathbf{X}) = \mathbf{X}^\top \boldsymbol{w}$ with ground truth $\boldsymbol{w} \in \mathbb{R}^d$ (e.g., Evron et al. (2022); Lin et al. (2023)), which serves as a foundation for understanding DNN generalization performance (Belkin et al. (2018); Ju et al. (2020)). Then, for the task arrival in the $t$-th training round, it corresponds to a linear regression problem, where the training dataset is denoted by $\mathcal{D}_t = (\mathbf{X}_t, \mathbf{y}_t)$. Here $\mathbf{X}_t \in \mathbb{R}^{d \times s_t}$ is the feature matrix with $s_t$ samples of $d$-dimensional vectors, and $\mathbf{y}_t \in \mathbb{R}^{s_t}$ is the output vector. In this study, we focus on the overparameterized regime, where $s_t < d$. Consequently, there exist numerous linear models that can perfectly fit the data.

**Ground truth and dataset.** Let $\mathcal{W} = \{\boldsymbol{w}_1, \cdots, \boldsymbol{w}_N\}$ represent the collection of ground truth vectors of all $N$ tasks. For any two tasks $n, n' \in [N]$, we assume $\|\boldsymbol{w}_n - \boldsymbol{w}_{n'}\|_\infty = \mathcal{O}(\sigma_0)$, where $\sigma_0 \in (0, 1)$ denotes the variance. Moreover, we assume that task $n$ possesses a unique feature signal $\boldsymbol{v}_n \in \mathbb{R}^d$ with $\|\boldsymbol{v}_n\|_\infty = \mathcal{O}(1)$ (Chen et al. (2022); Huang et al. (2024)).

In each training round $t \in [T]$, let $n_t \in [N]$ denote the index of the current task arrival with ground truth $\boldsymbol{w}_{n_t} \in \mathcal{W}$. In the following, we formally define the generation of dataset per training round.

**Definition 1.** *At the beginning of each training round $t \in [T]$, the dataset $\mathcal{D}_t = (\mathbf{X}_t, \mathbf{y}_t)$ of the new task arrival $n_t$ is generated by the following steps:*

*1) Uniformly draw a ground truth $\boldsymbol{w}_n$ from ground-truth pool $\mathcal{W}$ and let $\boldsymbol{w}_{n_t} = \boldsymbol{w}_n$.*

*2) Independently generate a random variable $\beta_t \in (0, C]$, where $C$ is a constant satisfying $C = \mathcal{O}(1)$.*

*3) Generate $\mathbf{X}_t$ as a collection of $s_t$ samples, where one sample is given by $\beta_t \boldsymbol{v}_{n_t}$ and the rest of the $s_t - 1$ samples are drawn from normal distribution $\mathcal{N}(\mathbf{0}, \sigma_t^2 \mathbf{I}_d)$, where $\sigma_t \geq 0$ is the noise level.*

*4) Generate the output to be $\mathbf{y}_t = \mathbf{X}_t^\top \boldsymbol{w}_{n_t}$.*

In any training round $t \in [T]$, the actual ground truth $\boldsymbol{w}_{n_t}$ of task arrival $n_t$ is unknown. However, according to Definition 1, task $n_t$ can be classified into one of $N$ clusters based on its feature signal $\boldsymbol{v}_{n_t}$. Although the position of $\boldsymbol{v}_{n_t}$ in feature matrix $\mathbf{X}_t$ is not specified for each task $n_t$, we can address this binary classification sub-problem over $\mathbf{X}_t$ using a single gating network in MoE (Shazeer et al. (2016); Fedus et al. (2022); Chen et al. (2022)). In this context, we aim to investigate whether the MoE model can enhance the learning performance in CL. For ease of exposition, we assume $s_t = s$ for all $t \in [T]$ in this paper. Then we will introduce the MoE model in the following subsections.

## 3.2 STRUCTURE OF THE MOE MODEL

As shown in Figure 1, an MoE model comprises a collection of $M$ experts, a router, and a gating network which is typically set to be linear (Shazeer et al. (2016); Fedus et al. (2022); Chen et al. (2022)). In the $t$-th round, upon the arrival of task $n_t$ and input of its data $\mathcal{D}_t = (\mathbf{X}_t, \mathbf{y}_t)$, the gating network computes its linear output $h_m(\mathbf{X}_t, \boldsymbol{\theta}_t^{(m)})$ for each expert $m \in [M]$, where $\boldsymbol{\theta}_t^{(m)} \in \mathbb{R}^d$ is gating network parameter for expert $m$. Define $\mathbf{h}(\mathbf{X}_t, \boldsymbol{\Theta}_t) := [h_1(\mathbf{X}_t, \boldsymbol{\theta}_t^{(1)}) \cdots h_M(\mathbf{X}_t, \boldsymbol{\theta}_t^{(M)})]$ and $\boldsymbol{\Theta}_t := [\boldsymbol{\theta}_t^{(1)} \cdots \boldsymbol{\theta}_t^{(M)}]$ as the outputs and the parameters of the gating network for all experts, respectively. Then we obtain $\mathbf{h}(\mathbf{X}_t, \boldsymbol{\Theta}_t) = \sum_{i \in [s_t]} \boldsymbol{\Theta}_t^\top \mathbf{X}_{t,i}$, where $\mathbf{X}_{t,i}$ is the $i$-th sample of the feature matrix $\mathbf{X}_t$.

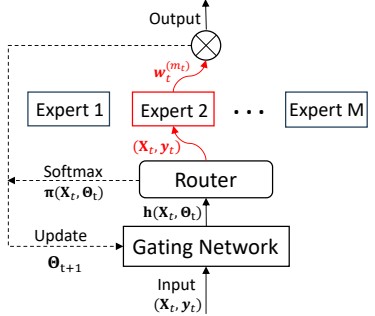

Figure 1: An illustration of the MoE model.

To sparsify the gating network and reduce the computation cost, we employ top-1 "switch routing", which maintains model quality while lowering routing computation, as demonstrated by Fedus et al. (2022); Chen et al. (2022); Yang et al. (2021). Although this top-1 gating model is simple, it is fundamental gaining a theoretical understanding of the behavior of MoE in CL, and its theoretical analysis is already non-trivial. Extending to the top-$k$ routing strategy (as introduced by Shazeer et al. (2016)) is nontrivial and falls outside the scope of this work. However, we still provide a discussion on the learning performance of the top-$k$ routing strategy later in Section 5.2.

In each round $t$, as depicted in Figure 1, for task $n_t$, the router selects the expert with the maximum gate output $h_m(\mathbf{X}_t, \boldsymbol{\theta}_t^{(m)})$, denoted as $m_t$, from the $M$ experts (Chen et al. (2022); Nguyen et al. (2024)). In practice, to encourage exploration across experts and stabilize MoE training, we add perturbations to the router (Shazeer et al. (2016); Fedus et al. (2022); Chen et al. (2022)). Specifically, task $n_t$ will be routed to the expert that satisfies

$$m_t = \arg\max_m \{h_m(\mathbf{X}_t, \boldsymbol{\theta}_t^{(m)}) + r_t^{(m)}\}, \tag{1}$$

where $r_t^{(m)}$ for any $m \in [M]$ is drawn independently from the uniform distribution $\text{Unif}[0, \lambda]$. We analyze in Appendix B that this routing strategy Eq. (1) ensures continuous and stable transitions for tasks. Additionally, the router calculates the softmax gate outputs, derived by

$$\pi_m(\mathbf{X}_t, \boldsymbol{\Theta}_t) = \frac{\exp(h_m(\mathbf{X}_t, \boldsymbol{\theta}_t^{(m)}))}{\sum_{m'=1}^M \exp(h_{m'}(\mathbf{X}_t, \boldsymbol{\theta}_t^{(m)}))}, \quad \forall m \in [M], \tag{2}$$

for the MoE to update the gating network parameter $\boldsymbol{\Theta}_{t+1}$ for all experts.

## 3.3 TRAINING OF THE MOE MODEL WITH KEY DESIGNS

**Expert model.** Let $\boldsymbol{w}_t^{(m)}$ denote the model of expert $m$ in the $t$-th training round, where each model is initialized from zero, i.e., $\boldsymbol{w}_0^{(m)} = 0$ for any $m \in [M]$. After the router determines the expert $m_t$ by Eq. (1), it transfers the dataset $\mathcal{D}_t = (\mathbf{X}_t, y_t)$ to this expert for updating $\boldsymbol{w}_t^{(m_t)}$. For any other expert $m \in [M]$ not selected ( i.e., $m \neq m_t$), its model $\boldsymbol{w}_t^{(m)}$ remains unchanged from $\boldsymbol{w}_{t-1}^{(m)}$. In

each round $t$, the training loss is defined by the mean-squared error (MSE) relative to dataset $\mathcal{D}_t$:

$$\mathcal{L}_t^{tr}(\boldsymbol{w}_t^{(m_t)}, \mathcal{D}_t) = \frac{1}{s_t}\|(\mathbf{X}_t)^\top \boldsymbol{w}_t^{(m_t)} - \mathbf{y}_t\|_2^2. \tag{3}$$

Since we focus on the overparameterized regime, there exist infinitely many solutions that perfectly satisfy $\mathcal{L}_t^{tr}(\boldsymbol{w}_t^{(m_t)}, \mathcal{D}_t) = 0$ in Eq. (3). Among these solutions, gradient descent (GD) starting from the previous expert model $\boldsymbol{w}_{t-1}^{(m_t)}$ at the convergent point provides a unique solution for minimizing $\mathcal{L}_t^{tr}(\boldsymbol{w}_t^{(m_t)}, \mathcal{D}_t)$ in Eq. (3), which is determined by the following optimization problem (Evron et al. (2022); Gunasekar et al. (2018); Lin et al. (2023)):

$$\min_{\boldsymbol{w}_t} \ \|\boldsymbol{w}_t - \boldsymbol{w}_{t-1}^{(m_t)}\|_2, \quad \text{s.t.} \ \mathbf{X}_t^\top \boldsymbol{w}_t = \mathbf{y}_t. \tag{4}$$

Solving Eq. (4), we update the selected expert in the MoE model for the current task arrival $n_t$ as follows, while keeping the other experts unchanged:

$$\boldsymbol{w}_t^{(m_t)} = \boldsymbol{w}_{t-1}^{(m_t)} + \mathbf{X}_t(\mathbf{X}_t^\top \mathbf{X}_t)^{-1}(\mathbf{y}_t - \mathbf{X}_t^\top \boldsymbol{w}_{t-1}^{(m_t)}). \tag{5}$$

**Gating network parameters.** After obtaining $\boldsymbol{w}_t^{(m_t)}$ in Eq. (5), the MoE updates the gating network parameter from $\boldsymbol{\Theta}_t$ to $\boldsymbol{\Theta}_{t+1}$ using GD for the next training round. On one hand, we aim for $\boldsymbol{\theta}_{t+1}^{(m)}$ of each expert $m$ to specialize in a specific task, which helps mitigate learning loss caused by the incorrect routing of distinct tasks. On the other hand, the router needs to balance the load among all experts (Fedus et al. (2022); Shazeer et al. (2016); Li et al. (2024)) to reduce the risk of model overfitting and enhance the learning performance in CL. To achieve this, we introduce our first key design of multi-objective training loss for gating network updates.

*Key design I: Multi-objective training loss.* First, based on the updated expert model $\boldsymbol{w}_t^{(m_t)}$ in Eq. (5), we propose the following locality loss function for updating $\boldsymbol{\Theta}_t$:

$$\mathcal{L}_t^{loc}(\boldsymbol{\Theta}_t, \mathcal{D}_t) = \sum_{m \in [M]} \pi_m(\mathbf{X}_t, \boldsymbol{\Theta}_t)\|\boldsymbol{w}_t^{(m)} - \boldsymbol{w}_{t-1}^{(m)}\|_2, \tag{6}$$

where $\pi_m(\mathbf{X}_t, \boldsymbol{\Theta}_t)$ is the softmax output defined in Eq. (2). Since our designed locality loss in Eq. (6) is minimized when the tasks with similar ground truths are routed to the same expert $m$ (e.g., $\boldsymbol{w}_t^{(m)} = \boldsymbol{w}_{t-1}^{(m)}$), it enjoys several benefits as shown later in our theoretical results: each expert will specialize in a particular set of tasks which leads to fast convergence of expert model $\boldsymbol{w}_t^{(m)}$, and the performance of CL in terms of forgetting and generalization error will be improved. Note that in Eq. (6), we only need to calculate the locality loss for the single expert $m_t$, as $\|\boldsymbol{w}_t^{(m)} - \boldsymbol{w}_{t-1}^{(m)}\|_2 = 0$ for any expert $m \neq m_t$ that has not updated its model, leading to low computational complexity.

In addition to the novel locality loss in Eq. (6), we follow the existing MoE literature (e.g., Fedus et al. (2022); Shazeer et al. (2016); Li et al. (2024)) where an auxiliary loss is typically defined to characterize load balance among the experts:

$$\mathcal{L}_t^{aux}(\boldsymbol{\Theta}_t, \mathcal{D}_t) = \alpha \cdot M \cdot \sum_{m \in [M]} f_t^{(m)} \cdot P_t^{(m)}, \tag{7}$$

where $\alpha$ is constant, $f_t^{(m)} = \frac{1}{t}\sum_{\tau=1}^{t} \mathbb{1}\{m_\tau = m\}$ is the fraction of tasks dispatched to expert $m$ since $t = 1$, and $P_t^{(m)} = \frac{1}{t}\sum_{\tau=1}^{t} \pi_m(\mathbf{X}_\tau, \boldsymbol{\Theta}_\tau) \cdot \mathbb{1}\{m_\tau = m\}$ is the average probability that the router chooses expert $m$ since $t = 1$. The auxiliary loss in Eq. (7) encourages exploration across all experts since it is minimized under a uniform routing with $f_t^{(m)} = \frac{1}{M}$ and $P_t^{(m)} = \frac{1}{M}$. Although the definition of auxiliary loss in Eq. (7) is not new, it is necessary and plays a crucial role for balancing the load across experts in the MoE model for CL.

Based on Eq. (3), Eq. (6) and Eq. (7), we finally define the task loss for each task arrival

---

**Algorithm 1** Training of the MoE model for CL

1: **Input:** $T, \sigma_0, \Gamma = \mathcal{O}(\sigma_0^{1.25}), \lambda = \Theta(\sigma_0^{1.25}), I^{(m)} = 0, \alpha = \mathcal{O}(\sigma_0^{0.5}), \eta = \mathcal{O}(\sigma_0^{0.5}), T_1 = \lceil \eta^{-1}M \rceil$;
2: Initialize $\boldsymbol{\theta}_0^{(m)} = \mathbf{0}$ and $\boldsymbol{w}_0^{(m)} = \mathbf{0}, \forall m \in [M]$;
3: **for** $t = 1, \cdots, T$ **do**
4:    Generate $r_t^{(m)}$ for any $m \in [M]$;
5:    Select $m_t$ in Eq. (1) and update $\boldsymbol{w}_t^{(m_t)}$ in Eq. (5);
6:    **if** $t > T_1$ **then**
7:       **for** $\forall m \in [M]$ with $|h_m - h_{m_t}| < \Gamma$ **do**
8:          $I^{(m)} = 1$;   // Convergence flag
9:       **end for**
10:   **end if**
11:   **if** $\exists m$, s.t. $I^{(m)} = 0$ **then**
12:      Update $\boldsymbol{\theta}_t^{(m)}$ as in Eq. (9) for any $m \in [M]$;
13:   **end if**
14: **end for**

---

$n_t$ as follows:
$$\mathcal{L}_t^{task}(\boldsymbol{\Theta}_t, \boldsymbol{w}_t^{(m_t)}, \mathcal{D}_t) = \mathcal{L}_t^{tr}(\boldsymbol{w}_t^{(m_t)}, \mathcal{D}_t) + \mathcal{L}_t^{loc}(\boldsymbol{\Theta}_t, \mathcal{D}_t) + \mathcal{L}_t^{aux}(\boldsymbol{\Theta}_t, \mathcal{D}_t). \tag{8}$$
Commencing from the initialization $\boldsymbol{\Theta}_0$, the gating network is updated based on GD:
$$\boldsymbol{\theta}_{t+1}^{(m)} = \boldsymbol{\theta}_t^{(m)} - \eta \cdot \nabla_{\boldsymbol{\theta}_t^{(m)}} \mathcal{L}_t^{task}(\boldsymbol{\Theta}_t, \boldsymbol{w}_t^{(m_t)}, \mathcal{D}_t), \forall m \in [M], \tag{9}$$
where $\eta > 0$ is the learning rate. Note that $\boldsymbol{w}_t^{(m_t)}$ in Eq. (5) is also the optimal solution for minimizing $\mathcal{L}_t^{task}(\boldsymbol{\Theta}_t, \boldsymbol{w}_t^{(m_t)}, \mathcal{D}_t)$ in Eq. (8). This is because both $\mathcal{L}_t^{loc}(\boldsymbol{\Theta}_t, \mathcal{D}_t)$ and $\mathcal{L}_t^{aux}(\boldsymbol{\Theta}_t, \mathcal{D}_t)$ are derived after updating $\boldsymbol{w}_t^{(m_t)}$, making $\mathcal{L}_t^{tr}(\boldsymbol{w}_t^{(m_t)}, \mathcal{D}_t)$ the sole objective for $\boldsymbol{w}_t^{(m_t)}$ in Eq. (8).

*Key design II: Early termination.* To ensure the stable convergence of the system with balanced loads among experts (which we will theoretically justify in Section 4), after training sufficient (i.e., $T_1$) rounds for expert exploration, we introduce an early termination strategy in Algorithm 1 by evaluating a convergence flag $I^{(m)}$ for each expert $m$. This flag assesses the output gap, defined as $|h_m(\mathbf{X}_t, \boldsymbol{\theta}_t) - h_{m_t}(\mathbf{X}_t, \boldsymbol{\theta}_t)|$, between the expert itself and any selected expert $m_t$ for $t > T_1$. If this gap exceeds threshold $\Gamma$ for expert $m$, which indicates that gating network parameter $\boldsymbol{\theta}_t^{(m)}$ has not converged, then the MoE model continues updating $\boldsymbol{\Theta}_t$ for all experts based on Eq. (9). Otherwise, the update of $\boldsymbol{\Theta}_t$ is permanently terminated.

## 4 THEORETICAL RESULTS ON MOE TRAINING FOR CL

In this section, we provide theoretical analysis for the training of expert models and the gating network in Algorithm 1, which further justifies our key designs in Section 3. Specifically, (i) we first support our key design I by proving that the expert model converges fast via updating $\boldsymbol{\Theta}_t$ under our designed locality loss in Eq. (6). (ii) We then show that our key design II, early termination in updating $\boldsymbol{\Theta}_t$, is necessary to ensure a stable convergence system state with experts' balanced loads. For clarity, we study the case with $M > N$ in this section (labeled as $M > N$ version), and further extend the results to the $M < N$ version in appendices. To characterize expert specialization, we first show that each expert's gate output is determined by the input feature signal $\boldsymbol{v}_n$ of $\mathbf{X}_t$.

**Lemma 1** ($M > N$ version). *For any two feature matrices $\mathbf{X}$ and $\tilde{\mathbf{X}}$ with the same feature signal $\boldsymbol{v}_n$, with probability at least $1 - o(1)$, their corresponding gate outputs of the same expert $m$ satisfy*
$$\left| h_m(\mathbf{X}, \boldsymbol{\theta}_t^{(m)}) - h_m(\tilde{\mathbf{X}}, \boldsymbol{\theta}_t^{(m)}) \right| = \mathcal{O}(\sigma_0^{1.5}). \tag{10}$$

The full version containing $M < N$ case and the proof of Lemma 1 are given in Appendix C. According to Lemma 1, the router decides expert $m_t$ for task $n_t$ based primarily on its feature signal $\boldsymbol{v}_{n_t}$. Consequently, given $N$ tasks, all experts can be grouped into $N$ sets according to their specialty, i.e., their gating parameter $\boldsymbol{\theta}_t^{(m)}$'s to identify feature signal $\boldsymbol{v}_n$, where each expert set is defined as:
$$\mathcal{M}_n = \left\{ m \in [M] \big| n = \arg\max_{j \in [N]} (\boldsymbol{\theta}_t^{(m)})^\top \boldsymbol{v}_j \right\}. \tag{11}$$

The following proposition indicates the convergence of the expert model after sufficient training rounds under Algorithm 1.

**Proposition 1** ($M > N$ version). *Under Algorithm 1, with probability at least $1 - o(1)$, for any $t > T_1$, where $T_1 = \lceil \eta^{-1} M \rceil$, each expert $m \in [M]$ stabilizes within an expert set $\mathcal{M}_n$, and its expert model remains unchanged beyond time $T_1$, satisfying $\boldsymbol{w}_{T_1+1}^{(m)} = \cdots = \boldsymbol{w}_T^{(m)}$.*

The full version and the proof of Proposition 1 are given in Appendix E. Proposition 1 demonstrates that after $T_1$ rounds of expert exploration, each expert will specialize in a specific task, enforced by minimizing locality loss in Eq. (6). After that, expert models remain unchanged until the end of $T$.

Next, the following proposition characterizes the dynamics of gate outputs if there is no termination of updating gating network parameters $\boldsymbol{\Theta}_t$ in Algorithm 1.

**Proposition 2** ($M > N$ version). *If the MoE keeps updating $\boldsymbol{\Theta}_t$ by Eq. (9) at any round $t \in [T]$, we obtain: 1) At round $t_1 = \lceil \eta^{-1} \sigma_0^{-0.25} M \rceil$, the following property holds*
$$\left| h_m(\mathbf{X}_{t_1}, \boldsymbol{\theta}_{t_1}^{(m)}) - h_{m'}(\mathbf{X}_{t_1}, \boldsymbol{\theta}_{t_1}^{(m')}) \right| = \begin{cases} \mathcal{O}(\sigma_0^{1.75}), & \text{if } m, m' \in \mathcal{M}_n, \\ \Theta(\sigma_0^{0.75}), & \text{otherwise.} \end{cases} \tag{12}$$
*2) At round $t_2 = \lceil \eta^{-1} \sigma_0^{-0.75} M \rceil$, the following property holds*
$$\left| h_m(\mathbf{X}_{t_2}, \boldsymbol{\theta}_{t_2}^{(m)}) - h_{m'}(\mathbf{X}_{t_2}, \boldsymbol{\theta}_{t_2}^{(m')}) \right| = \mathcal{O}(\sigma_0^{1.75}), \forall m, m' \in [M]. \tag{13}$$

The full version and the proof of Proposition 2 are given in Appendix F. According to Proposition 2, if the MoE updates $\boldsymbol{\Theta}_t$ at any round $t$, the gap between gate outputs of experts within the same expert set converges to $\mathcal{O}(\sigma_0^{1.75})$ by round $t_1$ (Eq. (12)). In contrast, the gap between experts in different sets is sufficiently large, i.e., $\Theta(\sigma_0^{0.75})$, indicating that the MoE has successfully diversified experts into different sets at round $t_1$ as in Eq. (11). However, unlike MoE in single-task learning that can stop training at any time after the expert models converge (e.g., Celik et al. (2022); Li et al. (2024)), MoE in CL requires both the gating network and the experts to be suitably updated with the continuous arrival of new tasks. This is necessary to balance the load on each expert and maximize the system capacity utilization. However, continuing updating $\boldsymbol{\Theta}_t$ according to Eq. (9) will eventually reduce the output gap between any two experts to $\mathcal{O}(\sigma_0^{1.75})$ in Eq. (13) at training round $t_2$, causing the router to select wrong experts for subsequent task arrivals and incurring additional training errors.

Based on Proposition 2, it is necessary to terminate the update of $\boldsymbol{\Theta}_t$ to preserve a sufficiently large output gap between any two experts in different sets, ensuring expert diversity as in Eq. (12) at round $t_1$. This motivates our design of early termination in Algorithm 1, outlined from Line 7 to Line 13. In the next proposition, we prove the benefit of terminating updating $\boldsymbol{\Theta}_t$ in Algorithm 1.

**Proposition 3** ($M > N$ version)**.** *Under Algorithm 1, the MoE terminates updating $\boldsymbol{\Theta}_t$ since round $T_2 = \mathcal{O}(\eta^{-1}\sigma_0^{-0.25}M)$. Then for any task arrival $n_t$ at $t > T_2$, the router selects any expert $m \in \mathcal{M}_{n_t}$ with an identical probability of $\frac{1}{|\mathcal{M}_{n_t}|}$, where $|\mathcal{M}_{n_t}|$ is the number of experts in set $\mathcal{M}_n$.*

The full version and the proof of Proposition 3 are given in Appendix G. According to Algorithm 1, once the MoE terminates updating $\boldsymbol{\Theta}_t$, the random noise $r_t^{(m)}$ in Eq. (1) will guide the router to select experts in the same expert set with identical probability, effectively balancing the loads across experts therein. Our theoretical analysis will be further corroborated by the experiments later in Section 6.

## 5 THEORETICAL RESULTS ON FORGETTING AND GENERALIZATION

For the MoE model described in Section 3, we define $\mathcal{E}_t(\boldsymbol{w}_t^{(m_t)})$ as the model error in the $t$-th round:
$$\mathcal{E}_t(\boldsymbol{w}_t^{(m_t)}) = \|\boldsymbol{w}_t^{(m_t)} - \boldsymbol{w}_{n_t}\|_2^2, \tag{14}$$
which characterizes the generalization performance of the selected expert $m_t$ with model $\boldsymbol{w}_t^{(m_t)}$ for task $n_t$ at round $t$. Following the existing literature on CL (e.g., Lin et al. (2023); Chaudhry et al. (2018)), we assess the performance of MoE in CL using the metrics of *forgetting* and *overall generalization error*, defined as follows:

(1) *Forgetting:* Define $F_t$ as the forgetting of old tasks after learning task $n_t$ for $t \in \{2, \cdots, T\}$:
$$F_t = \frac{1}{t-1} \sum_{\tau=1}^{t-1} (\mathcal{E}_\tau(\boldsymbol{w}_t^{(m_\tau)}) - \mathcal{E}_\tau(\boldsymbol{w}_\tau^{(m_\tau)})). \tag{15}$$

(2) *Overall generalization error:* We evaluate the generalization performance of the model $\boldsymbol{w}_T^{(m)}$ from the last training round $T$ by computing the average model error across all tasks:
$$G_T = \frac{1}{T} \sum_{\tau=1}^{T} \mathcal{E}_\tau(\boldsymbol{w}_T^{(m_\tau)}). \tag{16}$$
In the following, we present explicit forms of the above two metrics for learning with a single expert (i.e., $M = 1$) as a benchmark (cf. Lin et al. (2023)). Here we define $r := 1 - \frac{s}{d}$ as the overparameterization ratio.

**Proposition 4.** *If $M = 1$, for any training round $t \in \{2, \cdots, T\}$, we have*
$$\mathbb{E}[F_t] = \frac{1}{t-1} \sum_{\tau=1}^{t-1} \left\{ \frac{r^t - r^\tau}{N} \sum_{n=1}^{N} \|\boldsymbol{w}_n\|^2 + \frac{r^\tau - r^t}{N^2} \sum_{n \neq n'} \|\boldsymbol{w}_{n'} - \boldsymbol{w}_n\|^2 \right\}, \tag{17}$$
$$\mathbb{E}[G_T] = \frac{r^T}{N} \sum_{n=1}^{N} \|\boldsymbol{w}_n\|^2 + \frac{1 - r^T}{N^2} \sum_{n \neq n'} \|\boldsymbol{w}_n - \boldsymbol{w}_n'\|^2. \tag{18}$$

Note that the setting here (with $M = 1$) differs slightly from Lin et al. (2023) as we have $N$ tasks in total. Hence a proof of Proposition 4 is provided in Appendix H. Proposition 4 implies that distinct tasks with large model gap $\sum_{n \neq n'} \|\boldsymbol{w}_n - \boldsymbol{w}_n'\|^2$ lead to poor performance of both $\mathbb{E}[F_t]$ in Eq. (17) and $\mathbb{E}[G_t]$ in Eq. (18), which is missing in the existing CL literature (e.g., Lesort et al. (2023)).

Next, we investigate the impact of MoE on CL under two cases: (I) when there are more experts than tasks ($M > N$), and (II) when there are fewer experts than tasks ($M < N$). The benefit of MoE will be characterized by comparing our results with the single-expert baseline in Proposition 4.

## 5.1 CASE I: MORE EXPERTS THAN TASKS

Based on Proposition 1, we derive the explicit upper bounds for both forgetting and overall generalization error in the following theorem. To simplify notations, we define $L_t^{(m)} := t \cdot f_t^{(m)}$ as the cumulative number of task arrivals routed to expert $m$ up to round $t$, where $f_t^{(m)}$ is given in Eq. (7).

**Theorem 1.** *If $M = \Omega(N \ln(N))$, for each round $t \in \{2, \cdots, T_1\}$, the expected forgetting satisfies*

$$\mathbb{E}[F_t] < \tfrac{1}{t-1} \sum_{\tau=1}^{t-1} \left\{ \frac{r^{L_t^{(m_\tau)}} - r^{L_\tau^{(m_\tau)}}}{N} \sum_{n=1}^{N} \|\boldsymbol{w}_n\|^2 + \frac{r^{L_\tau^{(m_\tau)}} - r^{L_t^{(m_\tau)}}}{N^2} \sum_{n \neq n'} \|\boldsymbol{w}_{n'} - \boldsymbol{w}_n\|^2 \right\}. \tag{19}$$

*For each $t \in \{T_1 + 1, \cdots, T\}$, we have $\mathbb{E}[F_t] = \frac{T_1-1}{t-1}\mathbb{E}[F_{T_1}]$. Further, after training task $n_T$ in the last round $T$, the overall generalization error satisfies*

$$\mathbb{E}[G_T] < \tfrac{1}{T} \sum_{\tau=1}^{T} \left\{ \frac{r^{L_{T_1}^{(m_\tau)}}}{N} \sum_{n=1}^{N} \|\boldsymbol{w}_n\|^2 + \frac{1 - r^{L_{T_1}^{(m_\tau)}}}{N^2} \sum_{n \neq n'} \|\boldsymbol{w}_{n'} - \boldsymbol{w}_n\|^2 \right\}. \tag{20}$$

The proof of Theorem 1 is given in Appendix I, and we have the following insights.

1) *Forgetting.* If $t \leq T_1$, in Eq. (19), the coefficient $r^{L_t^{(m_\tau)}} - r^{L_\tau^{(m_\tau)}}$ of the term $\sum_{n=1}^{N} \|\boldsymbol{w}_n\|^2$ is smaller than 0 because $L_t^{(m_\tau)} \geq L_\tau^{(m_\tau)}$ and $r < 1$, indicating that the training of new tasks enhances the performance of old tasks due to the repeated task arrivals in this phase. Meanwhile, the coefficient of model gap $\sum_{n \neq n'} \|\boldsymbol{w}_{n'} - \boldsymbol{w}_n\|^2$ is greater than 0, indicating that the forgetting is due to experts' *exploration* of distinct tasks. However, as stated in Proposition 1, once the expert models converge at $t = T_1$, training on newly arriving tasks with correct routing no longer causes forgetting of previous tasks. Consequently, for $t \in \{T_1 + 1, \cdots, T\}$, $\mathbb{E}[F_t] = \frac{T_1-1}{t-1}\mathbb{E}[F_{T_1}]$ decreases with $t$ and converges to zero as $T \to \infty$. This result highlights that, unlike the oscillatory forgetting observed in Eq. (17) for a single expert, the MoE model effectively minimizes expected forgetting in CL through its correct routing mechanism. Furthermore, a decrease in *task similarity*, i.e., larger model gaps, further amplifies the learning benefit of the MoE model.

2) *Generalization error.* Note that the second term $\sum_{n \neq n'}^{N} \|\boldsymbol{w}_{n'} - \boldsymbol{w}_n\|^2$ dominates the generalization error when tasks are *less similar* with large model gaps. In Eq. (20), the coefficient $1 - r^{L_{T_1}^{(m_\tau)}}$ of the model gaps $\sum_{n \neq n'}^{N} \|\boldsymbol{w}_{n'} - \boldsymbol{w}_n\|^2$ is smaller than $1 - r^T$ in Eq. (18), due to the convergence of expert models after round $T_1$. Therefore, the generalization error under MoE is reduced compared to that of a single expert, especially as $T$ increases (where $1 - r^T$ approaches 1 in Eq. (18)).

3) *Expert number.* According to Theorem 1, for $t > T_1$, $\mathbb{E}[F_t]$ increases with $T_1$ as additional rounds of expert exploration accumulate more model errors in Eq. (19). Regarding $\mathbb{E}[G_T]$ in Eq. (20), a longer exploration period $T_1$ for experts increases the coefficient $1 - r^{L_{T_1}^{(m_\tau)}}$ of the model gaps, leading to an increase in $\mathbb{E}[G_T]$ when the model gaps across tasks are large. Since $T_1$ increases with expert number $M$, adding more experts does not enhance learning performance but delays convergence. Note that if $M = 1$ in Theorem 1, $L_t^{(m_\tau)}$ becomes $t$ for the single expert, causing Eq. (19) and Eq. (20) to specialize to Eq. (17) and Eq. (18), respectively.

## 5.2 CASE II: FEWER EXPERTS THAN TASKS

Next, we consider a more general case with fewer experts than tasks, i.e., $M < N$, where Algorithm 1 still works efficiently. In particular, we assume that the $N$ ground truths in $\mathcal{W}$ can be classified into $K$ clusters, where $K < M$, based on the task similarity. Let $\mathcal{W}_k$ denote the $k$-th task cluster. For any two tasks $n, n' \in [N]$ in the same cluster with $\boldsymbol{w}_n, \boldsymbol{w}_{n'} \in \mathcal{W}_k$, we assume $\|\boldsymbol{w}_n - \boldsymbol{w}_{n'}\|_\infty = \mathcal{O}(\sigma_0^{1.5})$. Then we let set $\mathcal{M}_k$ include all experts that specialize in tasks within the $k$-th task cluster.

Recall Proposition 1 indicates that expert models converge after $T_1$ rounds of exploration if $M > N$. However, in the case of fewer experts than tasks ($M < N$), each expert has to specialize in learning a cluster of similar tasks. Consequently, as similar tasks within the same cluster are continuously

routed to each expert, the expert models keep updating after round $T_1$, behaving differently from the $M > N$ case in Proposition 1. Given the above understanding, we have the following theorem.

**Theorem 2.** *If $M < N$ and $M = \Omega(K \ln(K))$, for any $t \in \{1, \cdots, T_1\}$, the expected forgetting $\mathbb{E}[F_t]$ is the same as Eq. (19). While for any $t \in \{T_1 + 1, \cdots, T\}$, the expected forgetting satisfies*

$$\mathbb{E}[F_t] < \frac{1}{t-1} \sum_{\tau=1}^{t-1} \frac{r^{L_t^{(m_\tau)}} - r^{L_\tau^{(m_\tau)}}}{N} \sum_{n=1}^{N} \|\boldsymbol{w}_n\|^2 + \frac{1}{t-1} \sum_{\tau=1}^{T_1} \frac{r^{L_i^{(m_\tau)}} - r^{L_{T_1}^{(m_\tau)}}}{N^2} \sum_{n \neq n'} \|\boldsymbol{w}_{n'} - \boldsymbol{w}_n\|^2$$
$$+ \frac{\Psi_1}{t-1} \sum_{\tau=T_1+1}^{t} (1 - r^{L_t^{(m_\tau)} - L_\tau^{(m_\tau)}}) r^{L_t^{(m_\tau)} - L_{T_1}^{(m_\tau)} - 1}, \tag{21}$$

*where $\Psi_1 = \frac{1}{N} \sum_{n=1}^{N} \sum_{n,n' \in \mathcal{W}_k} \frac{\|\boldsymbol{w}_{n'} - \boldsymbol{w}_n\|^2}{|\mathcal{W}_k|}$ is the expected model gap between any two tasks in the same task cluster. After training task $n_T$ in round $T$, the overall generalization error satisfies*

$$\mathbb{E}[G_T] < \frac{1}{T} \sum_{\tau=1}^{T} \frac{r^{L_T^{(m_\tau)}}}{N} \sum_{n=1}^{N} \|\boldsymbol{w}_n\|^2 + \frac{1}{T} \sum_{\tau=1}^{T_1} \frac{r^{L_T^{(m_\tau)} - L_{T_1}^{(m_\tau)}} (1 - r^{L_{T_1}^{(m_\tau)}})}{N^2} \sum_{n \neq n'} \|\boldsymbol{w}_{n'} - \boldsymbol{w}_n\|^2$$
$$+ \frac{\Psi_2}{T} \left\{ \sum_{\tau=1}^{T_1} (1 - r^{L_T^{(m_\tau)} - L_{T_1}^{(m_\tau)}}) + \sum_{\tau=T_1+1}^{T} r^{L_T^{(m_\tau)} - L_{T_1}^{(m_\tau)}} (1 - r^{L_{T_1}^{(m_\tau)}}) \right\}$$
$$+ \frac{\Psi_1}{T} \sum_{\tau=T_1+1}^{T} (1 - r^{L_T^{(m_\tau)} - L_{T_1}^{(m_\tau)}}), \tag{22}$$

*where $\Psi_2 = \frac{1}{N} \sum_{n=1}^{N} \frac{1}{K} \sum_{k=1}^{K} \frac{1}{|\mathcal{W}_k|} \sum_{n' \in \mathcal{W}_k} \|\boldsymbol{w}_{n'} - \boldsymbol{w}_n\|^2$ is the expected model gap between a randomly chosen task in $\mathcal{W}$ and any task in a fixed ground-truth cluster $\mathcal{W}_k$.*

The proof of Theorem 2 is given in Appendix J, and we provide the following insights.

1) *Forgetting.* Compared to Theorem 1, $\mathbb{E}[F_t]$ in Eq. (21) introduces an additional term $\Psi_1$, which measures the forgetting of task arrivals during $\{T_1 + 1, \cdots, \tau\}$ caused by updating expert models for new task arrival in round $\tau \in \{T_1 + 1, \cdots, T\}$. However, since tasks routed to the same expert during $\{T_1 + 1, \cdots, T\}$ belong to the same cluster, their small model gaps lead to minimal forgetting. If there is only one task in each cluster, $\Psi$ becomes 0 and $\mathbb{E}[F_t]$ becomes the same as Eq. (19).

2) *Generalization error.* As expert models continuously update at any $t$, $\mathbb{E}[G_T]$ in Eq. (22) comprises three terms: a) the expected model gap $\frac{1}{N^2} \sum_{n \neq n'}^{N} \|\boldsymbol{w}_{n'} - \boldsymbol{w}_n\|^2$ between any two random task arrivals for $t < T_1$, b) the expected model gap $\Psi_1$ between two tasks in the same cluster for $t > T_1$, and c) the expected model gap $\Psi_2$ between a random task arrival for $t < T_1$ and any task arrival in a fixed ground-truth cluster $\mathcal{W}_k$ for $t > T_1$. If each cluster contains only one task, the coefficient of $\Psi_2$ simplifies to $\sum_{\tau=T_1+1}^{T} (1 - r^{L_{T_1}^{(m_\tau)}})$, given $L_T^{(m_\tau)} = L_{T_1}^{(m_\tau)}$ without updates after $T_1$. Moreover, $\Psi_2 = \frac{1}{N^2} \sum_{n \neq n'}^{N} \|\boldsymbol{w}_{n'} - \boldsymbol{w}_n\|^2$ and $\Psi_1 = 0$, resulting in Eq. (22) specializing to Eq. (20).

Note that under our assumption $\|\boldsymbol{w}_n - \boldsymbol{w}_{n'}\|_\infty = \mathcal{O}(\sigma_0^{1.5})$, the router cannot distinguish between similar tasks $n$ and $n'$ within the same cluster. Consequently, adding more experts cannot avoid the errors $\Psi_1$ and $\Psi_2$ in Eq. (21) and Eq. (22). Therefore, similar to our insights of Theorem 1, when there are enough experts than clusters (i.e., $M = \Omega(K \ln(K))$), adding more experts does not enhance learning performance but delays convergence. Although the learning performance degrades compared to Theorem 1, it still benefits from MoE compared to the single expert in Proposition 4.

Note that if we extend to the top-$k$ routing strategy in Eq. (1), the router will select $k$ experts to train the same data at a time. In this case, the forgetting described in Eq. (21) may decrease, as each expert may handle a smaller cluster of tasks compared to the case with top-1 routing strategy. However, similar tasks that belong to the same cluster in the top-1 case now may be divided into different clusters and handled by different expert, which may reduce the potential positive knowledge transfer among these tasks. Consequently, the generalization error in Eq. (22) may not be smaller for the top-$k$ case.

## 6 EXPERIMENTS

In this section, we present extensive experiments on both linear models and DNNs to validate our theoretical analysis. Due to space constraints, we include detailed experimental setups and additional results on datasets such as MNIST LeCun et al. (1989), CIFAR-100 Krizhevsky et al. (2009) and Tiny ImageNet Le & Yang (2015) in Appendix A.

**Key design of early termination.** In the first experiment, we aim to check the necessity of terminating the update of $\Theta_t$ in Line 11 of Algorithm 1. Here we set $T = 2000, N = 6, K = 3$ and vary

expert number $M \in \{1, 5, 10, 20\}$. As depicted in Figure 2(a) and Figure 2(c), both forgetting and generalization error first increase due to the expert exploration and then converge to almost zero for all MoE models with termination of update, verifying Theorem 1 and Theorem 2. In stark contrast, learning without termination leads to poor performance with large oscillations in Figure 2(b) and Figure 2(d), as the router selects the wrong expert for a new task arrival after the continual update of $\mathbf{\Theta}_t$. In addition, in both Figure 2(a) and Figure 2(c), the MoE model significantly improves the performance of CL compared to a single model. The comparison between $M = 10$ and $M = 20$ also indicates that adding extra experts delays the convergence if $M > N$, which does not improve learning performance, verifying our analysis in Theorem 1 and Theorem 2.

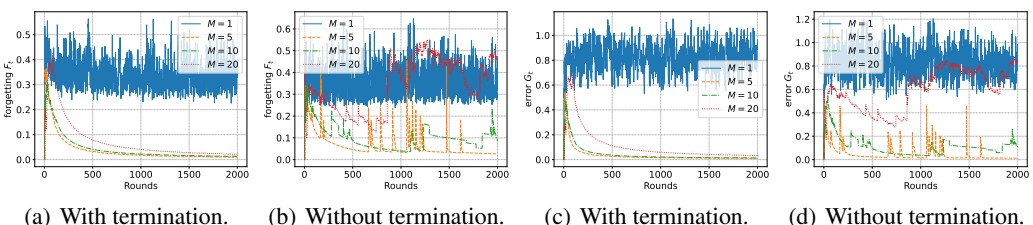

   (a) With termination.    (b) Without termination.    (c) With termination.    (d) Without termination.

Figure 2: The dynamics of forgetting and overall generalization errors with and without termination of updating $\mathbf{\Theta}_t$ in Algorithm 1. Here we set $N = 6$ with $K = 3$ clusters and vary $M \in \{1, 5, 10, 20\}$.

**Real-data validation**. Finally, we extend our Algorithm 1 and insights from linear models to DNNs by conducting experiments on the CIFAR-10 dataset (Krizhevsky et al. (2009)). The details of our experiment setup is given in Appendix A.3. We set $K = 4, N = 300$ and vary $M \in \{1, 4, 12\}$. In each training round, to diversify the model gaps of different tasks, we transform the $d \times d$ matrix into a $d \times d$ dimensional normalized vector to serve as input for the gating network. Then we calculate the variance $\sigma_0$ of each element across all tasks from the input vector, which is then used for parameter setting in Algorithm 1. Figure 3 illustrates that our theoretical insights from linear models also hold for DNNs, in terms of the impact of MoE and early termination on the performance of CL in practice.

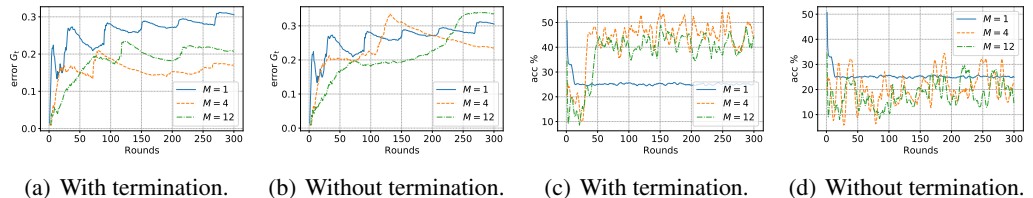

   (a) With termination.    (b) Without termination.    (c) With termination.    (d) Without termination.

Figure 3: The dynamics of overall generalization error and test accuracy under the CIFAR-10 dataset (Krizhevsky et al. (2009)). Here we set $K = 4, N = 300$ and $M \in \{1, 4, 12\}$.

# 7 CONCLUSION

In this work, we conducted the first theoretical analysis of MoE and its impact on learning performance in CL, focusing on an overparameterized linear regression problem. We establish the benefit of MoE over a single expert by proving that the MoE model can diversify its experts to specialize in different tasks, while its router learns to select the right expert for each task and balance the loads across all experts. Then we demonstrated that, under CL, terminating the updating of gating network parameters after sufficient training rounds is necessary for system convergence. Furthermore, we provided explicit forms of the expected forgetting and overall generalization error to assess the impact of MoE. Interestingly, adding more experts requires additional rounds before convergence, which may not enhance the learning performance. Finally, we conducted experiments on real datasets using DNNs to show that certain insights can extend beyond linear models.

ACKNOWLEDGMENTS

This work has been supported in part by the U.S. National Science Foundation under the grants: NSF AI Institute (AI-EDGE) 2112471, CNS-2312836, and was sponsored by the Army Research Laboratory under Cooperative Agreement Number W911NF-23-2-0225. The views and conclusions contained in this document are those of the authors and should not be interpreted as representing the official policies, either expressed or implied, of the Army Research Laboratory or the U.S. Government. The U.S. Government is authorized to reproduce and distribute reprints for Government purposes notwithstanding any copyright notation herein.

This work was also supported in part by the Ministry of Education, Singapore, under its Academic Research Fund Tier 2 Grant with Award no. MOE-T2EP20121-0001; in part by SUTD Kickstarter Initiative (SKI) Grant with no. SKI 2021_04_07; and in part by the Joint SMU-SUTD Grant with no. 22-LKCSB-SMU-053.

The work of Yingbin Liang was supported in part by the U.S. National Science Foundation with the grants RINGS-2148253, CNS-2112471, and ECCS-2413528.

We sincerely thank Qinhang Wu for his invaluable assistance in conducting extensive experiments during our rebuttal, which greatly contributed to the acceptance of this paper.

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

# APPENDIX

## A EXPERIMENTAL DETAILS AND ADDITIONAL EXPERIMENTS

### A.1 EXPERIMENTS COMPUTE RESOURCES

Operating system: Red Hat Enterprise Linux Server 7.9 (Maipo)

Type of CPU: 2.9 GHz 48-Core Intel Xeon 8268s

Type of GPU: NVIDIA Volta V100 w/32 GB GPU memory

### A.2 EXPERIMENTAL DETAILS OF FIGURE 2

**Synthetic data generation**. We first generate $N$ ground truths and their corresponding feature signals. For each ground truth $\boldsymbol{w}_n \in \mathbb{R}^d$, where $n \in [N]$, we randomly generate $d$ elements by a normal distribution $\mathcal{N}(0, \sigma_0)$. These ground truths are then scaled by a constant to obtain their feature signals $\boldsymbol{v}_n$. In each training round $t$, we generate $(\mathbf{X}_t, \mathbf{y}_t)$ according to Definition 1 based on ground-truth pool $\mathcal{W}$ and feature signals. Specifically, after drawing $\boldsymbol{w}_{n_t}$ from $\mathcal{W}$, for $\mathbf{X}_t \in \mathbb{R}^{d \times s}$, we randomly select one out of $s$ samples to fill with $\mathbf{v}_{n_t}$. The other $s-1$ samples are generated from $\mathcal{N}(\mathbf{0}, \sigma_t^2 \boldsymbol{I}_d)$. Finally, we compute the output $\mathbf{y}_t = \mathbf{X}_t^\top \boldsymbol{w}_{n_t}$. Here we set $\sigma_0 = 0.4, \sigma_t = 0.1, d = 10$ and $s = 6$. In Figure 2, we set $\eta = 0.5, \alpha = 0.5$ and $\lambda = 0.3$.

### A.3 EXPERIMENTAL DETAILS OF FIGURE 3

*Datasets.* We use the CIFAR-10 (Krizhevsky et al. (2009)) dataset, selecting 512 samples randomly for training and 2000 samples for testing at each training round.

*DNN architecture and training details.* We employ a non-pretrained ResNet-18 as our base model. Each task is learned using Adam with a learning rate governed by a cosine annealing schedule for 5 epoches, with a minibatch size of 32, a weight decay of 0.005. The initial learning rate is set to 0.0005, and it is reduced to a minimum value of $10^{-6}$ over a total of 300 rounds.

*Task setups.* We define the ground truth pool as $\mathcal{W} = \{(0), (4), (5), (9)\}$, representing $K = 4$ clusters of tasks for recognizing the image classes airplane, deer, dog, truck, respectively. The experiment spans $T = 300$ training rounds with $N = 300$ tasks. We randomly generate the task arrival sequence $[n_t]_{t \in [T]}$, where each $n_t$ is drawn from $(0), (4), (5), (9)$ with equal probability $\frac{1}{4}$. We then conduct two experiments (with and without termination) using the same task arrival order. For each task $t \in [T]$, we randomly select its type (e.g., task $(0)$ for recognizing the airplane class image) and 512 corresponding samples, ensuring that tasks have distinct distributions and features.

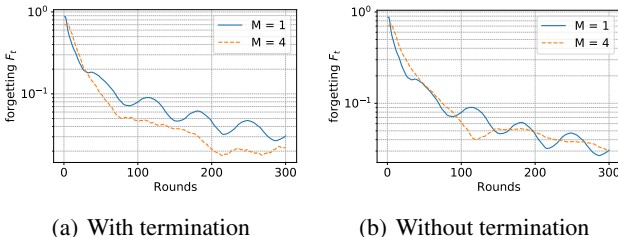

(a) With termination          (b) Without termination

Figure 4: The dynamics of forgetting under the CIFAR-10 dataset. Here we set $N = 4$ and $M \in \{1, 4\}$.

### A.4 EXPERIMENTS ON THE MNIST DATASET

*Datasets.* We use the MNIST dataset LeCun et al. (1989), selecting 100 samples randomly for training and 1000 samples for testing at each training round.

*DNN architecture and training details.* We use a five-layer neural network, consisting of two convolutional layers and three fully connected layers. ReLU activation is applied to the first four layers, while Sigmoid is used for the final layer. The first convolutional layer is followed by a 2D max-pooling operation with a stride of 2. Each task is learned using SGD with a learning rate of

0.2 for 600 epochs. The forgetting and overall generalization error are evaluated as described in Eq. (15) and Eq. (16), respectively. Here, $\mathcal{E}_t(\boldsymbol{w}_t^{(m_t)})$ is defined as the mean-squared test error instead of Eq. (14).

*Task setups.* We define the ground truth pool as $\mathcal{W} = \{(1), (4), (7)\}$, representing $K = 3$ clusters of tasks for recognizing the numbers 1, 4, and 7, respectively. The experiment spans $T = 60$ training rounds with $N = 60$ tasks. Before the experiments in Figure 5, we randomly generate the task arrival sequence $[n_t]_{t \in [T]}$, where each $n_t$ is drawn from $(1), (4), (7)$ with equal probability $\frac{1}{3}$. We then conduct two experiments (with and without termination) using the same task arrival order. For each task $t \in [T]$, we randomly select its type (e.g., task $(1)$ for recognizing the number 1) and 100 corresponding samples, ensuring that tasks have distinct distributions and features.

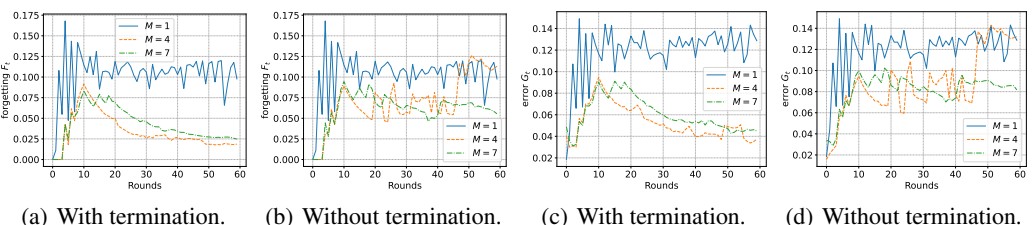

(a) With termination.     (b) Without termination.     (c) With termination.     (d) Without termination.

Figure 5: Learning performance under the MNIST dataset (LeCun et al. (1989)). Here we set $K = 3, N = 60$ and $M \in \{1, 4, 7\}$.

## A.5 EXPERIMENTS ON THE CIFAR-100 DATASET

*Datasets.* We use the CIFAR-100 (Krizhevsky et al. (2009)) dataset, selecting 192 samples randomly for training and 600 samples for testing at each training round.

*DNN architecture and training details.* They are the same as the experiments on the CIFAR-10 dataset in Appendix A.3.

*Task setups.* We define the ground truth pool as $\mathcal{W} = \{(28), (40), (52), (72), (79), (99)\}$, representing $K = 6$ clusters of tasks for recognizing the image classes telephone, bee, mountain, bear, turtle, tractor respectively. The experiment spans $T = 350$ training rounds with $N = 350$ tasks. We randomly generate the task arrival sequence $[n_t]_{t \in [T]}$, where each $n_t$ is drawn from $(28), (40), (52), (72), (79), (99)$ with equal probability $\frac{1}{6}$. We then conduct two experiments (with and without termination) using the same task arrival order. For each task $t \in [T]$, we randomly select its type (e.g., task $(28)$ for recognizing the telephone class image) and 192 corresponding samples, ensuring that tasks have distinct distributions and features.

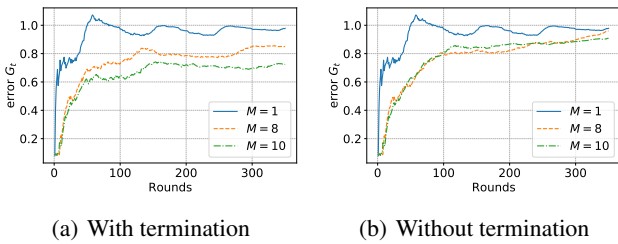

(a) With termination     (b) Without termination

Figure 6: Learning performance under the CIFAR-100 dataset. Here we set $N = 6$ and $M \in \{1, 8, 10\}$.

## A.6 EXPERIMENTS ON THE TINY IMAGENET DATASET

*Datasets.* We use the Tiny ImageNet (Le & Yang (2015)) dataset, selecting 192 samples randomly for training and 300 samples for testing at each training round.

Table 1: The average incremental accuracy under the CIFAR-100 dataset.

| Expert number | Test accuracy (%) |
|---|---|
| $M = 1$ | 17.1 |
| $M = 8$ | 25.5 |
| $M = 8$ w/o ET | 10.3 |
| $M = 10$ | 32.9 |
| $M = 10$ w/o ET | 9.1 |

*DNN architecture and training details.* They are the same as the experiments on the CIFAR-10 dataset in Appendix A.3.

*Task setups.* We define the ground truth pool as $\mathcal{W} = \{(20), (50), (83), (145), (168), (179)\}$, representing $K = 6$ clusters of tasks for recognizing six disjoint image classes from Tiny Imagenet respectively. The experiment spans $T = 300$ training rounds with $N = 300$ tasks. We randomly generate the task arrival sequence $[n_t]_{t \in [T]}$, where each $n_t$ is drawn from $(20), (50), (83), (145), (168), (179)$ with equal probability $\frac{1}{6}$. We then conduct two experiments (with and without termination) using the same task arrival order. For each task $t \in [T]$, we randomly select its type (e.g., task $(20)$) and 192 corresponding samples, ensuring that tasks have distinct distributions and features.

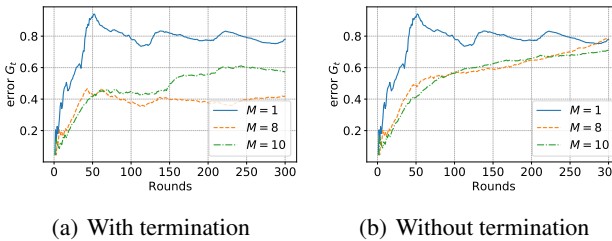

(a) With termination      (b) Without termination

Figure 7: Learning performance under the Tiny ImageNet dataset. Here we set $N = 6$ and $M \in \{1, 8, 10\}$.

Table 2: The average incremental accuracy under the Tiny ImageNet dataset.

| Expert number | Test accuracy (%) |
|---|---|
| $M = 1$ | 17.4 |
| $M = 8$ | 37.6 |
| $M = 8$ w/o ET | 10.5 |
| $M = 10$ | 28.3 |
| $M = 10$ w/o ET | 10.3 |

## A.7 Experiments on termination threshold and load balance

In additional experiments, we vary termination threshold $\Gamma \in \{\sigma_0^{0.75}, \sigma_0, \sigma_0^{1.25}, \sigma_0^{1.5}\}$ in Line 7 of Algorithm 1 to investigate its effect on load balance and learning performance, under the same synthetic data generation as Figure 2 in Appendix A.2.

Initially, we set $\sigma_0 = 0.4$, $\lambda = \sigma_0^{1.25}$, $M = 5$, and $N = 6$ with $K = 3$ task clusters: $\mathcal{W}_1 = \{1, 4\}, \mathcal{W}_2 = \{2, 5\}$, and $\mathcal{W}_3 = \{3, 6\}$. Figure 8 illustrates the recorded task arrivals per round and their routed experts. Figure 8(a) and Figure 8(b) depict that if the MoE model terminates the update based on $\Gamma > \lambda$, the small noise $r_t^{(m)}$ cannot alter the router's decision from the expert with the maximum gate output for each task cluster (e.g., expert 5 for $\mathcal{W}_2 = \{2, 5\}$ in Figure 8(a)) in Eq. (1), leading to imbalanced expert load. While for $\Gamma \leq \lambda$ in Figure 8(c) and Figure 8(d), the random

noise $r_t^{(m)}$ in Eq. (1) can mitigate the gaps of gate outputs among experts within the same expert set, ensuring load balance (e.g., experts 3 and 4 for $\mathcal{W}_2 = \{2, 5\}$ in Figure 8(c)).

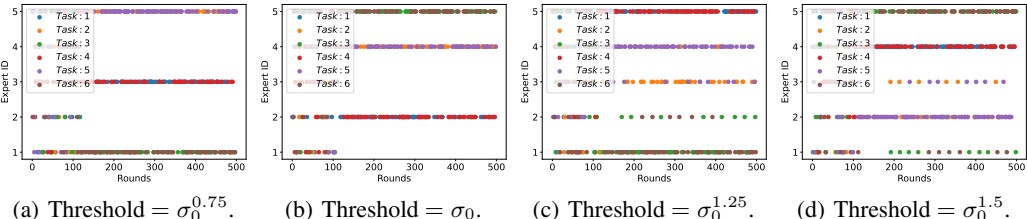

(a) Threshold $= \sigma_0^{0.75}$.    (b) Threshold $= \sigma_0$.    (c) Threshold $= \sigma_0^{1.25}$.    (d) Threshold $= \sigma_0^{1.5}$.

Figure 8: The records of task arrivals and their selected experts under different termination thresholds in Line 7 of Algorithm 1: $\Gamma \in \{\sigma_0^{0.75}, \sigma_0, \sigma_0^{1.25}, \sigma_0^{1.5}\}$. Here we set $M = 5, N = 6$ and $K = 3$.

To further examine how load balance affects learning performance, we increase the task number to $N = 30$. We repeat the experiment 100 times and plot the average forgetting and generalization errors in Figure 9. Figure 9(a) illustrates that the forgetting is robust to a wide range of $\Gamma$, due to the convergence of expert models after $T_1$ rounds' exploration. However, Figure 9(b) shows that balanced loads under $\Gamma \in \{\sigma_0^{1.25}, \sigma_0^{1.5}\}$ lead to smaller generalization errors compared to imbalanced loads under $\Gamma \in \{\sigma_0^{0.75}, \sigma_0^1\}$. This is because diverse expert models help mitigate model errors compared to a single overfitted model.

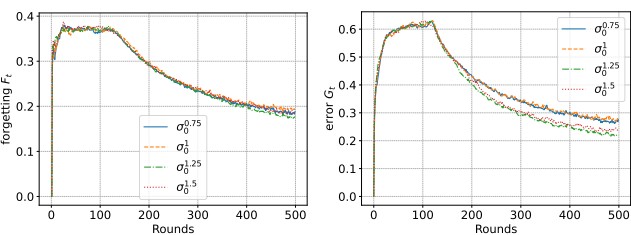

Figure 9: We run the experiment for 100 times to take the average learning performance under four termination thresholds: $\Gamma \in \{\sigma_0^{0.75}, \sigma_0, \sigma_0^{1.25}, \sigma_0^{1.5}\}$. Here we set $M = 5, N = 30$ and $K = 3$.

## B SMOOTH ROUTER

We first prove that Eq. (1) ensures a smooth transition between different routing behaviors, which makes the router more stable. Suppose that there are two different datasets $(\mathbf{X}, \mathbf{y})$ and $(\hat{\mathbf{X}}, \hat{\mathbf{y}})$ simultaneously acting as input of the MoE. Let $\mathbf{h}$ and $\hat{\mathbf{h}}$ denote the corresponding output of the gating network, respectively. Denote the probability vectors by $\mathbf{p}$ and $\hat{\mathbf{p}}$, which tell the probabilities that each expert gets routed for the two datasets. For example, $p_m = \mathbb{P}(\arg\max_{m' \in [M]} \{h_{m'} + r^{(m')}\} = m)$ and $\hat{p}_m = \mathbb{P}(\arg\max_{m' \in [M]} \{\hat{h}_{m'} + r^{(m')}\} = m)$ according to Eq. (1). Then we propose the following lemma to prove the smooth router.

**Lemma 2.** *The two probability vectors satisfy* $\|\mathbf{p} - \hat{\mathbf{p}}\|_\infty \le \lambda M^2 \|\mathbf{h} - \hat{\mathbf{h}}\|_\infty$.

*Proof.* Let $m_1 = \arg\max_m \{h_m + r^{(m)}\}$ and $m_2 = \arg\max_m \{\hat{h}_m + r^{(m)}\}$. We first consider the event that $m_1 \ne m_2$. In this case, we have
$$h_{m_1} + r^{(m_1)} \ge h_{m_2} + r^{(m_2)}, \hat{h}_{m_2} + r^{(m_2)} \ge \hat{h}_{m_1} + r^{(m_1)},$$
which implies that
$$\hat{h}_{m_2} - \hat{h}_{m_1} > r^{(m_1)} - r^{(m_2)} \ge h_{m_2} - h_{m_1}. \tag{23}$$
Define $C(m_1, m_2) = \frac{\hat{h}_{m_2} - \hat{h}_{m_1} + h_{m_2} - h_{m_1}}{2}$. Based on Eq. (23), we obtain
$$|r^{(m_1)} - r^{(m_2)} - C(m_1, m_2)| \le \frac{\hat{h}_{m_2} - \hat{h}_{m_1} - h_{m_2} + h_{m_1}}{2} \le \|\hat{\mathbf{h}} - \mathbf{h}\|_\infty. \tag{24}$$

Therefore, we calculate that $m_1 \neq m_2$ below

$$\mathbb{P}(\arg\max_m\{h_m + r^{(m)}\} \neq \arg\max_m\{\hat{h}_m + r^{(m)}\})$$

$$\leq \mathbb{P}(\exists\, m_1 \neq m_2 \in [M], \text{s.t. } |r^{(m_1)} - r^{(m_2)} - C(m_1, m_2)| \leq \|\hat{\mathbf{h}} - \mathbf{h}\|_\infty)$$

$$\leq \sum_{m_1 < m_2} \mathbb{P}\big(|r^{(m_1)} - r^{(m_2)} - C(m_1, m_2)| \leq \|\hat{\mathbf{h}} - \mathbf{h}\|_\infty\big)$$

$$= \sum_{m_1 < m_2} \mathbb{E}\big[\mathbb{P}(r^{(m_2)} + C(m_1, m_2) - \|\hat{\mathbf{h}} - \mathbf{h}\|_\infty \leq r^{(m_1)} \leq r^{(m_2)} + C(m_1, m_2) + \|\hat{\mathbf{h}} - \mathbf{h}\|_\infty)|r^{(m_2)}\big]$$

$$\leq \lambda M^2 \|\hat{\mathbf{h}} - \mathbf{h}\|_\infty,$$

where the first inequality is derived by Eq. (24), the second inequality is because of union bound, and the last inequality is due to the fact that $r^{(m)}$ is drawn from Unif $[0, \lambda]$.

Then for any $j \in [M]$, we have

$$|\hat{p}_i - p_i| \leq \left|\mathbb{E}\left[\mathbb{1}(\arg\max_m\{\hat{h}_m + r^{(m)} = i\}) - \mathbb{1}(\arg\max_m\{\hat{h}_m + r^{(m)} = i\})\right]\right|$$

$$\leq \mathbb{E}\left[\left|\mathbb{1}(\arg\max_m\{\hat{h}_m + r^{(m)} = i\}) - \mathbb{1}(\arg\max_m\{\hat{h}_m + r^{(m)} = i\})\right|\right]$$

$$\leq \mathbb{P}(\arg\max_m\{h_m + r^{(m)}\} \neq \arg\max_m\{\hat{h}_m + r^{(m)}\})$$

$$\leq M^2 \|\hat{\mathbf{h}} - \mathbf{h}\|_\infty.$$

This completes the proof of Lemma 2. $\qquad\square$

Intuitively, Lemma 2 tells that if the outputs $\mathbf{h}$ and $\hat{\mathbf{h}}$ of the gating network are similar for two tasks, their data sets $(\mathbf{X}, \mathbf{y})$ and $(\hat{\mathbf{X}}, \hat{\mathbf{y}})$ will be routed to the same expert with a high probability. It means that the router transitions are smooth and continuous.

## C    FULL VERSION AND PROOF OF LEMMA 1

**Lemma 1** (Full version). *For any two feature matrices $\mathbf{X}$ and $\tilde{\mathbf{X}}$ with feature signals $\boldsymbol{v}_n$ and $\boldsymbol{v}'_n$, if $\boldsymbol{w}_n = \boldsymbol{w}_{n'}$ under $M > N$ or $\boldsymbol{w}_n, \boldsymbol{w}_{n'} \in \mathcal{W}_k$ under $M < N$, with probability at least $1 - o(1)$, their corresponding gate outputs of the same expert $m$ satisfy*

$$\left|h_m(\mathbf{X}, \boldsymbol{\theta}_t^{(m)}) - h_m(\tilde{\mathbf{X}}, \boldsymbol{\theta}_t^{(m)})\right| = \mathcal{O}(\sigma_0^{1.5}). \tag{25}$$

*Proof.* We first focus on the $M > N$ case to prove Lemma 1. Then we consider the $M < N$ case to prove Lemma 1. For dataset $(\mathbf{X}_t, \mathbf{y}_t)$ generated in Definition 1 per round $t$, we can assume that the first sample of $\mathbf{X}_t$ is the signal vector. Therefore, we rewrite $\mathbf{X}_t = [\beta_t \boldsymbol{v}_n\, \mathbf{X}_{t,2}\, \cdots\, \mathbf{X}_{t,s}]$. Let $\tilde{\mathbf{X}}_t = [\beta_t \boldsymbol{v}_{n_t}\, 0\, \cdots\, 0]$ represents the matrix that only keeps the feature signal.

Based on the definition of the gating network in Section 3, we have $h_m(\mathbf{X}_t, \boldsymbol{\theta}_t^{(m)}) = \sum_{i=1}^s (\boldsymbol{\theta}_t^{(m)})^\top \mathbf{X}_{t,i}$. Then we calculate

$$\left|h_m(\mathbf{X}_t, \boldsymbol{\theta}_t^{(m)}) - h_m(\tilde{\mathbf{X}}_t, \boldsymbol{\theta}_t^{(m)})\right| = \left|(\boldsymbol{\theta}_t^{(m)})^\top \sum_{i=2}^s \mathbf{X}_{t,i}\right|$$

$$= \left|\sum_{i=2}^s \sum_{j=1}^d (\theta_{t,j}^{(m)})^\top X_{t,(i,j)}\right|$$

$$\leq \left|\max_{t,j}\{\theta_{t,j}^{(m)}\}\right| \cdot \left|\sum_{i=2}^s \sum_{j=1}^d X_{t,(i,j)}\right|,$$

where $\theta_{t,j}^{(m)}$ is the $j$-th element of vector $\boldsymbol{\theta}_t^{(m)}$ and $X_{t,(i,j)}$ is the $(i, j)$-th element of matrix $\mathbf{X}_t$.

Then we apply Hoeffding's inequality to obtain

$$\mathbb{P}\Big(|\sum_{i=2}^{s}\sum_{j=1}^{d}X_{t,(i,j)}| < s \cdot d \cdot \sigma_0\Big) \geq 1 - 2\exp{(-\frac{\sigma_0^2 s^2 d^2}{\|\mathbf{X}_{t,i}\|_\infty})}.$$

As $X_{t,(i,j)} \sim \mathcal{N}(0, \sigma_t^2)$, we have $\|\mathbf{X}_{t,i}\|_\infty = \mathcal{O}(\sigma_t)$, indicating $\exp{(-\frac{\sigma_0^2 s^2 d^2}{\|\mathbf{X}_{t,i}\|_\infty})} = o(1)$. Therefore, with probability at least $1 - o(1)$, we have $\big|\sum_{i=2}^{s}\sum_{j=1}^{d}X_{t,(i,j)}\big| = \mathcal{O}(\sigma_0)$. Consequently, we obtain $\big|h_m(\mathbf{X}_t, \boldsymbol{\theta}_t^{(m)}) - h_m(\tilde{\mathbf{X}}_t, \boldsymbol{\theta}_t^{(m)})\big| = \mathcal{O}(\sigma_0^{1.5})$ due to the fact that $\big|\sum_{i=2}^{s}\sum_{j=1}^{d}X_{t,(i,j)}\big| = \mathcal{O}(\sigma_0)$ and $\theta_{t,j}^{(m)} = \mathcal{O}(\sigma_0^{0.5})$ proven in Lemma 6 later.

If $M < N$, we calculate

$$\begin{aligned}
\Big|h_m(\mathbf{X}_t, \boldsymbol{\theta}_t^{(m)}) - h_m(\tilde{\mathbf{X}}_t, \boldsymbol{\theta}_t^{(m)})\Big| &= \Big|(\boldsymbol{\theta}_t^{(m)})^\top \sum_{i=1}^{s}\mathbf{X}_{t,i}\Big| \\
&\leq \Big|(\boldsymbol{\theta}_t^{(m)})^\top \sum_{i=2}^{s}\mathbf{X}_{t,i}\Big| + \Big|(\boldsymbol{\theta}_t^{(m)})^\top(\boldsymbol{v}_n - \boldsymbol{v}_{n'})\Big| \\
&\leq \Big|\max_{t,j}\{\theta_{t,j}^{(m)}\}\Big| \cdot \Big|\sum_{i=2}^{s}\sum_{j=1}^{d}X_{t,(i,j)}\Big| + \mathcal{O}(\sigma_0^2) \\
&= \mathcal{O}(\sigma_0^{1.5}),
\end{aligned}$$

where the second inequality is based on the union bound, the third inequality is because of $\theta_{t,j}^{(m)} = \mathcal{O}(\sigma_0^{0.5})$ and our assumption $\|\boldsymbol{v}_n - \boldsymbol{v}_{n'}\|_\infty = \mathcal{O}(\sigma_0^{1.5})$, and the last inequality is based on our proof of the $M > N$ case above. This completes the proof of the full version of Lemma 1. $\qquad\square$

Based on Lemma 1, we revisit Lemma 2 to obtain the following conclusion for the smooth router.

**Corollary 1.** *If datasets $(\mathbf{X}, \mathbf{y})$ and $(\hat{\mathbf{X}}, \hat{\mathbf{y}})$ are generated by the same ground truth, then the two probability vectors in Lemma 2 satisfy $\|\mathbf{p} - \hat{\mathbf{p}}\|_\infty = \mathcal{O}\left(\lambda M^2 \sigma_0^{1.5}\right)$.*

## D  ANALYSIS OF LOSS FUNCTION

In this section, we analyze the loss function for both gating network parameters and expert models before analyzing MoE.

**Lemma 3.** *Under update rule Eq. (5), if the current task $n_t$ routed to expert $m_t$ have the same ground truth with the last task $n_\tau$, where $\tau < t$, routed to expert $m_t$, i.e., $\boldsymbol{w}_{n_t} = \boldsymbol{w}_{n_\tau}$, then the model of expert $m_t$ satisfies $\boldsymbol{w}_t^{(m_t)} = \boldsymbol{w}_{t-1}^{(m_t)} = \cdots = \boldsymbol{w}_\tau^{(m_t)}$.*

It is easy to prove Lemma 3 by the updating rule Eq. (5) such that we skip the proof here.

Next, we examine the training of the gating network parameter.

**Lemma 4.** *For any training round $t \geq 1$, we have that $\sum_{m=1}^{M}\nabla_{\boldsymbol{\theta}_t^{(m)}}\mathcal{L}_t^{task} = \mathbf{0}$.*

*Proof.* As the training loss $\nabla_{\boldsymbol{\theta}^{(m)}}\mathcal{L}_t^{tr} = 0$, we obtain

$$\nabla_{\boldsymbol{\theta}^{(m)}}\mathcal{L}_t^{task} = \nabla_{\boldsymbol{\theta}^{(m)}}\mathcal{L}_t^{loc} + \nabla_{\boldsymbol{\theta}^{(m)}}\mathcal{L}_t^{aux}. \tag{26}$$

Next, we will prove $\sum_{m=1}^{M}\nabla_{\boldsymbol{\theta}_t^{(m)}}\mathcal{L}_t^{loc} = \mathbf{0}$ and $\sum_{m=1}^{M}\nabla_{\boldsymbol{\theta}_t^{(m)}}\mathcal{L}_t^{loc} = \mathbf{0}$, respectively. Based on the two equations, we can prove Lemma 4.

According to the definition of locality loss in Eq. (6), we calculate

$$\nabla_{\boldsymbol{\theta}_t^{(m)}}\mathcal{L}_t^{loc} = \frac{\partial \pi_{m_t}(\mathbf{X}_t, \boldsymbol{\Theta}_t)}{\partial \boldsymbol{\theta}_t^{(m)}}\|\boldsymbol{w}_t^{(m_t)} - \boldsymbol{w}_{t-1}^{(m_t)}\|_2. \tag{27}$$

If $m = m_t$, we obtain

$$\frac{\partial \pi_{m_t}(\mathbf{X}_t, \boldsymbol{\Theta}_t)}{\partial \boldsymbol{\theta}_t^{(m)}} = \pi_{m_t}(\mathbf{X}_t, \boldsymbol{\Theta}_t) \cdot \Big( \sum_{m' \neq m_t} \pi_{m'}(\mathbf{X}_t, \boldsymbol{\Theta}_t) \Big) \cdot \frac{\partial h_m(\mathbf{X}_t, \boldsymbol{\theta}_t^{(m)})}{\partial \boldsymbol{\theta}_t^{(m)}}$$

$$= \pi_{m_t}(\mathbf{X}_t, \boldsymbol{\Theta}_t) \cdot \Big( \sum_{m' \neq m_t} \pi_{m'}(\mathbf{X}_t, \boldsymbol{\Theta}_t) \Big) \cdot \sum_{i \in [s_t]} \mathbf{X}_{t,i}. \tag{28}$$

If $m \neq m_t$, we obtain

$$\frac{\partial \pi_{m_t}(\mathbf{X}_t, \boldsymbol{\Theta}_t)}{\partial \boldsymbol{\theta}_t^{(m)}} = -\pi_{m_t}(\mathbf{X}_t, \boldsymbol{\Theta}_t) \cdot \pi_m(\mathbf{X}_t, \boldsymbol{\Theta}_t) \cdot \sum_{i \in [s_t]} \mathbf{X}_{t,i}. \tag{29}$$

Based on Eq. (27), Eq. (28) and Eq. (29), we obtain

$$\sum_{m=1}^{M} \nabla_{\boldsymbol{\theta}_t^{(m)}} \mathcal{L}_t^{loc} = \|\boldsymbol{w}_t^{(m_t)} - \boldsymbol{w}_{t-1}^{(m_t)}\|_2 \sum_{m=1}^{M} \frac{\partial \pi_{m_t}(\mathbf{X}_t, \boldsymbol{\Theta}_t)}{\partial \boldsymbol{\theta}_t^{(m)}} = \mathbf{0}.$$

According to the definition of auxiliary loss in Eq. (7), we calculate

$$\nabla_{\boldsymbol{\theta}_t^{(m)}} \mathcal{L}_t^{aux} = \alpha M \sum_{m'=1}^{M} f_t^{(m')} \cdot \frac{\partial P_t^{(m')}}{\partial \boldsymbol{\theta}_t^{(m)}}$$

$$= \frac{\alpha M}{t} f_t^{(m_t)} \cdot \frac{\partial \pi_{m_t}(\mathbf{X}_t, \boldsymbol{\Theta}_t)}{\partial \boldsymbol{\theta}_t^{(m)}}, \tag{30}$$

where the second equality is due to the fact that $\frac{\partial P_t^{(m_t)}}{\partial \boldsymbol{\theta}_t^{(m)}} = \frac{1}{t} \cdot \frac{\partial \pi_{m_t}(\mathbf{X}_t, \boldsymbol{\Theta}_t)}{\partial \boldsymbol{\theta}_t^{(m)}}$ and $\frac{\partial P_t^{(m')}}{\partial \boldsymbol{\theta}_t^{(m)}} = 0$ for any $m' \neq m_t$ by Eq. (7). Then based on Eq. (28) and Eq. (29), we similarly obtain

$$\sum_{m=1}^{M} \nabla_{\boldsymbol{\theta}_t^{(m)}} \mathcal{L}_t^{aux} = \frac{\alpha M}{t} f_t^{(m_t)} \sum_{m=1}^{M} \frac{\partial \pi_{m_t}(\mathbf{X}_t, \boldsymbol{\Theta}_t)}{\partial \boldsymbol{\theta}_t^{(m)}} = \mathbf{0}.$$

In summary, we finally prove $\sum_{m=1}^{M} \nabla_{\boldsymbol{\theta}_t^{(m)}} \mathcal{L}_t^{task} = \mathbf{0}$ in Eq. (26). $\qquad\square$

In the following lemma, we analyze the gradient of loss function with respect to each expert.

**Lemma 5.** *For any training round $t \in \{1, \cdots, T\}$, the following property holds*

$$\|\nabla_{\boldsymbol{\theta}_t^{(m)}} \mathcal{L}_t^{task}\|_\infty = \begin{cases} \Omega(\sigma_0), & \text{if } t \in \{1, \cdots, T_1\}, \\ \mathcal{O}(\sigma_0), & \text{if } t \in \{T_1 + 1, \cdots, T\} \end{cases} \tag{31}$$

*for any expert $m \in [M]$, where $T_1 = \lceil \eta^{-1} M \rceil$ is the length of the exploration stage.*

*Proof.* We prove Eq. (31) by analyzing $\nabla_{\boldsymbol{\theta}^{(m)}} \mathcal{L}_t^{loc} = \mathcal{O}(\sigma_0)$ and $\nabla_{\boldsymbol{\theta}^{(m)}} \mathcal{L}_t^{aux} = \mathcal{O}(\frac{\alpha M}{t})$ in Eq. (26), respectively.

First, we calculate $\|\nabla_{\boldsymbol{\theta}_t^{(m)}} \mathcal{L}_t^{loc}\|_\infty$. Based on Eq. (27), we have

$$\nabla_{\boldsymbol{\theta}_t^{(m)}} \mathcal{L}_t^{loc} = \frac{\partial \pi_{m_t}(\mathbf{X}_t, \boldsymbol{\Theta}_t)}{\partial \boldsymbol{\theta}_t^{(m)}} \|\boldsymbol{w}_t^{(m_t)} - \boldsymbol{w}_{t-1}^{(m_t)}\|_2$$

$$= \mathbb{1}\{\boldsymbol{w}_{n_\tau} = \boldsymbol{w}_{n_t}\} \cdot 0 + \mathbb{1}\{\boldsymbol{w}_{n_\tau} \neq \boldsymbol{w}_{n_t}\} \cdot \|\boldsymbol{w}_t^{(m_t)} - \boldsymbol{w}_{t-1}^{(m_t)}\|_2 \frac{\partial \pi_{m_t}(\mathbf{X}_t, \boldsymbol{\Theta}_t)}{\partial \boldsymbol{\theta}_t^{(m)}}$$

$$\leq \|\boldsymbol{w}_t^{(m_t)} - \boldsymbol{w}_{t-1}^{(m_t)}\|_2 \frac{\partial \pi_{m_t}(\mathbf{X}_t, \boldsymbol{\Theta}_t)}{\partial \boldsymbol{\theta}_t^{(m)}},$$

where $\tau$ is the index of the last task that routed to expert $m_t$, the second equality is derived by Lemma 3. As $\frac{\partial \pi_{m_t}(\mathbf{X}_t, \boldsymbol{\Theta}_t)}{\partial \boldsymbol{\theta}_t^{(m)}} = \mathcal{O}(1)$ and $\boldsymbol{w}_n \sim \mathcal{N}(\mathbf{0}, \boldsymbol{\sigma}_0^2)$, we finally obtain

$$\|\nabla_{\boldsymbol{\theta}_t^{(m)}} \mathcal{L}_t^{loc}\|_\infty = \mathcal{O}(\sigma_0).$$

Next, we further calculate $\|\nabla_{\boldsymbol{\theta}_t^{(m)}} \mathcal{L}_t^{aux}\|_\infty$, which contains the following two cases.

If $t \in \{1, \cdots, T_1\}$, by Eq. (30), we have

$$\|\nabla_{\boldsymbol{\theta}_t^{(m)}} \mathcal{L}_t^{aux}\|_\infty \geq \|\frac{\alpha M}{T_1} f_t^{(m_t)} \cdot \frac{\partial \pi_{m_t}(\mathbf{X}_t, \boldsymbol{\Theta}_t)}{\partial \boldsymbol{\theta}_t^{(m)}}\|_\infty$$

$$\geq \|\sigma_0 f_t^{(m_t)} \cdot \frac{\partial \pi_{m_t}(\mathbf{X}_t, \boldsymbol{\Theta}_t)}{\partial \boldsymbol{\theta}_t^{(m)}}\|_\infty = \Omega(\sigma_0),$$

where the second inequality is derived by setting $\eta = \Omega(\sigma_0^{0.5})$ to make $T_1 = \lceil \sigma_0^{-0.5} M \rceil$.

If $t \in \{T_1 + 1, \cdots, T\}$, we calculate

$$\|\nabla_{\boldsymbol{\theta}_t^{(m)}} \mathcal{L}_t^{aux}\|_\infty \leq \|\frac{\alpha M}{T_1} f_t^{(m_t)} \cdot \frac{\partial \pi_{m_t}(\mathbf{X}_t, \boldsymbol{\Theta}_t)}{\partial \boldsymbol{\theta}_t^{(m)}}\|_\infty$$

$$= \mathcal{O}(\sigma_0).$$

Based on the derived $\|\nabla_{\boldsymbol{\theta}_t^{(m)}} \mathcal{L}_t^{loc}\|_\infty$ and $\|\nabla_{\boldsymbol{\theta}_t^{(m)}} \mathcal{L}_t^{aux}\|_\infty$ above, we can finally calculate $\|\nabla_{\boldsymbol{\theta}_t^{(m)}} \mathcal{L}_t^{task}\|_\infty$ based on Eq. (26).

For $t \in \{1, \cdots, T_1\}$, if $m \neq m_t$, we have

$$\|\nabla_{\boldsymbol{\theta}_t^{(m)}} \mathcal{L}_t^{task}\|_\infty = \|\nabla_{\boldsymbol{\theta}_t^{(m)}} \mathcal{L}_t^{loc} + \nabla_{\boldsymbol{\theta}_t^{(m)}} \mathcal{L}_t^{aux}\|_\infty$$

$$\leq \|\mathcal{O}(\sigma_0) + \frac{\alpha M}{T_1} f_t^{(m_t)} \cdot \frac{\partial \pi_{m_t}(\mathbf{X}_t, \boldsymbol{\Theta}_t)}{\partial \boldsymbol{\theta}_t^{(m)}}\|_\infty$$

$$= \Omega(\sigma_0).$$

Similarly, for any $t \in \{T_1 + 1, \cdots, T\}$, we can can obtain

$$\|\nabla_{\boldsymbol{\theta}_t^{(m)}} \mathcal{L}_t^{task}\|_\infty = \|\nabla_{\boldsymbol{\theta}_t^{(m)}} \mathcal{L}_t^{loc} + \nabla_{\boldsymbol{\theta}_t^{(m)}} \mathcal{L}_t^{aux}\|_\infty$$

$$\geq \|\mathcal{O}(\sigma_0) + \frac{\alpha M}{T_1} f_t^{(m_t)} \cdot \frac{\partial \pi_{m_t}(\mathbf{X}_t, \boldsymbol{\Theta}_t)}{\partial \boldsymbol{\theta}_t^{(m)}}\|_\infty$$

$$= \mathcal{O}(\sigma_0).$$

This completes the proof of Eq. (31). $\qquad \square$

Given $\boldsymbol{\theta}_0^{(m)} = \mathbf{0}$ for any expert $m \in [M]$, in the next lemma, we obtain the upper bound of $\boldsymbol{\theta}_t^{(m)}$ at any round $t \in \{1, \cdots, T\}$.

**Lemma 6.** *For any training round $t \in \{1, \cdots, T\}$, the gating network parameter of any expert $m$ satisfies $\|\boldsymbol{\theta}_t^{(m)}\|_\infty = \mathcal{O}(\sigma_0^{0.5})$.*

*Proof.* Based on Lemma 5, for any $t \in \{1, \cdots, T_1\}$ the accumulated update of $\boldsymbol{\theta}_t^{(m)}$ throughout the exploration stage satisfies

$$\|\boldsymbol{\theta}_t^{(m)}\|_\infty \leq \eta \cdot T_1 \cdot \alpha M = \mathcal{O}(\sigma_0^{0.5}).$$

For any $t \in \{T_1 + 1, \cdots, T\}$, the accumulated update of $\boldsymbol{\theta}_t^{(m)}$ throughout the router learning phase satisfies

$$\|\boldsymbol{\theta}_t^{(m)}\|_\infty \leq \|\boldsymbol{\theta}_{T_1}^{(m)}\|_\infty + \eta \cdot (T - T_1) \cdot \frac{\alpha M}{T_1}$$

$$= \mathcal{O}(\sigma_0^{0.5}) + \mathcal{O}(\sigma_0^{0.5} - \sigma_0) = \mathcal{O}(\sigma_0^{0.5}).$$

In summary, $\|\boldsymbol{\theta}_t^{(m)}\|_\infty = \mathcal{O}(\sigma_0^{0.5})$ for any round $t \in \{1, \cdots, T\}$. $\qquad \square$

## E   FULL VERSION AND PROOF OF PROPOSITION 1

**Proposition 1** (Full version). *Under Algorithm 1, with probability at least $1 - o(1)$, for any $t > T_1$, where $T_1 = \lceil \eta^{-1} M \rceil$, each expert $m \in [M]$ satisfies the following properties:*

*1) If $M > N$, expert $m$ stabilizes within an expert set $\mathcal{M}_n$, and its expert model remains unchanged beyond time $T_1$, satisfying $\boldsymbol{w}_{T_1+1}^{(m)} = \cdots = \boldsymbol{w}_T^{(m)}$.*

*2) If $M < N$, expert $m$ stabilizes within an expert set $\mathcal{M}_k$, and its expert model satisfies $\|\boldsymbol{w}_t^{(m)} - \boldsymbol{w}_{T_1+1}^{(m)}\|_\infty = \mathcal{O}(\sigma_0^{1.5})$ for any $t \in \{T_1 + 2, \cdots, T\}$.*

We first propose the following lemmas before formally proving Proposition 1. Then we prove Proposition 1 in Appendix E.8.

**Lemma 7.** *At any training round $t \in \{1, \cdots, T_1\}$, for any feature signal $\boldsymbol{v}_n$, the gating network parameter of expert $m \in [M]$ satisfies*

$$\langle \boldsymbol{\theta}_{t+1}^{(m)} - \boldsymbol{\theta}_t^{(m)}, \boldsymbol{v}_n \rangle = \begin{cases} -\mathcal{O}(\sigma_0), & \text{if } m = m_t, \\ \mathcal{O}(M^{-1}\sigma_0), & \text{if } m \neq m_t. \end{cases}$$

Lemma 7 tells that for any expert $m_t$ being selected by the router, its softmax value under the updated $\boldsymbol{\theta}_{t+1}^{m_t}$ is reduced since the next task $t$. While for any expert $m$ without being selected, its softmax value is increased. This is to ensure fair exploration of each expert $m$ under the auxiliary loss function in Eq. (7). In addition, for any expert $m$ without being selected, its gating network parameter $\boldsymbol{\theta}_t^{(m)}$ is updated at the same speed with others for any signal vector $\boldsymbol{v}_n$.

**Lemma 8.** *At the end of the exploration stage, with probability at least $1 - \delta$, the fraction of tasks dispatched to any expert $m \in [M]$ satisfies*

$$\left| f_{T_1}^{(m)} - \frac{1}{M} \right| = \mathcal{O}(\eta^{0.5} M^{-1}). \tag{32}$$

Lemma 8 tells that during the exploration stage, all the $M$ experts are expected to be evenly explored by all tasks. Therefore, the gating network parameter $\boldsymbol{\theta}_t^{(m)}$ of expert $m$ is updated similarly to all the others.

**Lemma 9.** *At the end of the exploration stage, i.e., $t = T_1$, the following property holds*

$$\left\| \boldsymbol{\theta}_{T_1}^{(m)} - \boldsymbol{\theta}_{T_1}^{(m')} \right\|_\infty = \mathcal{O}(\eta^{-0.5}\sigma_0),$$

*for any $m, m' \in [M]$ and $m \neq m'$.*

Define $\delta_{\boldsymbol{\Theta}} = |h_m(\mathbf{X}_t, \boldsymbol{\theta}_t^{(m)}) - h_{m'}(\mathbf{X}_t, \boldsymbol{\theta}_t^{(m)})|$. Then we obtain the following lemma.

**Lemma 10.** *At any round $t$, if $\delta_{\boldsymbol{\Theta}_t}$ is close to $0$, it satisfies $|\pi_m(\mathbf{X}_t, \boldsymbol{\Theta}_t) - \pi_{m'}(\mathbf{X}_t, \boldsymbol{\Theta}_t)| = \mathcal{O}(\delta_{\boldsymbol{\Theta}})$. Otherwise, $|\pi_m(\mathbf{X}_t, \boldsymbol{\Theta}_t) - \pi_{m'}(\mathbf{X}_t, \boldsymbol{\Theta}_t)| = \Omega(\delta_{\boldsymbol{\Theta}})$.*

**Lemma 11.** *If $M = \Omega(N \ln(N))$, we have $|\mathcal{M}_n| \geq 1$ for all $n \in [N]$ with probability at least $1 - o(1)$. If $M < N$, given $M = \Omega(K \ln(K))$, we have $|\mathcal{M}_k| \geq 1$ for all $k \in [K]$ with probability at least $1 - o(1)$.*

**Lemma 12.** *At any round $t$, we have the following properties:*

*1) for task arrival $n_t$ with ground truth $\boldsymbol{w}_{n_t} = \boldsymbol{w}_n$ under $M > N$, if it is routed to a correct expert $m_t \in \mathcal{M}_n$, then $\nabla_{\boldsymbol{\theta}_t^{(m)}} \mathcal{L}_t^{loc} = \mathbf{0}$ for any expert $m \in [M]$.*

*2) for task arrival $n_t$ with ground truth $\boldsymbol{w}_{n_t} \in \mathcal{W}_k$ under $M < N$, if it is routed to a correct expert $m_t \in \mathcal{M}_k$, then $\|\nabla_{\boldsymbol{\theta}_t^{(m)}} \mathcal{L}_t^{loc}\|_\infty = \mathcal{O}(\sigma_0^{1.5})$ for any expert $m \in [M]$.*

Let $\mathbf{X}_n$ and $\mathbf{X}_{n'}$ denote two feature matrices containing feature signals $\boldsymbol{v}_n$ and $\boldsymbol{v}_{n'}$, respectively.

**Lemma 13.** *If $n$ and $n'$ satisfy: 1) $n \neq n'$ under $M > N$ or 2) $\boldsymbol{w}_n \in \mathcal{W}_k$ and $\boldsymbol{w}_{n'} \in \mathcal{W}_{k'}$ with $k \neq k'$ under $M < N$, then if expert $m$ satisfies 1) $m \in \mathcal{M}_n$ under $M > N$ or 2) $m \in \mathcal{M}_k$ under $M < N$, at the beginning of the router learning stage $t = T_1 + 1$, then the following property holds at any round $t \in \{T_1 + 2, \cdots, T\}$:*

$$\pi_m(\mathbf{X}_n, \boldsymbol{\Theta}_t) > \pi_m(\mathbf{X}_{n'}, \boldsymbol{\Theta}_t), \forall m \in [M]. \tag{33}$$

Based on these lemmas, the proof of Proposition 1 is given in Appendix E.8.

### E.1 PROOF OF LEMMA 7

*Proof.* According to Lemma 5, for any round $t \in \{1, \cdots, T_1\}$, the auxiliary loss is the primary loss to update $\boldsymbol{\Theta}_t$ of the gating network. Then based on the update rule of $\boldsymbol{\theta}_t^{(m)}$ in Eq. (9) and the gradient

of $\nabla_{\boldsymbol{\theta}_t^{(m_t)}} \mathcal{L}_t^{(aux)}$ in Eq. (30), we obtain

$$
\begin{aligned}
\|\boldsymbol{\theta}_{t+1}^{(m_t)} - \boldsymbol{\theta}_t^{(m_t)}\|_\infty &= \|\eta \cdot \nabla_{\boldsymbol{\theta}_t^{(m_t)}} \mathcal{L}_t^{(aux)}\|_\infty \\
&= \mathcal{O}(\sigma_0^{0.5}\eta) \cdot \left\|\pi_{m_t}(\mathbf{X}_t, \boldsymbol{\Theta}_t) \cdot \sum_{m' \neq m_t} \pi_{m'}(\mathbf{X}_t, \boldsymbol{\Theta}_t) \cdot \sum_{i \in [s_t]} \mathbf{X}_{t,i}\right\|_\infty \\
&= \mathcal{O}(\sigma_0),
\end{aligned}
$$

based on the fact that $\pi_{m_t}(\mathbf{X}_t, \boldsymbol{\Theta}_t) \cdot \sum_{m' \neq m_t} \pi_{m'}(\mathbf{X}_t, \boldsymbol{\Theta}_t) = \pi_{m_t}(\mathbf{X}_t, \boldsymbol{\Theta}_t) \cdot (1 - \pi_{m_t}(\mathbf{X}_t, \boldsymbol{\Theta}_t)) \leq \frac{1}{4}$ and $\|X_t\|_\infty = \mathcal{O}(1)$.

While for any $m \neq m_t$, we calculate

$$
\begin{aligned}
\|\boldsymbol{\theta}_{t+1}^{(m)} - \boldsymbol{\theta}_t^{(m)}\|_\infty &= \|\eta \cdot \nabla_{\boldsymbol{\theta}_t^{(m)}} \mathcal{L}_t^{(aux)}\|_\infty \\
&= \mathcal{O}(\sigma_0^{0.5}\eta) \cdot \left\|\pi_{m_t}(\mathbf{X}_t, \boldsymbol{\Theta}_t) \cdot \pi_m(\mathbf{X}_t, \boldsymbol{\Theta}_t) \cdot \sum_{i \in [s_t]} \mathbf{X}_{t,i}\right\|_\infty \\
&= \mathcal{O}(M^{-1}\sigma_0),
\end{aligned}
$$

due to the fact that $\pi_{m_t}(\mathbf{X}_t, \boldsymbol{\Theta}_t) \cdot \pi_m(\mathbf{X}_t, \boldsymbol{\Theta}_t) = \mathcal{O}(M^{-1})$.

Note that by Eq. (30), we have $\nabla_{\boldsymbol{\theta}_t^{(m_t)}} \mathcal{L}_t^{(aux)} > 0$ and $\nabla_{\boldsymbol{\theta}_t^{(m)}} \mathcal{L}_t^{(aux)} < 0$ for any $m \neq m_t$. Consequently, for expert $m_t$, its corresponding output $h_{m_t}$ at the gating network will be reduced by $\mathcal{O}(\sigma_0)$ for the same feature signal $\boldsymbol{v}_n$ since task $t+1$. While for any expert $m \neq m_t$, its corresponding output $h_m$ is increased by $\mathcal{O}(M^{-1}\sigma_0)$. $\square$

### E.2 PROOF OF LEMMA 8

*Proof.* By the symmetric property, we have that for any $m \in [M]$, $\mathbb{E}[f_{T_1}^{(m)}] = \frac{1}{M}$.

By Hoeffding's inequality, we obtain

$$
\mathbb{P}(|f_{T_1}^{(m)} - \frac{1}{M}| \leq \epsilon) \geq 1 - 2\exp(-2\epsilon^2 T_1).
$$

Then we further obtain

$$
\begin{aligned}
\mathbb{P}(|f_{T_1}^{(m)} - \frac{1}{M}| \leq \epsilon, \forall m \in [M]) &\geq (1 - 2\exp(-2\epsilon^2 T_1))^M \\
&\geq 1 - 2M\exp(-2\epsilon^2 T_1)).
\end{aligned}
$$

Let $\delta = 1 - 2M\exp(-2\epsilon^2 T_1))$. Then we obtain $\epsilon = \mathcal{O}(\eta^{0.5} M^{-1})$. Subsequently, there is a probability of at least $1 - \delta$ that $\left|f_{T_1}^{(m)} - \frac{1}{M}\right| = \mathcal{O}(\eta^{0.5} M^{-1})$. $\square$

### E.3 PROOF OF LEMMA 9

*Proof.* Based on Lemma 7 and Lemma 8 and their corresponding proofs above, we can prove Lemma 9 below.

For experts $m$ and $m'$, they are selected by the router for $T_1 \cdot f_{T_1}^{(m)}$ and $T_1 \cdot f_{T_1}^{(m')}$ times during the exploration stage, respectively. Therefore, we obtain

$$
\begin{aligned}
\|\boldsymbol{\theta}_{T_1}^{(m)}\|_\infty &= f_{T_1}^{(m)} \cdot T_1 \cdot \mathcal{O}(\sigma_0) - (1 - f_{T_1}^{(m)}) \cdot T_1 \cdot \mathcal{O}(M^{-1}\sigma_0), \\
\|\boldsymbol{\theta}_{T_1}^{(m')}\|_\infty &= f_{T_1}^{(m')} \cdot T_1 \cdot \mathcal{O}(\sigma_0) - (1 - f_{T_1}^{(m')}) \cdot T_1 \cdot \mathcal{O}(M^{-1}\sigma_0).
\end{aligned}
$$

Then by Eq. (9) and Lemma 7, we calculate

$$
\begin{aligned}
\left\|\boldsymbol{\theta}_{T_1}^{(m)} - \boldsymbol{\theta}_{T_1}^{(m')}\right\|_\infty &= \left|(f_{T_1}^{(m)} - f_{T_1}^{(m')}) \cdot T_1 \cdot \mathcal{O}(\sigma_0) - ((1 - f_{T_1}^{(m)}) - (1 - f_{T_1}^{(m')})) \cdot T_1 \cdot \mathcal{O}(M^{-1}\sigma_0)\right| \\
&= |f_{T_1}^{(m)} - f_{T_1}^{(m')}| \cdot T_1 \cdot \mathcal{O}(\sigma_0) \\
&= \mathcal{O}(\eta^{-0.5}\sigma_0),
\end{aligned}
$$

where the first equality is derived based on the update steps in Lemma 7, and the last equality is because of $T_1 \cdot |f_{T_1}^{(m)} - f_{T_1}^{(m')}| = \mathcal{O}(\eta^{-0.5})$ by Eq. (32) in Lemma 8. $\square$

### E.4 PROOF OF LEMMA 10

*Proof.* At any round $t$, we calculate

$$|\pi_m(\mathbf{X}_t, \mathbf{\Theta}_t) - \pi_{m'}(\mathbf{X}_t, \mathbf{\Theta}_t)| = \left|\pi_{m'}(\mathbf{X}_t, \mathbf{\Theta}_t) \exp\left(h_m(\mathbf{X}_t, \boldsymbol{\theta}_t^{(m)}) - h_{m'}(\mathbf{X}_t, \boldsymbol{\theta}_t^{(m')})\right) - \pi_{m'}(\mathbf{X}_t, \mathbf{\Theta}_t)\right|$$

$$= \pi_{m'}(\mathbf{X}_t, \mathbf{\Theta}_t)\left|\exp\left(h_m(\mathbf{X}_t, \boldsymbol{\theta}_t^{(m)}) - h_{m'}(\mathbf{X}_t, \boldsymbol{\theta}_t^{(m')})\right) - 1\right|,$$

where the first equality is by solving Eq. (2). Then if $\delta_{\mathbf{\Theta}_t}$ is close to 0, by applying Taylor series with sufficiently small $\delta_{\mathbf{\Theta}}$, we obtain

$$|\pi_m(\mathbf{X}_t, \mathbf{\Theta}_t) - \pi_{m'}(\mathbf{X}_t, \mathbf{\Theta}_t)| \approx \pi_{m'}(\mathbf{X}_t, \mathbf{\Theta}_t)|h_m(\mathbf{X}_t, \boldsymbol{\theta}_t^{(m)}) - h_{m'}(\mathbf{X}_t, \boldsymbol{\theta}_t^{(m')})|$$

$$= \mathcal{O}(\delta_{\mathbf{\Theta}}),$$

where the last equality is because of $\pi_m(\tilde{\mathbf{X}}_t, \mathbf{\Theta}_t) \leq 1$.

While if $\delta_{\mathbf{\Theta}_t}$ is not sufficiently small, we obtain

$$|\pi_m(\mathbf{X}_t, \mathbf{\Theta}_t) - \pi_{m'}(\mathbf{X}_t, \mathbf{\Theta}_t)| > \pi_{m'}(\mathbf{X}_t, \mathbf{\Theta}_t)|h_m(\mathbf{X}_t, \boldsymbol{\theta}_t^{(m)}) - h_{m'}(\mathbf{X}_t, \boldsymbol{\theta}_t^{(m')})|$$

$$= \Omega(\delta_{\mathbf{\Theta}}).$$

This completes the proof. □

### E.5 PROOF OF LEMMA 11

*Proof.* If $M > N$, by the symmetric property, we have that for all $n \in [N], m \in [M]$,

$$\mathbb{P}(m \in \mathcal{M}_n) = \frac{1}{N}.$$

Therefore, the probability that $|\mathcal{M}_n|$ at least includes one expert is

$$\mathbb{P}(|\mathcal{M}_n| \geq 1) \geq 1 - \left(1 - \frac{1}{N}\right)^M.$$

By applying union bound, we obtain

$$\mathbb{P}(|\mathcal{M}_n| \geq 1, \forall n) \geq \left(1 - \left(1 - \frac{1}{N}\right)^M\right)^N \geq 1 - N\left(1 - \frac{1}{N}\right)^M \geq 1 - N\exp\left(-\frac{M}{N}\right) \geq 1 - \delta,$$

where the second inequality is because $(1 - N^{-1})^M$ is small enough, and the last inequality is because of $M = \Omega\left(N\ln\left(\frac{N}{\delta}\right)\right)$.

While if $M < N$, we can use the same method to prove that $\mathbb{P}(|\mathcal{M}_k| \geq 1, \forall k) \geq 1 - \delta$, given $M = \Omega\left(K\ln\left(\frac{K}{\delta}\right)\right)$ and $\mathbb{P}(m \in \mathcal{M}_k) = \frac{1}{K}$ by the symmetric property. □

### E.6 PROOF OF LEMMA 12

*Proof.* In the case of $M > N$, as $|\mathcal{M}_n| = 1$, if task $n_t$ with $n_t = n$ is routed to the correct expert $m_t \in \mathcal{M}_n$, we have $\boldsymbol{w}_t^{m_t} = \boldsymbol{w}_{t-1}^{m_t}$ by Eq. (5). Consequently, the caused locality loss $\mathcal{L}_t^{loc}(\mathbf{\Theta}_t, \mathcal{D}_t) = 0$, based on its definition in Eq. (6).

In the case of $M < N$, as $|\mathcal{M}_k| \geq 1$ for each cluster $k$, if task $n_t$ with $\boldsymbol{w}_{n_t} \in \mathcal{W}_k$ is routed to the correct expert $m_t \in \mathcal{M}_k$, we have

$$\|\boldsymbol{w}_t^{(m_t)} - \boldsymbol{w}_{t-1}^{(m_t)}\|_\infty = \|\mathbf{X}_t(\mathbf{X}_t^\top \mathbf{X}_t)^{-1}(\mathbf{y}_t - \mathbf{X}_t^\top \boldsymbol{w}_{t-1}^{(m_t)})\|_\infty$$

$$= \|\mathbf{X}_t(\mathbf{X}_t^\top \mathbf{X}_t)^{-1}\mathbf{X}_t^\top(\boldsymbol{w}_{n_t} - \boldsymbol{w}_{t-1}^{(m_t)})\|_\infty$$

$$= \mathcal{O}(\|\boldsymbol{w}_t - \boldsymbol{w}_{t-1}^{(m_t)}\|_\infty)$$

$$= \mathcal{O}(\sigma_0^{1.5}),$$

where the second equality is because of $\mathbf{y}_t = \mathbf{X}_t^\top \boldsymbol{w}_t$, and the third equality is because of $\|\mathbf{X}_t\|_\infty = \mathcal{O}(1)$. Therefore, we obtain $\nabla_{\boldsymbol{\theta}_t^{(m)}}\mathcal{L}_t^{loc} = \mathcal{O}(\sigma_0^{1.5})$ by solving Eq. (27), for any $m \in [M]$. □

### E.7 PROOF OF LEMMA 13

*Proof.* We first focus on the $M > N$ case to prove Lemma 13. Based on Eq. (35), we obtain $\pi_m(\mathbf{X}_n, \mathbf{\Theta}_{T_1}) - \pi_m(\mathbf{X}_{n'}, \mathbf{\Theta}_{T_1}) = \Omega(\sigma_0^{0.5})$ for $m \in \mathcal{M}_n$, given $\|\boldsymbol{\theta}_t^{(m)}\|_\infty = \mathcal{O}(\sigma_0^{0.5})$ derived in

Lemma 6. To prove Eq. (33), we will prove that $\pi_m(\mathbf{X}_n, \mathbf{\Theta}_t) - \pi_m(\mathbf{X}_{n'}, \mathbf{\Theta}_t) = \Omega(\sigma_0^{0.5})$ holds for any $t \geq T_1 + 1$.

Based on Lemma 5, we have $\|\nabla_{\boldsymbol{\theta}_t^{(m)}} \mathcal{L}_t^{task}\|_\infty = \mathcal{O}(\sigma_0)$, leading to $\langle \boldsymbol{\theta}_{t+1}^{(m_t)} - \boldsymbol{\theta}_t^{(m_t)}, \boldsymbol{v}_n \rangle = -\mathcal{O}(\sigma_0^{1.5})$ for expert $m_t$ and $\langle \boldsymbol{\theta}_{t+1}^{(m)} - \boldsymbol{\theta}_t^{(m)}, \boldsymbol{v}_n \rangle = \mathcal{O}(\sigma_0^{1.5})$ for any other expert $m \neq m_t$. As $T_2 = \lceil \sigma_0^{-0.5} \eta^{-1} M \rceil$, we calculate

$$\|\boldsymbol{\theta}_{T_2}^{(m)}\|_\infty < \mathcal{O}(\sigma_0^{0.5}) - \|\nabla_{\boldsymbol{\theta}_{T_1}^{(m)}} \mathcal{L}_{T_1}^{task}\|_\infty \cdot \eta \cdot (T_2 - T_1)$$
$$\leq \mathcal{O}(\sigma_0^{0.5}) - \mathcal{O}(\sigma_0) \cdot \mathcal{O}(\sigma_0^{-0.25})$$
$$= \mathcal{O}(\sigma_0^{0.5}).$$

Therefore, $\|\boldsymbol{\theta}_t^{(m)}\|_\infty = \mathcal{O}(\sigma_0^{0.5})$ is always true for $t \in \{T_1 + 1, \cdots, T_2\}$, and thus $\pi_m(\mathbf{X}_n, \mathbf{\Theta}_t) - \pi_m(\mathbf{X}_{n'}, \mathbf{\Theta}_t) = \Omega(\sigma_0^{0.5})$ holds, meaning $\pi_m(\mathbf{X}_n, \mathbf{\Theta}_t) > \pi_m(\mathbf{X}_{n'}, \mathbf{\Theta}_t), \forall m \in [M]$.

For the case of $M < N$, we can use the same method to prove Eq. (33). This completes the proof. $\square$

### E.8 FINAL PROOF OF PROPOSITION 1

*Proof.* **In the case of** $M > N$, to prove Proposition 1, we equivalently prove that at the end of the exploration stage with $t = T_1$, for any two experts $m \in \mathcal{M}_n$ and $m' \in \mathcal{M}_{n'}$ with $n \neq n'$, the following properties hold:

$$\pi_m(\mathbf{X}_n, \mathbf{\Theta}_t) > \pi_{m'}(\mathbf{X}_n, \mathbf{\Theta}_t), \quad \pi_{m'}(\mathbf{X}_{n'}, \mathbf{\Theta}_t) > \pi_m(\mathbf{X}_{n'}, \mathbf{\Theta}_t), \tag{34}$$

where $\mathbf{X}_n$ and $\mathbf{X}_{n'}$ contain feature signals $\boldsymbol{v}_n$ and $\boldsymbol{v}_{n'}$, respectively. Based on Eq. (34), we have each expert $m \in [M]$ stabilizes within an expert set $\mathcal{M}_n$.

According to Lemma 1, the gating network only focuses on feature signals. Then for each expert $m$, we calculate

$$|h_m(\mathbf{X}_n, \boldsymbol{\theta}_t^{(m)}) - h_m(\mathbf{X}_{n'}, \boldsymbol{\theta}_t^{(m)})| = |\langle \boldsymbol{\theta}_t^{(m)}, \boldsymbol{v}_n - \boldsymbol{v}_{n'} \rangle|$$
$$= \|\boldsymbol{\theta}_t^{(m)}\|_\infty \cdot \|\boldsymbol{v}_n - \boldsymbol{v}_{n'}\|_\infty$$
$$= \mathcal{O}(\sigma_0^{0.5}),$$

where the last equality is because of $\|\boldsymbol{\theta}_t^{(m)}\|_\infty = \mathcal{O}(\sigma_0^{0.5})$ in Lemma 6 and $\|\boldsymbol{v}_n - \boldsymbol{v}_{n'}\|_\infty = \mathcal{O}(1)$.

Then based on Lemma 10, we obtain

$$|\pi_m(\mathbf{X}_n, \mathbf{\Theta}_t) - \pi_m(\mathbf{X}_{n'}, \mathbf{\Theta}_t)| = \Omega(\sigma_0^{0.5}). \tag{35}$$

Next, we prove Eq. (34) by contradiction. Assume there exist two experts $m \in \mathcal{M}_n$ and $m' \in \mathcal{M}_{n'}$ such that

$$\pi_m(\mathbf{X}_n, \mathbf{\Theta}_t) > \pi_{m'}(\mathbf{X}_n, \mathbf{\Theta}_t), \quad \pi_m(\mathbf{X}_{n'}, \mathbf{\Theta}_t) > \pi_{m'}(\mathbf{X}_{n'}, \mathbf{\Theta}_t),$$

which is equivalent to

$$\pi_m(\mathbf{X}_n, \mathbf{\Theta}_t) > \pi_m(\mathbf{X}_{n'}, \mathbf{\Theta}_t) > \pi_{m'}(\mathbf{X}_{n'}, \mathbf{\Theta}_t) > \pi_{m'}(\mathbf{X}_n, \mathbf{\Theta}_t), \tag{36}$$

because of $\pi_m(\mathbf{X}_n, \mathbf{\Theta}_t) > \pi_m(\mathbf{X}_{n'}, \mathbf{\Theta}_t)$ and $\pi_{m'}(\mathbf{X}_{n'}, \mathbf{\Theta}_t) > \pi_{m'}(\mathbf{X}_n, \mathbf{\Theta}_t)$ based on the definition of expert set $\mathcal{M}_n$ in Eq. (11). Then we prove Eq. (36) does not exist at $t = T_1$.

For task $t = T_1$, we calculate

$$|h_m(\mathbf{X}_n, \boldsymbol{\theta}_{T_1}^{(m)}) - h_{m'}(\mathbf{X}_n, \boldsymbol{\theta}_{T_1}^{(m)})| \leq \|\boldsymbol{\theta}_{T_1}^{(m)} - \boldsymbol{\theta}_{T_1}^{(m')}\|_\infty \|\boldsymbol{v}_n\|_\infty$$
$$= \mathcal{O}(\sigma_0 \eta^{-0.5}), \tag{37}$$

where the first inequality is derived by union bound, and the second equality is because of $\|\boldsymbol{v}_n\|_\infty = \mathcal{O}(1)$ and $\|\boldsymbol{\theta}_{T_1}^{(m)} - \boldsymbol{\theta}_{T_1}^{(m')}\|_\infty = \mathcal{O}(\sigma_0 \eta^{-0.5})$ derived in Lemma 9 at the end of the exploration phase.

Then according to Lemma 10 and Eq. (37), we obtain

$$|\pi_m(\mathbf{X}_n, \mathbf{\Theta}_{T_1}) - \pi_{m'}(\mathbf{X}_n, \mathbf{\Theta}_{T_1})| = \mathcal{O}(\sigma_0 \eta^{-0.5}). \tag{38}$$

Based on Eq. (36), we further calculate

$$|\pi_m(\mathbf{X}_n, \mathbf{\Theta}_{T_1}) - \pi_{m'}(\mathbf{X}_n, \mathbf{\Theta}_{T_1})| \geq |\pi_m(\mathbf{X}_n, \mathbf{\Theta}_{T_1}) - \pi_m(\mathbf{X}_{n'}, \mathbf{\Theta}_{T_1})|$$
$$= \Omega(\sigma_0^{0.5}),$$

where the first inequality is derived by Eq. (36), and the last equality is derived in Eq. (35). This contradicts with Eq. (38) as $\sigma_0 \eta^{-0.5} < \sigma_0^{0.5}$ given $\eta = \mathcal{O}(\sigma_0^{0.5})$. Therefore, Eq. (36) does not exist for $t = T_1$, and Eq. (34) is true for $t = T_1$.

Based on Lemma 13, we obtain that each expert set $\mathcal{M}_n$ is stable during the router learning stage. Therefore, at any round $t \in \{T_1 + 1, \cdots, T\}$, task $n_t$ with ground truth $\boldsymbol{w}_{n_t}$ will be routed to one of its best experts in $\mathcal{M}_{n_t}$. Then based on Lemma 12, we have that $\nabla_{\boldsymbol{\theta}_t^{(m)}} \mathcal{L}_t^{loc} = 0$ holds in the router learning stage. Subsequently, $\boldsymbol{w}_t^{(m)}$ of any expert $m$ remains unchanged, based on Lemma 3.

**In the case of** $M < N$, we similarly obtain that at the end of the exploration phase with $t = T_1$, for any two experts $m \in \mathcal{M}_k$ and $m' \in \mathcal{M}_{k'}$ with $k \neq k'$, the following property holds
$$\pi_m(\mathbf{X}_k, \boldsymbol{\Theta}_t) > \pi_{m'}(\mathbf{X}_k, \boldsymbol{\Theta}_t), \quad \pi_{m'}(\mathbf{X}_{k'}, \boldsymbol{\Theta}_t) > \pi_m(\mathbf{X}_{k'}, \boldsymbol{\Theta}_t),$$
where $\mathbf{X}_k$ and $\mathbf{X}_{k'}$ contain feature signals $\boldsymbol{v}_k$ and $\boldsymbol{v}_{k'}$, respectively.

Then during the router learning stage with $t > T_1$, any task $n_t$ with $\boldsymbol{w}_{n_t} \in \mathcal{W}_k$ will be routed to the correct expert $m \in \mathcal{M}_k$. Let $\boldsymbol{w}^{(m)}$ denote the minimum $\ell^2$-norm offline solution for expert $m$. Based on the update rule of $\boldsymbol{w}_t^{(m_t)}$ in Eq. (5), we calculate
$$\boldsymbol{w}_t^{(m_t)} - \boldsymbol{w}^{(m_t)} = \boldsymbol{w}_{t-1}^{(m_t)} + \mathbf{X}_t(\mathbf{X}_t^\top \mathbf{X}_t)^{-1}(\mathbf{y}_t - \mathbf{X}_t^\top \boldsymbol{w}_{t-1}^{(m_t)}) - \boldsymbol{w}^{(m_t)}$$
$$= (\boldsymbol{I} - \mathbf{X}_t(\mathbf{X}_t^\top \mathbf{X}_t)^{-1}\mathbf{X}_t^\top)\boldsymbol{w}_{t-1}^{(m_t)} + \mathbf{X}_t(\mathbf{X}_t^\top \mathbf{X}_t)^{-1}\mathbf{X}_t^\top \boldsymbol{w}^{(m_t)} - \boldsymbol{w}^{(m_t)}$$
$$= (\boldsymbol{I} - \mathbf{X}_t(\mathbf{X}_t^\top \mathbf{X}_t)^{-1}\mathbf{X}_t^\top)(\boldsymbol{w}_{t-1}^{(m_t)} - \boldsymbol{w}^{(m_t)}),$$
where the second equality is because of $\mathbf{y}_t = \mathbf{X}_t^\top \boldsymbol{w}^{(m_t)}$. Define $\boldsymbol{P}_t = \mathbf{X}_t(\mathbf{X}_t^\top \mathbf{X}_t)^{-1}\mathbf{X}_t^\top$ for task $n_t$, which is the projection operator on the solution space $\boldsymbol{w}_{n_t}$. Then we obtain
$$\boldsymbol{w}_t^{(m)} - \boldsymbol{w}^{(m)} = (\boldsymbol{I} - \boldsymbol{P}_t) \cdots (\boldsymbol{I} - \boldsymbol{P}_{T_1+1})(\boldsymbol{w}_{T_1}^{(m)} - \boldsymbol{w}^{(m)})$$
for each expert $m \in [M]$.

Since orthogonal projections $\boldsymbol{P}_t$'s are non-expansive operators, it also follows that
$$\forall t \in \{T_1 + 1, \cdots, T\}, \|\boldsymbol{w}_t^{(m)} - \boldsymbol{w}^{(m)}\| \leq \|\boldsymbol{w}_{t-1}^{(m)} - \boldsymbol{w}^{(m)}\| \leq \cdots \leq \|\boldsymbol{w}_{T_1}^{(m)} - \boldsymbol{w}^{(m)}\|.$$
As the solution spaces $\mathcal{W}_k$ is fixed for each expert $m \in \mathcal{M}_k$, we further obtain
$$\|\boldsymbol{w}_t^{(m)} - \boldsymbol{w}_{T_1+1}^{(m)}\|_\infty = \|\boldsymbol{w}_t^{(m)} - \boldsymbol{w}^{(m)} + \boldsymbol{w}^{(m)} - \boldsymbol{w}_{T_1+1}^{(m)}\|_\infty$$
$$\leq \|\boldsymbol{w}_t^{(m)} - \boldsymbol{w}^{(m)}\|_\infty + \|\boldsymbol{w}_{T_1+1}^{(m)} - \boldsymbol{w}^{(m)}\|_\infty$$
$$\leq \max_{\boldsymbol{w}_n, \boldsymbol{w}_{n'} \in \mathcal{W}_k} \|\boldsymbol{w}_n - \boldsymbol{w}_{n'}\|_\infty$$
$$= \mathcal{O}(\sigma_0^{1.5}),$$
where the first inequality is derived by the union bound, the second inequality is because of the orthogonal projections for the update of $\boldsymbol{w}_t^{(m)}$ per task, and the last equality is because of $\|\boldsymbol{w}_n - \boldsymbol{w}_{n'}\|_\infty = \mathcal{O}(\sigma_0^{1.5})$ for any two ground truths in the same set $\mathcal{W}_k$. $\qquad \square$

## F FULL VERSION AND PROOF OF PROPOSITION 2

**Proposition 2** (Full version). *If the MoE keeps updating $\boldsymbol{\Theta}_t$ by Eq. (9) at any round $t \in [T]$, we obtain: 1) At round $t_1 = \lceil \eta^{-1} \sigma_0^{-0.25} M \rceil$, the following property holds*
$$\left| h_m(\mathbf{X}_{t_1}, \boldsymbol{\theta}_{t_1}^{(m)}) - h_{m'}(\mathbf{X}_{t_1}, \boldsymbol{\theta}_{t_1}^{(m')}) \right| = \begin{cases} \mathcal{O}(\sigma_0^{1.75}), & \text{if } m, m' \in \mathcal{M}_n \text{ under } M > N \\ & \text{or } m, m' \in \mathcal{M}_k \text{ under } M < N, \\ \Theta(\sigma_0^{0.75}), & \text{otherwise.} \end{cases}$$
*2) At round $t_2 = \lceil \eta^{-1} \sigma_0^{-0.75} M \rceil$, the following property holds*
$$\left| h_m(\mathbf{X}_{t_2}, \boldsymbol{\theta}_{t_2}^{(m)}) - h_{m'}(\mathbf{X}_{t_2}, \boldsymbol{\theta}_{t_2}^{(m')}) \right| = \mathcal{O}(\sigma_0^{1.75}), \forall m, m' \in [M].$$

We first propose the following lemmas as preliminaries to prove Proposition 2. Then we prove Proposition 2 in Appendix F.3.

For any two experts $m, m'$, define $\Delta_{\boldsymbol{\Theta}} = |\pi_m(\mathbf{X}_t, \boldsymbol{\Theta}_t) - \pi_{m'}(\mathbf{X}_t, \boldsymbol{\Theta}_t)|$.

**Lemma 14.** *At any round $t \in \{T_1 + 1, \cdots, t_1\}$, $\forall m \neq m_t$, the following property holds*
$$\left\| \nabla_{\boldsymbol{\theta}_t^{(m)}} \mathcal{L}_t^{task} - \nabla_{\boldsymbol{\theta}_t^{(m_t)}} \mathcal{L}_t^{task} \right\|_\infty = \mathcal{O}(\sigma_0). \tag{39}$$

Let $\mathbf{X}_n$ and $\mathbf{X}_{n'}$ denote two feature matrices containing feature signals $\boldsymbol{v}_n$ and $\boldsymbol{v}_{n'}$, respectively.

**Lemma 15.** *In the router learning stage with $t \in \{T_1 + 1, \cdots, t_1\}$, for any two experts satisfying 1) $m \in \mathcal{M}_n$ and $m' \in \mathcal{M}_{n'}$ with $n \neq n'$ under $M > N$, or 2) $m \in \mathcal{M}_k$ and $m' \in \mathcal{M}_{k'}$ with $\boldsymbol{w}_n \in \mathcal{W}_k, \boldsymbol{w}_{n'} \in \mathcal{W}_{k'}$ and $k \neq k'$ under $M < N$, the following properties hold:*

$$\pi_m(\mathbf{X}_n, \boldsymbol{\Theta}_t) > \pi_{m'}(\mathbf{X}_n, \boldsymbol{\Theta}_t), \ \ \pi_{m'}(\mathbf{X}_{n'}, \boldsymbol{\Theta}_t) > \pi_m(\mathbf{X}_{n'}, \boldsymbol{\Theta}_t). \tag{40}$$

## F.1 Proof of Lemma 14

*Proof.* Based on the proof of Lemma 5, for any $m \neq m_t$, we calculate

$$\big\|\nabla_{\boldsymbol{\theta}_t^{(m)}} \mathcal{L}_t^{task} - \nabla_{\boldsymbol{\theta}_t^{(m_t)}} \mathcal{L}_t^{task}\big\|_\infty = \big\|(\|\boldsymbol{w}_t^{(m_t)} - \boldsymbol{w}_{t-1}^{(m_t)}\|_2 + \frac{\alpha M}{t} f_t^{m_t})\big(\frac{\partial \pi_{m_t}}{\partial \boldsymbol{\theta}_t^{(m)}} - \frac{\partial \pi_{m_t}}{\partial \boldsymbol{\theta}_t^{(m_t)}}\big)\big\|_\infty$$

$$= \mathcal{O}\big(\sigma_0 + \frac{\alpha M}{t}\big) \cdot \big\|\frac{\partial \pi_{m_t}}{\partial \boldsymbol{\theta}_t^{(m)}} - \frac{\partial \pi_{m_t}}{\partial \boldsymbol{\theta}_t^{(m_t)}}\big\|_\infty$$

$$= \mathcal{O}(\sigma_0),$$

where the last equality is because of $\pi_{m_t(\mathbf{X}_t, \boldsymbol{\Theta}_t)} < 1$, $\|\mathbf{X}_t\|_\infty = \mathcal{O}(1)$ and $\frac{\alpha M}{t} \leq \sigma_0$ for any $t \in \{T_1 + 1, \cdots, t_1\}$. $\square$

## F.2 Proof of Lemma 15

*Proof.* We will use the same method as in Appendix E.8 to prove Lemma 15 by contradiction. Here, we only prove the case of $M > N$. The proof for the case of $M < N$ is similar.

Recall Proposition 1 that Eq. (40) is true at $t = T_1 + 1$. Based on Lemma 13, we have $|\pi_m(\mathbf{X}_n, \boldsymbol{\Theta}_t) - \pi_m(\mathbf{X}_{n'}, \boldsymbol{\Theta}_t)| = \Omega(\sigma_0^{0.5})$. Then in the following, we aim to prove $|\pi_m(\mathbf{X}_n, \boldsymbol{\Theta}_{T_1}) - \pi_{m'}(\mathbf{X}_n, \boldsymbol{\Theta}_{T_1})| = \mathcal{O}(\sigma_0 \eta^{-0.5})$ in Eq. (38) is always true for any $m \in \mathcal{M}_n$ and $m' \in \mathcal{M}_{n'}$ during the router learning stage. Then according to the proof of Proposition 1 in Appendix E.8, Eq. (40) is also true.

Under Eq. (40) at $t = T_1 + 1$, the router will route task $t$ to its best expert $m_t \in \mathcal{M}_{n_t}$, leading to $\nabla^*_{\boldsymbol{\theta}_t^{(m)}} \mathcal{L}_t^{loc} = 0$ by Lemma 12. Therefore, $\nabla^*_{\boldsymbol{\theta}_t^{(m)}} \mathcal{L}_t^{task} = \mathcal{O}(\sigma_0)$ makes $\langle \boldsymbol{\theta}_{t+1}^{(m_t)} - \boldsymbol{\theta}_t^{(m_t)}, \boldsymbol{v}_n \rangle = -\mathcal{O}(\sigma_0^{1.5})$ for expert $m_t$ and $\langle \boldsymbol{\theta}_{t+1}^{(m)} - \boldsymbol{\theta}_t^{(m)}, \boldsymbol{v}_n \rangle = \mathcal{O}(\sigma_0^{1.5})$ any other expert $m \neq m_t$ at task $t = T_1 + 1$.

Subsequently, for any two experts $m \in \mathcal{M}_n$ and $m' \in \mathcal{M}_{n'}$, we calculate

$$\|\boldsymbol{\theta}_{t_1}^{(m)} - \boldsymbol{\theta}_{t_1}^{(m')}\|_\infty \leq \|\boldsymbol{\theta}_{T_1+2}^{(m)} - \boldsymbol{\theta}_{T_1+2}^{(m')}\|_\infty + (t_1 - T_1) \cdot \mathcal{O}(\sigma_0^{1.5})$$

$$\leq \mathcal{O}(\sigma_0 \eta^{-0.5}) + \mathcal{O}(\eta^{-1} \sigma_0^{-0.25}) \cdot \mathcal{O}(\sigma_0^{1.5})$$

$$= \mathcal{O}(\sigma_0 \eta^{-0.5}),$$

where the first inequality is because of $\langle \boldsymbol{\theta}_{t+1}^{(m)} - \boldsymbol{\theta}_t^{(m)}, \boldsymbol{v}_n \rangle = \mathcal{O}(\sigma_0^{1.5})$, and the second inequality is because of $t_1 - T_1 \leq t_1$.

As $\|\boldsymbol{\theta}_{t_1}^{(m)} - \boldsymbol{\theta}_{t_1}^{(m')}\|_\infty = \mathcal{O}(\sigma_0 \eta^{-0.5})$ and $|\pi_m(\mathbf{X}_n, \boldsymbol{\Theta}_t) - \pi_m(\mathbf{X}_{n'}, \boldsymbol{\Theta}_t)| = \Omega(\sigma_0^{0.5})$, Eq. (40) is true for any $t \in \{T_1 + 2, \cdots, t_1\}$, based on our proof of Proposition 1. $\square$

## F.3 Final proof of Proposition 2

*Proof.* We first focus on the $M > N$ case to prove $|h_m(\mathbf{X}_{t_1}, \boldsymbol{\theta}_{t_1}^{(m)}) - h_{m'}(\mathbf{X}_{t_1}, \boldsymbol{\theta}_{t_1}^{(m')})| = \mathcal{O}(\sigma_0^{1.75})$ for any two different experts $m, m' \in \mathcal{M}_n$ in the same expert set in Eq. (12). Then we prove $|h_m(\mathbf{X}_{t_1}, \boldsymbol{\theta}_{t_1}^{(m)}) - h_{m'}(\mathbf{X}_{t_1}, \boldsymbol{\theta}_{t_1}^{(m')})| = \Theta(\sigma_0^{0.75})$ for any two experts $m \in \mathcal{M}_n$ and $m \in \mathcal{M}_{n'}$ in different expert sets in Eq. (12). After that, we prove Eq. (13) at round $t_2 = \lceil \eta^{-1} \sigma_0^{-0.75} M \rceil$. Finally, we prove that the above analysis can be generalized to the case of $M < N$.

In the case of $M > N$, let $M'$ and $m'$ denote the two experts within set $\mathcal{M}_n$ with the maximum and minimum softmax values, respectively. In other words, we have

$$M' = \arg \max_{m \in \mathcal{M}_n} \{\pi_m(\mathbf{X}_t, \boldsymbol{\Theta}_t)\}, \ m' = \arg \min_{m \in \mathcal{M}_n} \{\pi_m(\mathbf{X}_t, \boldsymbol{\Theta}_t)\}, \tag{41}$$

where $\boldsymbol{v}_n \in \mathbf{X}_t$. If the two experts satisfies $|h_{M'}(\mathbf{X}_{t_1}, \boldsymbol{\theta}_{t_1}^{(M')}) - h_{m'}(\mathbf{X}_{t_1}, \boldsymbol{\theta}_{t_1}^{(m')})| = \mathcal{O}(\sigma_0^{1.75})$, then this equation holds for any two experts in $\mathcal{M}_t$.

At the beginning of the router learning stage, we have

$$|\pi_{M'}(\mathbf{X}_{T_1}, \boldsymbol{\Theta}_{T_1}) - \pi_{m'}(\mathbf{X}_{T_1}, \boldsymbol{\Theta}_{T_1})| = \mathcal{O}(\sigma_0 \eta^{-0.5}) = \mathcal{O}(\sigma_0^{0.75}),$$

based on Proposition 1.

According to the routing strategy in Eq. (1), if the new task $t$ has ground truth $\boldsymbol{w}_n$, it is always routed to expert $M'$ until its softmax value is reduced to smaller than others. Therefore, we calculate the reduced output of gating network for expert $M'$ in the router learning stage:

$$\langle \boldsymbol{\theta}_{T_1+1}^{(M')} - \boldsymbol{\theta}_{t_1}^{(M')}, \boldsymbol{v}_n \rangle \leq \mathcal{O}(\sigma_0) \cdot \eta \cdot (t_1 - T_1)$$
$$= \mathcal{O}(\sigma_0^{0.75}).$$

While for expert $m'$, it will not be routed until its softmax value increased to the maximum. Therefore, we similarly calculate the increased gating output $\langle \boldsymbol{\theta}_{t_1}^{(m')} - \boldsymbol{\theta}_{T_1+1}^{(m')}, \boldsymbol{v}_n \rangle = \mathcal{O}(\sigma_0^{0.75})$ for expert $m'$.

Based on Lemma 4 and Lemma 14, the gating network parameters of experts $m'$ and $M'$ will converge to the same value, with an error smaller than the update step of $\boldsymbol{\Theta}_t$. Therefore, we obtain

$$\|\boldsymbol{\theta}_{t_1}^{(M')} - \boldsymbol{\theta}_{t_1}^{(m')}\|_\infty = \|\nabla_{\boldsymbol{\theta}_{t_1-1}^{(m_{t_1}-1)}} \mathcal{L}_{t_1-1}^{task} \cdot \eta\|_\infty$$
$$= \|\nabla_{\boldsymbol{\theta}_{t_1-1}^{(m_{t_1}-1)}} \mathcal{L}_{t_1-1}^{aux} \cdot \eta\|_\infty$$
$$= \mathcal{O}(\sigma_0^{1.75}),$$

based on the fact that $\nabla_{\boldsymbol{\theta}_{t_1-1}^{(m_{t_1}-1)}} \mathcal{L}_{t_1-1}^{aux} = \mathcal{O}(\sigma_0^{1.25})$ and $\eta = \mathcal{O}(\sigma_0^{0.5})$. Then according to Lemma 10, we obtain $|h_{M'}(\mathbf{X}_{t_1}, \boldsymbol{\theta}_{t_1}^{(M')}) - h_{m'}(\mathbf{X}_{t_1}, \boldsymbol{\theta}_{t_1}^{(m')})| = \mathcal{O}(\sigma_0^{1.75})$, which also holds for any two experts in the same expert set $\mathcal{M}_n$.

Next, we prove $|h_m(\mathbf{X}_{t_1}, \boldsymbol{\theta}_{t_1}^{(m)}) - h_{m'}(\mathbf{X}_{t_1}, \boldsymbol{\theta}_{t_1}^{(m')})| = \Theta(\sigma_0^{0.75})$ in Eq. (12) for expert $m'$ in Eq. (41) and another expert $m \notin \mathcal{M}_n$.

Let $\bar{m} \in \mathcal{M}_{n'}$ denote the index of the expert in other expert sets with the maximum softmax value of dataset $\mathbf{X}_n$, where $\boldsymbol{v}_n \in \mathbf{X}_t$. In other words, $\bar{m} = \arg\max_{m \notin \mathcal{M}_n} \{\pi_m(\mathbf{X}_t, \boldsymbol{\Theta}_t)\}$. According to the proof of Proposition 1, we obtain $\pi_{m'}(\mathbf{X}_{T_1}, \boldsymbol{\Theta}_{T_1}) - \pi_{\bar{m}}(\mathbf{X}_{T_1}, \boldsymbol{\Theta}_{T_1}) = \mathcal{O}(\sigma_0^{0.75})$. This equation indicates that during the router learning stage with $t \in \{T_1 + 1, \cdots, t_1\}$, any task arrival $n_t$ with ground truth $\boldsymbol{w}_{n_t} = \boldsymbol{w}_n$ will not be routed to expert $\bar{m}$. Therefore, $\pi_{\bar{m}}(\mathbf{X}_t, \boldsymbol{\Theta}_t)$ keeps increasing with $t$. Then we calculate the difference between the parameter gradient of expert $\bar{m}$ and expert $m'$ per round $t$:

$$\nabla_{\boldsymbol{\theta}_t^{(\bar{m})}} \mathcal{L}_t^{task} - \nabla_{\boldsymbol{\theta}_t^{(m')}} \mathcal{L}_t^{task} = \frac{\alpha M}{t} f_t^{m_t} \pi_{m_t}(\mathbf{X}_t, \boldsymbol{\Theta}_t)(\pi_{m'}(\mathbf{X}_t, \boldsymbol{\Theta}_t) - \pi_{\bar{m}}(\mathbf{X}_t, \boldsymbol{\Theta}_t)) \cdot \sum_{i \in [s_t]} \mathbf{X}_{t,i}$$
$$\geq 0,$$

based on the fact that $\pi_{m'}(\mathbf{X}_t, \boldsymbol{\Theta}_t) - \pi_{\bar{m}}(\mathbf{X}_t, \boldsymbol{\Theta}_t) > 0$ under Lemma 13. As $\nabla_{\boldsymbol{\theta}_t^{(\bar{m})}} \mathcal{L}_t^{task} < 0$ and $\nabla_{\boldsymbol{\theta}_t^{(m')}} \mathcal{L}_t^{task} < 0$, we obtain that $h_{m'}(\mathbf{X}_t, \boldsymbol{\theta}_t^{(m')}) - h_{\bar{m}}(\mathbf{X}_t, \boldsymbol{\theta}_t^{(\bar{m})})$ increases with $t$ during the router learning stage. According to our former analysis of $h_{m'}(\mathbf{X}_t, \boldsymbol{\theta}_t^{(m')})$, we obtain

$$|h_{m'}(\mathbf{X}_{t_1}, \boldsymbol{\theta}_{t_1}^{(m')}) - h_{\bar{m}}(\mathbf{X}_{t_1}, \boldsymbol{\theta}_{t_1}^{(\bar{m})})|$$
$$= |h_{m'}(\mathbf{X}_{T_1}, \boldsymbol{\theta}_{T_1}^{(m')}) - h_{\bar{m}}(\mathbf{X}_{T_1}, \boldsymbol{\theta}_{T_1}^{(\bar{m})})| + \|\nabla_{\boldsymbol{\theta}_t^{(\bar{m})}} \mathcal{L}_t^{task} - \nabla_{\boldsymbol{\theta}_t^{(m')}} \mathcal{L}_t^{task}\|_\infty \cdot \eta \cdot (t_1 - T_1)$$
$$= \mathcal{O}(\sigma_0^{0.75}) + \Theta(\sigma_0^{0.75}) = \Theta(\sigma_0^{0.75}),$$

where the first equality is because of $\|\nabla_{\boldsymbol{\theta}_t^{(\bar{m})}} \mathcal{L}_t^{task} - \nabla_{\boldsymbol{\theta}_t^{(m')}} \mathcal{L}_t^{task}\|_\infty = \mathcal{O}(\sigma_0)$. This completes the proof of Eq. (12).

Subsequently, we prove $|h_m(\mathbf{X}_{t_2}, \boldsymbol{\theta}_{t_2}^{(m)}) - h_{m'}(\mathbf{X}_{t_2}, \boldsymbol{\theta}_{t_2}^{(m')})| = \mathcal{O}(\sigma_0^{1.75}), \forall m, m' \in [M]$ in Eq. (13) by proving $|h_{M'}(\mathbf{X}_{t_2}, \boldsymbol{\theta}_{t_2}^{(M')}) - h_m(\mathbf{X}_{t_2}, \boldsymbol{\theta}_{t_2}^{(m)})| = \mathcal{O}(\sigma_0^{1.75})$ between expert $M' \in \mathcal{M}_n$ in Eq. (41) and any other expert $m \notin \mathcal{M}_n$.

Based on Lemma 7, we obtain

$$\langle \boldsymbol{\theta}_{t+1}^{(m)} - \boldsymbol{\theta}_t^{(m)}, \boldsymbol{v}_n \rangle = \begin{cases} -\mathcal{O}(\sigma_0^{1.25}), & \text{if } m = m_t, \\ \mathcal{O}(M^{-1}\sigma_0^{1.25}), & \text{if } m \neq m_t. \end{cases}$$

According to our analysis above, expert $M' \in \mathcal{M}_n$ is periodically selected by the router for training task arrivals $n_t = n$ during $t \in \{t_1, \cdots, t_2\}$. After each training of task $n$ at expert $M'$, its gate output $h_{M'}(\mathbf{X}_t, \boldsymbol{\theta}_t^{(M')})$ is reduced by $\mathcal{O}(\sigma_0^{1.25})$. While at other rounds without being selected, its gate output is increased by $\mathcal{O}(M^{-1}\sigma_0^{1.25})$. Under such training behavior, we obtain

$$\left| h_{M'}(\mathbf{X}_{t_1}, \boldsymbol{\theta}_{t_1}^{(M')}) - h_{M'}(\mathbf{X}_{t_2}, \boldsymbol{\theta}_{t_2}^{(M')}) \right| = \mathcal{O}(\sigma_0^{1.75}),$$

by assuming $\boldsymbol{v}_n \in \mathbf{X}_{t_1}$ and $\boldsymbol{v}_n \in \mathbf{X}_{t_2}$.

While for expert $m \notin \mathcal{M}_n$, its gate output $h_m(\mathbf{X}_t, \boldsymbol{\theta}_t^{(m)})$ keeps increasing for any data $\mathbf{X}_t$ of task $n$. For any $t \in \{t_1, \cdots, t_2\}$, we have $\|\nabla_{\boldsymbol{\theta}_t^{(m)}} \mathcal{L}_t^{aux}\|_\infty = \mathcal{O}(\sigma_0^{1.25})$ for expert $m$. Assuming that expert $m$ is never selected by the router for training task $n_t = n$ in the period, we obtain

$$|h_m(\mathbf{X}_{t_2}, \boldsymbol{\theta}_{t_2}^{(m)}) - h_m(\mathbf{X}_{t_1}, \boldsymbol{\theta}_{t_1}^{(m)})| = \|\nabla_{\boldsymbol{\theta}_t^{(m)}} \mathcal{L}_t^{aux}\|_\infty \cdot (t_2 - t_1) \cdot \eta$$

$$> |h_{M'}(\mathbf{X}_{t_1}, \boldsymbol{\theta}_{t_1}^{(M')}) - h_m(\mathbf{X}_{t_1}, \boldsymbol{\theta}_{t_1}^{(m)})|$$

where the inequality is because of $\|\nabla_{\boldsymbol{\theta}_t^{(m)}} \mathcal{L}_t^{aux}\|_\infty \cdot (t_2 - t_1) \cdot \eta = \mathcal{O}(\sigma_0^{0.5})$ and $|h_{M'}(\mathbf{X}_{t_1}, \boldsymbol{\theta}_{t_1}^{(M')}) - h_m(\mathbf{X}_{t_1}, \boldsymbol{\theta}_{t_1}^{(m)})| = \Theta(\sigma_0^{0.75})$. This inequality indicates that there exists a training round $t' \in \{t_1, \cdots, t_2\}$ such that $h_m(\mathbf{X}_{t'}, \boldsymbol{\theta}_{t'}^{(m)}) > h_{M'}(\mathbf{X}_{t'}, \boldsymbol{\theta}_{t'}^{(M')})$ for task arrival $n_{t'} = n$. Consequently, expert $m'$ is selected to train task $n$ again, meaning that $m' \in \mathcal{M}_n$ at round $t'$. Then the gating network parameters of experts $m$ and $M'$ will converge to the same value, with an error of $\mathcal{O}(\sigma_0^{1.75})$, based on Lemma 4 and Lemma 14. This completes the proof of Eq. (13) in the case of $M > N$.

In the case of $M < N$, let $M'$ and $m'$ denote the experts of set $\mathcal{M}_k$ with the maximum and minimum softmax values, respectively. In other words, we have

$$M' = \arg\max_{m \in \mathcal{M}_k} \{\pi_m(\mathbf{X}_t, \boldsymbol{\Theta}_t)\}, \ m' = \arg\min_{m \in \mathcal{M}_k} \{\pi_m(\mathbf{X}_t, \boldsymbol{\Theta}_t)\},$$

where the ground truth of task $t$ satisfies $\boldsymbol{w}_n \in \mathcal{W}_k$.

According to the proof of Proposition 2 in Appendix F.3, the gating network parameters of experts $m'$ and $M'$ will converge to the same value at the end of the router learning stage, with an error smaller than the update step of $\boldsymbol{\Theta}_t$. Therefore, we obtain

$$\|\boldsymbol{\theta}_{t_1}^{(M')} - \boldsymbol{\theta}_{t_1}^{(m')}\|_\infty = \|\nabla_{\boldsymbol{\theta}_{t_1-1}^{(m_{t_1-1})}} \mathcal{L}_{t_1-1}^{task} \cdot \eta\|_\infty$$

$$= \|(\nabla_{\boldsymbol{\theta}_{t_1-1}^{(m_{t_1-1})}} \mathcal{L}_{t_1-1}^{loc} + \nabla_{\boldsymbol{\theta}_{t_1-1}^{(m_{t_1-1})}} \mathcal{L}_{t_1-1}^{aux}) \cdot \eta\|_\infty$$

$$= \mathcal{O}(\sigma_0^{1.75}),$$

based on the fact that $\nabla_{\boldsymbol{\theta}_{t_1-1}^{(m_{t_1-1})}} \mathcal{L}_{t_1-1}^{aux} = \mathcal{O}(\sigma_0^{1.25})$ and $\nabla_{\boldsymbol{\theta}_{t_1-1}^{(m_{t_1-1})}} \mathcal{L}_{t_1-1}^{loc} = \mathcal{O}(\sigma_0^{1.5})$ derived in Lemma 12. Then we obtain $|h_{M'}(\mathbf{X}_{t_1}, \boldsymbol{\theta}_{t_1}^{(M')}) - h_{m'}(\mathbf{X}_{t_1}, \boldsymbol{\theta}_{t_1}^{(m')})| = \mathcal{O}(\sigma_0^{1.75})$, which also holds for any two experts in the same expert set $\mathcal{M}_k$.

Similarly, for any two experts in different expert sets, we can derive $\left| h_m(\mathbf{X}_{t_1}, \boldsymbol{\theta}_{t_1}^{(m)}) - h_{m'}(\mathbf{X}_{t_1}, \boldsymbol{\theta}_{t-1}^{(m')}) \right| = \Theta(\sigma_0^{0.75})$. For training round $t_2$, the proof for the case of $M < N$ is the same as the case of $M > N$ above, and thus we skip it here. $\qquad\square$

## G  FULL VERSION AND PROOF OF PROPOSITION 3

**Proposition 3** (Full version). *Under Algorithm 1, the MoE terminates updating $\boldsymbol{\Theta}_t$ since round $T_2 = \mathcal{O}(\eta^{-1}\sigma_0^{-0.25}M)$. Then for any task arrival $n_t$ at $t > T_2$, the following property holds:*

*1) If $M > N$, the router selects any expert $m \in \mathcal{M}_{n_t}$ with an identical probability of $\frac{1}{|\mathcal{M}_{n_t}|}$, where $|\mathcal{M}_{n_t}|$ is the number of experts in set $\mathcal{M}_n$.*

*2) If $M < N$ and $\boldsymbol{w}_{n_t} \in \mathcal{W}_k$, the router selects any expert $m \in \mathcal{M}_k$, with an identical probability of $\frac{1}{|\mathcal{M}_k|}$, where $|\mathcal{M}_k|$ is the number of experts in set $\mathcal{M}_k$.*

*Proof.* In the case of $M > N$, according to Algorithm 1 and Proposition 2, after the termination of gating network update, the following properties hold: 1) $|h_m(\mathbf{X}_t, \boldsymbol{\theta}_t^{(m)}) - h_{m'}(\mathbf{X}_t, \boldsymbol{\theta}_t^{(m')})| = \Theta(\sigma_0^{0.75})$ for any $m \in \mathcal{M}_n$ and $m' \in \mathcal{M}_{n'}$ and 2) $|h_m(\mathbf{X}_t, \boldsymbol{\theta}_t^{(m)}) - h_{m'}(\mathbf{X}_t, \boldsymbol{\theta}_t^{(m')})| = \mathcal{O}(\Gamma)$ for any $m, m' \in \mathcal{M}_n$, where $\Gamma = \mathcal{O}(\sigma_0^{1.25})$.

If the ground truth of task arrival $n_t$ satisfies $\boldsymbol{w}_{n_t} = \boldsymbol{w}_n$, for any experts $m \in \mathcal{M}_n$ and $m' \notin \mathcal{M}_n$, we have

$$h_m(\mathbf{X}_t, \boldsymbol{\theta}_t^{(m)}) + r_t^{(m)} - (h_{m'}(\mathbf{X}_t, \boldsymbol{\theta}_t^{(m')}) + r_t^{(m')}) \geq h_m(\mathbf{X}_t, \boldsymbol{\theta}_t^{(m)}) - h_{m'}(\mathbf{X}_t, \boldsymbol{\theta}_t^{(m')}) + r_t^{(m')}$$
$$= \Theta(\sigma_0^{0.75}),$$

given $r_t^{(m')} = \Theta(\sigma_0^{1.25})$ and $h_m(\mathbf{X}_t, \boldsymbol{\theta}_t^{(m)}) - h_{m'}(\mathbf{X}_t, \boldsymbol{\theta}_t^{(m')}) = \Theta(\sigma_0^{0.75})$. Therefore, any expert $m' \notin \mathcal{M}_n$ will not be selected to learn task $t$, and only experts in set $\mathcal{M}_n$ will be selected.

For any experts $m \in \mathcal{M}_n$, we calculate

$$\mathbb{P}\Big(m_t = m | m \in \mathcal{M}_n\Big) = \mathbb{P}\Big(m = \arg\max_{m' \in \mathcal{M}_n} \Big\{h_{m'}(\mathbf{X}_t, \boldsymbol{\theta}_t^{(m')}) + r_t^{(m')}\Big\}\Big)$$
$$= \mathbb{P}\Big(h_m(\mathbf{X}_t, \boldsymbol{\theta}_t^{(m)}) + r_t^{(m)} - (h_{m'}(\mathbf{X}_t, \boldsymbol{\theta}_t^{(m')}) + r_t^{(m')}) > 0, \forall m' \in \mathcal{M}_n\Big)$$
$$= \mathbb{P}\Big(r_t^{(m)} > r_t^{(m')}, \forall m' \in \mathcal{M}_n\Big) = \frac{1}{|\mathcal{M}_n|},$$

where the third equality is because of $r_t^{(m)} = \Theta(\sigma_0^{1.25})$ and $|h_m(\mathbf{X}_t, \boldsymbol{\theta}_t^{(m)}) - h_{m'}(\mathbf{X}_t, \boldsymbol{\theta}_t^{(m')})| = \mathcal{O}(\sigma_0^{1.25})$ derived under Algorithm 1, and the last equality is due to the fact that $r_t^{(m)}$ satisfies a uniform distribution $\mathrm{Unif}[0, \lambda]$.

In the case of $M < N$, we similarly derive the following properties: 1) $|h_m(\mathbf{X}_t, \boldsymbol{\theta}_t^{(m)}) - h_{m'}(\mathbf{X}_t, \boldsymbol{\theta}_t^{(m')})| = \Theta(\sigma_0^{0.75})$ for any $m \in \mathcal{M}_k$ and $m' \in \mathcal{M}_{k'}$ and 2) $|h_m(\mathbf{X}_t, \boldsymbol{\theta}_t^{(m)}) - h_{m'}(\mathbf{X}_t, \boldsymbol{\theta}_t^{(m')})| = \mathcal{O}(\Gamma)$ for any $m, m' \in \mathcal{M}_k$. Based on the two properties, $h_m(\mathbf{X}_t, \boldsymbol{\theta}_t^{(m)}) + r_t^{(m)} - (h_{m'}(\mathbf{X}_t, \boldsymbol{\theta}_t^{(m')}) + r_t^{(m')}) = \Theta(\sigma_0^{0.75})$ is always true for any two experts $m$ and $m'$ not in the same expert set $\mathcal{M}_k$. Furthermore, we can similarly calculate

$$\mathbb{P}\Big(m_t = m | m \in \mathcal{M}_k\Big) = \mathbb{P}\Big(r_t^{(m)} > r_t^{(m')}, \forall m' \in \mathcal{M}_k\Big) = \frac{1}{|\mathcal{M}_k|}.$$

This completes the proof of Proposition 3. $\qquad\square$

## H    PROOF OF PROPOSITION 4

*Proof.* In the single-expert system, based on the definition of forgetting in Eq. (15) and Eq. (42) in Lemma 16, for any training rounds $t \in [T]$ and $i \in \{1, \cdots, t\}$, we calculate:

$$\mathbb{E}[F_t] = \frac{1}{t-1} \sum_{i=1}^{t-1} \mathbb{E}\Big[\|\boldsymbol{w}_t^{(m_i)} - \boldsymbol{w}_{n_i}\|^2 - \|\boldsymbol{w}_{n_i}^{(m_i)} - \boldsymbol{w}_{n_i}\|^2\Big]$$

$$= \frac{1}{t-1} \sum_{i=1}^{t-1} \Big\{(r^t - r^i)\mathbb{E}[\|\boldsymbol{w}_{n_i}\|^2] + \sum_{l=1}^{t}(1-r)r^{t-l}\mathbb{E}[\|\boldsymbol{w}_{n_l} - \boldsymbol{w}_{n_i}\|^2]$$

$$- \sum_{j=1}^{i}(1-r)r^{i-j}\mathbb{E}[\|\boldsymbol{w}_{n_j} - \boldsymbol{w}_{n_i}\|^2]\Big\}$$

$$= \frac{1}{t-1} \sum_{i=1}^{t-1} \Big\{\frac{r^T - r^i}{N} \sum_{n=1}^{N}\|\boldsymbol{w}_n\|^2 + \frac{r^i - r^t}{N^2} \sum_{n \neq n'}^{N}\|\boldsymbol{w}_{n'} - \boldsymbol{w}_n\|^2\Big\},$$

where we let $\boldsymbol{w}_{n_i}$ denote the ground truth of the task arrival at round $i$. Here the last equality is derived by $\mathbb{E}[\|\boldsymbol{w}_{n_i}\|^2] = \frac{1}{N} \sum_{n=1}^{N} \|\boldsymbol{w}_n\|^2$ and

$$\mathbb{E}[\|\boldsymbol{w}_{n_j} - \boldsymbol{w}_{n_i}\|^2] = \mathbb{E}[\frac{1}{N} \sum_{n=1}^{N} \|\boldsymbol{w}_{n_j} - \boldsymbol{w}_n\|^2] = \frac{1}{N^2} \sum_{n \neq n'} \|\boldsymbol{w}_{n'} - \boldsymbol{w}_n\|^2$$

when there is only a single expert.

Similarly, we can calculate the generalization error

$$\begin{aligned}
\mathbb{E}[G_T] &= \tfrac{1}{T} \sum_{i=1}^{T} \mathbb{E}[\|\boldsymbol{w}_T^{(m_i)} - \boldsymbol{w}_{n_i}\|^2] \\
&= \tfrac{1}{T} \sum_{i=1}^{T} \left( r^T \mathbb{E}[\|\boldsymbol{w}_{n_i}\|^2] + \sum_{l=1}^{T} (1-r)r^{T-l} \mathbb{E}[\|\boldsymbol{w}_{n_l} - \boldsymbol{w}_{n_i}\|^2] \right) \\
&= \tfrac{r^T}{N} \sum_{n=1}^{N} \|\boldsymbol{w}_n\|^2 + \tfrac{1-r^T}{N^2} \sum_{n \neq n'} \|\boldsymbol{w}_n - \boldsymbol{w}_n'\|^2,
\end{aligned}$$

where the second equality is because of Eq. (42) in Lemma 16. This completes the proof of Proposition 4. □

# I  PROOF OF THEOREM 1

Before proving Theorem 1, we first propose the following lemma. Then we formally prove Theorem 1 in Appendix I.2.

For expert $m$, let $\tau^{(m)}(l) \in \{1, \cdots, T_1\}$ represent the training round of the $l$-th time that the router selects expert $m$ during the exploration stage. For instance, $\tau^{(1)}(2) = 5$ indicates that round $t = 5$ is the second time the router selects expert 1.

**Lemma 16.** *At any round $t \in \{T_1 + 1, \cdots, T\}$, for $i \in \{T_1 + 1, \cdots, t\}$, we have*
$$\|\boldsymbol{w}_t^{(m_i)} - \boldsymbol{w}_{n_i}\|^2 = \|\boldsymbol{w}_{T_1}^{(m_i)} - \boldsymbol{w}_{n_i}\|^2.$$
*While at any round $t \in \{1, \cdots, T_1\}$, for any $i \in \{1, \cdots, t\}$, we have*

$$\mathbb{E}[\|\boldsymbol{w}_t^{(m_i)} - \boldsymbol{w}_{n_i}\|^2] = r^{L_t^{(m_i)}} \mathbb{E}[\|\boldsymbol{w}_{n_i}\|^2] + \sum_{l=1}^{L_t^{(m_i)}} (1-r)r^{L_t^{(m_i)}-l} \mathbb{E}[\|\boldsymbol{w}_{\tau^{(m_i)}(l)} - \boldsymbol{w}_{n_i}\|^2], \quad (42)$$

*where $L_t^{(m_i)} = t \cdot f_t^{(m_i)}$ and $r = 1 - \frac{s}{d}$.*

## I.1  PROOF OF LEMMA 16

*Proof.* At any round $t \in \{T_1 + 1, \cdots, T\}$, we have $\boldsymbol{w}_t^{(m)} = \boldsymbol{w}_{T_1}^{(m)}$, based on Proposition 1 and Proposition 2. Therefore, $\|\boldsymbol{w}_t^{(m_i)} - \boldsymbol{w}_{n_i}\|^2 = \|\boldsymbol{w}_{T_1}^{(m_i)} - \boldsymbol{w}_{n_i}\|^2$ is true for any round $t \in \{T_1 + 1, \cdots, T\}$ and $i \in \{T_1 + 1, \ldots, t\}$.

Next, we prove Eq. (42) for round $t \in \{1, \cdots, T_1\}$. Define $\mathbf{P}_t = \mathbf{X}_t(\mathbf{X}_t^\top \mathbf{X}_t)^{-1}\mathbf{X}_t^\top$ for task $t$. At current task $t$, there are totally $L_t^{(m)} = t \cdot f_t^{(m)}$ tasks routed to expert $m$, where $f_t^{(m)}$ is in Eq. (7).

Based on the update rule of $\boldsymbol{w}_t^{(m)}$ in Eq. (5), we calculate

$$\begin{aligned}
\|\boldsymbol{w}_t^{(m_i)} - \boldsymbol{w}_{n_i}\|^2 &= \|\boldsymbol{w}_{\tau^{(m_i)}(L_t^{(m_i)})}^{(m_i)} - \boldsymbol{w}_{n_i}\|^2 \\
&= \|(\mathbf{I} - \mathbf{P}_t)\boldsymbol{w}_{\tau^{(m_i)}(L_t^{(m_i)}-1)}^{(m_i)} + \mathbf{P}_t \boldsymbol{w}_{\tau^{m_i}(L_t^{(m_i)})} - \boldsymbol{w}_{n_i}\|^2 \\
&= \|(\mathbf{I} - \mathbf{P}_t)(\boldsymbol{w}_{\tau^{(m_i)}(L_t^{(m_i)}-1)}^{(m_i)} - \boldsymbol{w}_{n_i}) + \mathbf{P}_t(\boldsymbol{w}_{\tau^{m_i}(L_t^{(m_i)})} - \boldsymbol{w}_{n_i})\|^2,
\end{aligned}$$

where the first equality is because there is no update of $\boldsymbol{w}_t^{(m_i)}$ for $t \in \{\tau^{(m_i)}(L_t^{(m_i)}), \cdots, t\}$, and the second equality is by Eq. (5).

As $\mathbf{P}_t$ is the orthogonal projection matrix for the row space of $\mathbf{X}_t$, based on the rotational symmetry of the standard normal distribution, it follows that $\mathbb{E}\|\mathbf{P}_t(\boldsymbol{w}_{\tau^{m_i}(L_t^{(m_i)})} - \boldsymbol{w}_{n_i})\| = \frac{s}{d}\|\boldsymbol{w}_{\tau^{m_i}(L_t^{(m_i)})} -$

$\boldsymbol{w}_{n_i}\|^2$. Then we further calculate

$$
\begin{aligned}
\mathbb{E}[\|\boldsymbol{w}_t^{(m_i)} - \boldsymbol{w}_{n_i}\|^2] &= (1 - \frac{s}{d})\mathbb{E}[\|\boldsymbol{w}_{\tau^{(m_i)}(L_t^{(m_i)}-1)}^{(m_i)} - \boldsymbol{w}_{n_i}\|^2] + \frac{s}{d}\mathbb{E}[\|\boldsymbol{w}_{\tau^{(m_i)}(L_t^{(m_i)})} - \boldsymbol{w}_{n_i}\|^2] \\
&= (1 - \frac{s}{d})^{L_t^{(m_i)}}\mathbb{E}[\|\boldsymbol{w}_0^{(m_i)} - \boldsymbol{w}_{n_i}\|^2] \\
&\quad + \sum_{l=1}^{L_t^{(m_i)}}(1 - \frac{s}{d})^{L_t^{(m_i)}-l}\frac{s}{d}\mathbb{E}[\|\boldsymbol{w}_{\tau^{(m_i)}(L_t^{(m_i)})} - \boldsymbol{w}_{n_i}\|^2] \\
&= r^{L_t^{(m_i)}}\mathbb{E}[\|\boldsymbol{w}_{n_i}\|^2] + \sum_{l=1}^{L_t^{(m_i)}}(1-r)r^{L_t^{(m_i)}-l}\mathbb{E}[\|\boldsymbol{w}_{\tau^{(m_i)}(l)} - \boldsymbol{w}_{n_i}\|^2],
\end{aligned}
$$

where the second equality is derived by iterative calculation, and the last equality is because of $\boldsymbol{w}_0^{(m)} = \boldsymbol{0}$ for any expert $m$. Here we denote by $r = 1 - \frac{s}{d}$ to simplify notations. $\qquad\square$

## I.2 FINAL PROOF OF THEOREM 1

*Proof.* Based on Eq. (42) in Lemma 16, we obtain

$$
\mathbb{E}[\|\boldsymbol{w}_i^{(m_i)} - \boldsymbol{w}_{n_i}\|^2] = r^{L_i^{(m_i)}}\mathbb{E}[\|\boldsymbol{w}_{n_i}\|^2] + \sum_{l=1}^{L_i^{(m_i)}}(1-r)r^{L_i^{(m_i)}-l}\mathbb{E}[\|\boldsymbol{w}_{\tau^{(m_i)}(l)} - \boldsymbol{w}_{n_i}\|^2],
$$

where $\tau^{(m_i)}(L_i^{(m_i)}) = i$.

Then at any round $t \in \{2, \cdots, T_1\}$, we calculate the expected forgetting as:

$$
\begin{aligned}
\mathbb{E}[F_t] &= \frac{1}{t-1}\sum_{i=1}^{t-1}\mathbb{E}\Big[\|\boldsymbol{w}_t^{(m_i)} - \boldsymbol{w}_{n_i}\|^2 - \|\boldsymbol{w}_i^{(m_i)} - \boldsymbol{w}_{n_i}\|^2\Big] \\
&= \frac{1}{t-1}\sum_{i=1}^{t-1}\Big\{(r^{L_t^{(m_i)}} - r^{L_i^{(m_i)}})\mathbb{E}[\|\boldsymbol{w}_{n_i}\|^2] + \sum_{l=1}^{L_t^{(m_i)}}(1-r)r^{L_t^{(m_i)}-l}\mathbb{E}[\|\boldsymbol{w}_{\tau^{(m_i)}(l)} - \boldsymbol{w}_{n_i}\|^2] \\
&\quad - \sum_{j=1}^{L_i^{(m_i)}}(1-r)r^{L_i^{(m_i)}-j}\mathbb{E}[\|\boldsymbol{w}_{\tau^{(m_i)}(j)} - \boldsymbol{w}_{n_i}\|^2]\Big\}
\end{aligned}
$$

where $c_{i,j} = (1-r)(r^{L_t^{(m_i)}-L_i^{(m_i)}} + r^{L_t^{(m_i)}-j} - r^{j-L_i^{(m_i)}})$.

As task $i$'s ground truth $\boldsymbol{w}_{n_i}$ is randomly drawn from ground truth pool $\mathcal{W}$ with identical probability $\frac{1}{N}$, we have

$$
\mathbb{E}[\|\boldsymbol{w}_{n_i}\|^2] = \frac{1}{N}\sum_{n=1}^{N}\|\boldsymbol{w}_n\|^2.
$$

According to Lemma 8 and Proposition 1, each expert $m$ will converge to an expert set $\mathcal{M}_n$ before $t = T_1$. Therefore, we obtain

$$
\begin{aligned}
\mathbb{E}[\|\boldsymbol{w}_{\tau^{(m_i)}(l)} - \boldsymbol{w}_{n_i}\|^2] &= \mathbb{E}[\frac{1}{N}\sum_{n=1}^{N}\|\boldsymbol{w}_{\tau^{(m_i)}(l)} - \boldsymbol{w}_n\|^2] \\
&< \frac{1}{N}\sum_{n=1}^{N}\frac{1}{N}\sum_{n'=1}^{N}\|\boldsymbol{w}_{n'} - \boldsymbol{w}_n\|^2 \\
&= \frac{1}{N^2}\sum_{n \neq n'}^{N}\|\boldsymbol{w}_{n'} - \boldsymbol{w}_n\|^2, \qquad (43)
\end{aligned}
$$

where the inequality is because the expected error $\mathbb{E}[\|\boldsymbol{w}_{\tau^{(m_i)}(l)} - \boldsymbol{w}_n\|^2]$ per round for $t < T_1$ is smaller than the uniformly random routing strategy with expected error $\mathbb{E}[\|\boldsymbol{w}_{\tau^{(m_i)}(l)} - \boldsymbol{w}_n\|^2] = \frac{1}{N}\sum_{n'=1}^{N}\|\boldsymbol{w}_{n'} - \boldsymbol{w}_n\|^2$, and the last equality is because of $\|\boldsymbol{w}_n - \boldsymbol{w}_{n'}\|^2 = 0$ for $n' = n$.

Therefore, we finally obtain

$$
\begin{aligned}
\mathbb{E}[F_t] &< \frac{1}{t-1}\sum_{i=1}^{t-1}\Bigg\{ \frac{r^{L_t^{(m_i)}}-r^{L_i^{(m_i)}}}{N}\sum_{n=1}^{N}\|\boldsymbol{w}_n\|^2 + \frac{1-r}{N^2}\sum_{l=1}^{L_t^{(m_i)}}r^{L_t^{(m_i)}-l}\sum_{n\neq n'}^{N}\|\boldsymbol{w}_{n'}-\boldsymbol{w}_n\|^2 \\
&\quad -\frac{1-r}{N^2}\sum_{j=1}^{L_i^{(m_i)}}r^{L_i^{(m_i)}-j}\sum_{n\neq n'}^{N}\|\boldsymbol{w}_{n'}-\boldsymbol{w}_n\|^2\Bigg\} \\
&= \frac{1}{t-1}\sum_{i=1}^{t-1}\Bigg\{ \frac{r^{L_t^{(m_i)}}-r^{L_i^{(m_i)}}}{N}\sum_{n=1}^{N}\|\boldsymbol{w}_n\|^2 + \frac{1-r^{L_t^{(m_i)}}}{N^2}\sum_{n\neq n'}^{N}\|\boldsymbol{w}_{n'}-\boldsymbol{w}_n\|^2\Bigg\} \\
&\quad -\frac{1-r^{L_i^{(m_i)}}}{N^2}\sum_{n\neq n'}^{N}\|\boldsymbol{w}_{n'}-\boldsymbol{w}_n\|^2\Bigg\} \\
&= \frac{1}{t-1}\sum_{i=1}^{t-1}\Bigg\{ \frac{r^{L_t^{(m_i)}}-r^{L_i^{(m_i)}}}{N}\sum_{n=1}^{N}\|\boldsymbol{w}_n\|^2 + \frac{r^{L_i^{(m_i)}}-r^{L_t^{(m_i)}}}{N^2}\sum_{n\neq n'}^{N}\|\boldsymbol{w}_{n'}-\boldsymbol{w}_n\|^2\Bigg\}.
\end{aligned}
$$

For any $t\in\{T_1+1,\cdots,T\}$, based on Proposition 2, the expert model $\boldsymbol{w}_t^{(m)}=\boldsymbol{w}_{T_1}^{(m)}$ for any expert $m\in[M]$. Therefore, we calculate the caused forgetting

$$
\begin{aligned}
\mathbb{E}[F_t] &= \frac{1}{t-1}\sum_{i=1}^{t-1}\mathbb{E}\Big[\|\boldsymbol{w}_t^{(m_i)}-\boldsymbol{w}_{n_i}\|^2 - \|\boldsymbol{w}_i^{(m_i)}-\boldsymbol{w}_{n_i}\|^2\Big] \\
&= \frac{1}{t-1}\sum_{i=1}^{T_1}\mathbb{E}\Big[\|\boldsymbol{w}_{T_1}^{(m_i)}-\boldsymbol{w}_{n_i}\|^2 - \|\boldsymbol{w}_i^{(m_i)}-\boldsymbol{w}_{n_i}\|^2\Big] \\
&= \frac{T_1-1}{t-1}\mathbb{E}[F_t].
\end{aligned}
$$

Based on Eq. (42), we can also calculate the close form of the generalization error:

$$
\begin{aligned}
\mathbb{E}[G_T] &= \frac{1}{T}\sum_{i=1}^{T}\mathbb{E}[\|\boldsymbol{w}_T^{(m_i)}-\boldsymbol{w}_{n_i}\|^2] \\
&= \frac{1}{T}\sum_{i=1}^{T}\mathbb{E}[\|\boldsymbol{w}_{T_1}^{(m_i)}-\boldsymbol{w}_{n_i}\|^2] \\
&= \frac{1}{T}\sum_{i=1}^{T}\left( r^{L_{T_1}^{(m_i)}}\mathbb{E}[\|\boldsymbol{w}_{n_i}\|^2] + \sum_{l=1}^{L_{T_1}^{(m_i)}}(1-r)r^{L_{T_1}^{(m_i)}-l}\mathbb{E}[\|\boldsymbol{w}_{\tau^{(m_i)}(l)}-\boldsymbol{w}_{n_i}\|^2]\right) \\
&< \frac{\sum_{i=1}^{T}r^{L_{T_1}^{(m_i)}}}{NT}\sum_{n=1}^{N}\|\boldsymbol{w}_n\|^2 + \frac{1-r}{N^2 T}\sum_{i=1}^{T}\sum_{l=1}^{L_{T_1}^{(m_i)}}r^{L_{T_1}^{(m_i)}-l}\sum_{n\neq n'}^{N}\|\boldsymbol{w}_{n'}-\boldsymbol{w}_n\|^2 \\
&= \frac{\sum_{i=1}^{T}r^{L_{T_1}^{(m_i)}}}{NT}\sum_{n=1}^{N}\|\boldsymbol{w}_n\|^2 + \frac{\sum_{i=1}^{T}(1-r^{L_{T_1}^{(m_i)}})}{N^2 T}\sum_{n\neq n'}^{N}\|\boldsymbol{w}_{n'}-\boldsymbol{w}_n\|^2,
\end{aligned}
$$

where the inequality is because of Eq. (43). $\qquad\square$

## J    PROOF OF THEOREM 2

*Proof.* For any round $t\in\{1,\cdots,T_1\}$, the forgetting is the same as the case of $M>N$ in Theorem 1, as tasks randomly arrive and are routed to different experts. Therefore, we skip the proof for $t<T_1$.

For $t\in\{T_1+1,\cdots,T\}$, the router will route tasks in the same cluster to each expert per round. Therefore, we divide the $t$ rounds into two subintervals: $i\in\{1,\cdots,T_1\}$ and $i\in\{T_1+1,\cdots,t\}$ to

calculate the forgetting as

$$\mathbb{E}[F_t] = \frac{1}{t-1}\sum_{i=1}^{t}\mathbb{E}\Big[\|\boldsymbol{w}_t^{(m_i)} - \boldsymbol{w}_{n_i}\|^2 - \|\boldsymbol{w}_i^{(m_i)} - \boldsymbol{w}_{n_i}\|^2\Big]$$

$$= \frac{1}{t-1}\sum_{i=1}^{T_1}\Big\{(r^{L_t^{(m_i)}} - r^{L_i^{(m_i)}})\mathbb{E}[\|\boldsymbol{w}_{n_i}\|^2] + \sum_{l=1}^{L_t^{(m_i)}}(1-r)r^{L_t^{(m_i)}-l}\mathbb{E}[\|\boldsymbol{w}_{\tau^{(m_i)}(l)} - \boldsymbol{w}_{n_i}\|^2]$$

$$- \sum_{j=1}^{L_i^{(m_i)}}(1-r)r^{L_i^{(m_i)}-j}\mathbb{E}[\|\boldsymbol{w}_{\tau^{(m_i)}(j)} - \boldsymbol{w}_{n_i}\|^2]\Big\}+$$

$$\frac{1}{t-1}\sum_{i=T_1+1}^{T}\Big\{(r^{L_t^{(m_i)}} - r^{L_i^{(m_i)}})\mathbb{E}[\|\boldsymbol{w}_{n_i}\|^2] + \underbrace{\sum_{l=L_{T_1}^{(m_i)}+1}^{L_t^{(m_i)}}(1-r)r^{L_t^{(m_i)}-l}\mathbb{E}[\|\boldsymbol{w}_{\tau^{(m_i)}(l)} - \boldsymbol{w}_{n_i}\|^2]}_{\text{term a}}$$

$$-\underbrace{\sum_{j=L_{T_1}^{(m_i)}+1}^{L_i^{(m_i)}}(1-r)r^{L_i^{(m_i)}-j}\mathbb{E}[\|\boldsymbol{w}_{\tau^{(m_i)}(j)} - \boldsymbol{w}_{n_i}\|^2]}_{\text{term b}}\Big\}$$

where the first term in the second equality is similarly derived as forgetting in Eq. (19). For the term a − term b in the second equality, we calculate

$$\text{term a} - \text{term b}$$

$$= \sum_{l=L_{T_1}^{(m_i)}+1}^{L_t^{(m_i)}-L_i^{(m_i)}+L_{T_1}^{(m_i)}+1}(1-r)r^{L_t^{(m_i)}-l}\mathbb{E}[\|\boldsymbol{w}_{\tau^{(m_i)}(l)} - \boldsymbol{w}_{n_i}\|^2]$$

$$= r^{L_t^{(m_i)}-L_{T_1}^{(m_i)}}(1 - r^{L_t^{(m_i)}-L_i^{(m_i)}})\mathbb{E}[\|\boldsymbol{w}_{\tau^{(m_i)}(l)} - \boldsymbol{w}_{n_i}\|^2]$$

$$= \frac{r^{L_t^{(m_i)}-L_{T_1}^{(m_i)}-1}(1 - r^{L_t^{(m_i)}-L_i^{(m_i)}})}{N}\sum_{n=1}^{N}\mathbb{E}[\|\boldsymbol{w}_{\tau^{(m_i)}(l)} - \boldsymbol{w}_{n}\|^2]$$

$$= \frac{r^{L_t^{(m_i)}-L_{T_1}^{(m_i)}-1}(1 - r^{L_t^{(m_i)}-L_i^{(m_i)}})}{N}\sum_{n=1}^{N}\sum_{n,n'\in\mathcal{W}_k}\frac{\|\boldsymbol{w}_{n'} - \boldsymbol{w}_{n}\|^2}{|\mathcal{W}_k|},$$

where the third equality is derived by $\mathbb{E}[\|\boldsymbol{w}_{\tau^{(m_i)}(l)} - \boldsymbol{w}_{n_i}\|^2] = \frac{1}{N}\sum_{n=1}^{N}\mathbb{E}[\|\boldsymbol{w}_{\tau^{(m_i)}(l)} - \boldsymbol{w}_{n}\|^2]$, and the last equality is because the router always routes task $\boldsymbol{w}_{\tau^{(m_i)}(l)} = \boldsymbol{w}_{n'}$ within the same cluster $\mathcal{W}_k$ to expert $m_i$. Taking the result of term a − term b into $\mathbb{E}[F_t]$, we obtain

$$\mathbb{E}[F_t] < \frac{1}{t-1}\sum_{i=1}^{t-1}\frac{r^{L_t^{(m_i)}} - r^{L_i^{(m_i)}}}{N}\sum_{n=1}^{N}\|\boldsymbol{w}_{n}\|^2 + \frac{1}{t-1}\sum_{i=1}^{T_1}\frac{r^{L_i^{(m_i)}} - r^{L_{T_1}^{(m_i)}}}{N^2}\sum_{n\neq n'}^{N}\|\boldsymbol{w}_{n'} - \boldsymbol{w}_{n}\|^2$$

$$+ \frac{1}{t-1}\sum_{i=T_1+1}^{t}\frac{r^{L_t^{(m_i)}-L_{T_1}^{(m_i)}-1}(1 - r^{L_t^{(m_i)}-L_i^{(m_i)}})}{N}\sum_{n=1}^{N}\sum_{n,n'\in\mathcal{W}_k}\frac{\|\boldsymbol{w}_{n'} - \boldsymbol{w}_{n}\|^2}{|\mathcal{W}_k|}.$$

Similarly, we can calculate the overall generalization error as

$$\mathbb{E}[G_T] = \frac{1}{T} \sum_{i=1}^{T} \mathbb{E}[\|\boldsymbol{w}_T^{(m_i)} - \boldsymbol{w}_{n_i}\|^2]$$

$$= \frac{1}{T} \sum_{i=1}^{T} \left( r^{L_T^{(m_i)}} \mathbb{E}[\|\boldsymbol{w}_{n_i}\|^2] + \sum_{l=1}^{L_T^{(m_i)}} (1-r)r^{L_T^{(m_i)}-l} \mathbb{E}[\|\boldsymbol{w}_{\tau^{(m_i)}(l)} - \boldsymbol{w}_{n_i}\|^2] \right)$$

$$= \frac{1}{T} \sum_{i=1}^{T} r^{L_T^{(m_i)}} \mathbb{E}[\|\boldsymbol{w}_{n_i}\|^2] + \frac{1}{T} \sum_{i=1}^{T_1} \left\{ \sum_{l=1}^{L_{T_1}^{(m_i)}} (1-r)r^{L_T^{(m_i)}-l} \underbrace{\mathbb{E}[\|\boldsymbol{w}_{\tau^{(m_i)}(l)} - \boldsymbol{w}_{n_i}\|^2]}_{\text{term 1}} \right.$$

$$+ \sum_{l=L_{T_1}^{(m_i)}+1}^{L_T^{(m_i)}} (1-r)r^{L_T^{(m_i)}-l} \underbrace{\mathbb{E}[\|\boldsymbol{w}_{\tau^{(m_i)}(l)} - \boldsymbol{w}_{n_i}\|^2]}_{\text{term 2}} \Bigg\}$$

$$+ \frac{1}{T} \sum_{i=T_1+1}^{T} \left\{ \sum_{l=1}^{L_{T_1}^{(m_i)}} (1-r)r^{L_T^{(m_i)}-l} \underbrace{\mathbb{E}[\|\boldsymbol{w}_{\tau^{(m_i)}(l)} - \boldsymbol{w}_{n_i}\|^2]}_{\text{term 2}} \right.$$

$$+ \sum_{l=L_{T_1}^{(m_i)}+1}^{L_T^{(m_i)}} (1-r)r^{L_T^{(m_i)}-l} \underbrace{\mathbb{E}[\|\boldsymbol{w}_{\tau^{(m_i)}(l)} - \boldsymbol{w}_{n_i}\|^2]}_{\text{term 3}} \Bigg\},$$

where the second equality is derived by Eq. (42), and the third equality is derived by dividing $T$ rounds into two subintervals $\{1, \cdots, T_1\}$ and $\{T_1 + 1, \cdots, T\}$.

In the above equation, term 1 means both $\boldsymbol{w}_{n_i}$ and $\boldsymbol{w}_{\tau^{(m_i)}(l)}$ are randomly drawn from the $N$ ground truths, due to the fact that $i \leq T_1$ and $\tau^{(m_i)}(l) \leq T_1$. Term 2 means one of $\boldsymbol{w}_{n_i}$ and $\boldsymbol{w}_{\tau^{(m_i)}(l)}$ is randomly drawn from the $N$ ground truths before the $T_1$-th round, while the other one has fixed to a cluster $\mathcal{M}_k$ after the $T_1$-th round (i.e., $i \leq T_1$ and $\tau^{(m_i)}(l) > T_1$ or $i > T_1$ and $\tau^{(m_i)}(l) \leq T_1$). Term 3 means $\boldsymbol{w}_{n_i}$ and $\boldsymbol{w}_{\tau^{(m_i)}(l)}$ are in the same cluster $\mathcal{M}_k$, as $i > T_1$ and $\tau^{(m_i)}(l) > T_1$.

For $i \in \{1, \cdots, T_1\}$, based on the proof of Theorem 1 in Appendix I, we obtain term 1:

$$\mathbb{E}[\|\boldsymbol{w}_{\tau^{(m_i)}(l)} - \boldsymbol{w}_{n_i}\|^2] < \frac{1}{N^2} \sum_{n \neq n'} \|\boldsymbol{w}_n - \boldsymbol{w}_{n'}\|^2.$$

While for $i \in \{T_1 + 1, \cdots, t\}$, we calculate term 3 as:

$$\mathbb{E}[\|\boldsymbol{w}_{\tau^{(m_i)}(l)} - \boldsymbol{w}_{n_i}\|^2] = \frac{1}{N} \sum_{n=1}^{N} \mathbb{E}[\|\boldsymbol{w}_{\tau^{(m_i)}(l)} - \boldsymbol{w}_n\|^2]$$

$$= \frac{1}{N} \sum_{n=1}^{N} \sum_{n,n' \in \mathcal{W}_k} \frac{\|\boldsymbol{w}_{n'} - \boldsymbol{w}_n\|^2}{|\mathcal{W}_k|}.$$

Finally, we calculate term 2:

$$\mathbb{E}[\|\boldsymbol{w}_{\tau^{(m_i)}(l)} - \boldsymbol{w}_{n_i}\|^2] = \frac{1}{N} \sum_{n=1}^{N} \mathbb{E}[\|\boldsymbol{w}_{\tau^{(m_i)}(l)} - \boldsymbol{w}_n\|^2]$$

$$< \frac{1}{N} \sum_{n=1}^{N} \frac{1}{K} \sum_{k=1}^{K} \frac{1}{|\mathcal{W}_k|} \sum_{n' \in \mathcal{W}_k} \|\boldsymbol{w}_{n'} - \boldsymbol{w}_n\|^2.$$

Based on the expressions of the three terms above, we obtain

$$\mathbb{E}[G_T] < \frac{1}{T}\sum_{i=1}^{T}\frac{r^{L_T^{(m_i)}}}{N}\sum_{n=1}^{N}\|\boldsymbol{w}_n\|^2 + \frac{1}{T}\sum_{i=1}^{T_1}\frac{r^{L_T^{(m_i)}-L_{T_1}^{(m_i)}}(1-r^{L_{T_1}^{(m_i)}})}{N^2}\sum_{n\neq n'}^{N}\|\boldsymbol{w}_{n'}-\boldsymbol{w}_n\|^2$$

$$+\frac{1}{T}\sum_{i=1}^{T_1}\frac{1-r^{L_T^{(m_i)}-L_{T_1}^{(m_i)}-1}}{N}\sum_{n=1}^{N}\frac{1}{K}\sum_{k=1}^{K}\frac{1}{|\mathcal{W}_k|}\sum_{n'\in\mathcal{W}_k}\|\boldsymbol{w}_{n'}-\boldsymbol{w}_n\|^2$$

$$+\frac{1}{T}\sum_{i=T_1+1}^{T}\frac{r^{L_T^{(m_i)}-L_{T_1}^{(m_i)}}(1-r^{L_{T_1}^{(m_i)}})}{N}\sum_{n=1}^{N}\frac{1}{K}\sum_{k=1}^{K}\frac{1}{|\mathcal{W}_k|}\sum_{n'\in\mathcal{W}_k}\|\boldsymbol{w}_{n'}-\boldsymbol{w}_n\|^2$$

$$+\frac{1}{T}\sum_{i=T_1+1}^{T}\frac{1-r^{L_T^{(m_i)}-L_{T_1}^{(m_i)}-1}}{N}\sum_{n=1}^{N}\sum_{n,n'\in\mathcal{W}_k}\frac{\|\boldsymbol{w}_{n'}-\boldsymbol{w}_n\|^2}{|\mathcal{W}_k|},$$

which completes the proof. $\qquad\square$

