# OpenReview forum: "Theory on Mixture-of-Experts in Continual Learning"
_ICLR.cc/2025/Conference — ICLR 2025 Spotlight_

### Official Review · Reviewer_nnJV · 2024-10-31

**Soundness:** 3
**Presentation:** 3
**Contribution:** 3
**Rating:** 8
**Confidence:** 4

**Summary:**

This work studies the theoretical understanding of mixture-of-experts (MoEs) module in continual learning. To examine the role of MoEs, the authors conduct experiments on overparameterized linear regression tasks. Theoretically, this work identifies the importance of the gating network update’s termination and analyzes the catastrophic forgetting and generalization.

**Strengths:**

This work is written clearly and well structured. Overall, the theoretical analysis over MoEs is necessary in developing MoE-based large models, and this work discusses some insights into the catastrophic forgetting and generalization. The experiments include the overparameterized linear regression and MNIST cases.

**Weaknesses:**

(1) The scope of this work seems a bit wide according to the title. I suggest to use terms like “theoretical understanding” to modify.

(2) The theoretical analysis is mainly on overparameterized linear regression cases, which might be a limitation in this work as nonlinear deep neural network cases can be more practical.

(3) There are some related work on MoE theories, continual learning with MoEs or MoEs for adaptation in the field that require discussions [1-4].

Reference:

[1] Nguyen H D, Chamroukhi F. Practical and theoretical aspects of mixture‐of‐experts modeling: An overview[J]. Wiley Interdisciplinary Reviews: Data Mining and Knowledge Discovery, 2018, 8(4): e1246.

[2] Jerfel, Ghassen, et al. "Reconciling meta-learning and continual learning with online mixtures of tasks." Advances in neural information processing systems 32 (2019).

[3] Wang Q, Van Hoof H. Learning expressive meta-representations with mixture of expert neural processes[J]. Advances in neural information processing systems, 2022, 35: 26242-26255.

[4] Lee S, Ha J, Zhang D, et al. A neural dirichlet process mixture model for task-free continual learning[J]. arXiv preprint arXiv:2001.00689, 2020.

---

***Post Rebuttal**

After reading the rebuttal, all my questions are well answered. I updated the score.

**Questions:**

(1) In Line 76-78, it sates the MoE enhances performance over the single expert case, which seems trivial. Is the condition, such as with the same model complexity, missing in the statement?

(2) In Line221-224, it seems the update of model parameterizes use the inversion matrix, which brings more computational overhead over gradient-based methods.

(3) In Line 293-294, it seems the early termination achieves stable convergence, but the motivation is also related to overfitting for alleviating imbalanced loads. So does the early termination improve balanced loads?

(4) In Line 532, “the first theoretical analysis of MoE” is overstated considering the existing work [1]. And I am also wondering how the developed strategy works in more complicated experiments.

---

> ### Author Response · Authors · 2024-11-22
> **Response to Reviewer nnJv (Part 1)**
>
> Thank you for your thorough reviews and constructive comments. We provide our responses to your comments below and have made major revisions in our revised manuscript. To enhance clarity, we have highlighted the revised text in blue for easy identification.
>
> **Q1.** [Title] The scope of this work seems a bit wide according to the title. I suggest to use terms like “theoretical understanding” to modify.
>
> **Response:** Thanks for the suggestion. To avoid any confusion during the review process, we have kept the current title. However, if the paper is accepted, we will follow your recommendation to revise the title accordingly.
>
>
> **Q2.** [Overparameterized linear regression] The theoretical analysis is mainly on overparameterized linear regression cases, which might be a limitation in this work as nonlinear deep neural network cases can be more practical.
>
> **Response:** Thank you for highlighting this point. Our focus on overparameterized linear regression aligns with state-of-the-art theoretical works in continual learning (e.g., Evron et al., (2022,2023); Lin et al., (2023)). The main motivation is to extract the key insights of forgetting and generalization in a tractable scenario without the complication brought by the model. Nonlinear deep neural networks (DNNs), while highly practical, are significantly more challenging to analyze rigorously. Current theoretical techniques for analyzing DNNs remain considerably underdeveloped, even for a single task—let alone in a continual learning setting, where analyzing task interactions over time introduces further complication.
>
> This work represents the first theoretical analysis of MoE’s impact on continual learning. By adopting the overparameterized linear regression framework, we derived explicit expressions for expected forgetting and generalization errors in both the single-expert case (Proposition 4) and the MoE model (Theorems 1 and 2). These results clearly characterize the advantages of MoE models over single experts, and we further analyzed how these benefits depend on system parameters and algorithms.
>
> Importantly, the theoretical insights derived from this framework could guide the practical design of MoE models for nonlinear DNNs, as demonstrated in our real-data experiments in Section 6.
>
>
> **Q3.** [Related works] There are some related work on MoE theories, continual learning with MoEs or MoEs for adaptation in the field that require discussions [1-4] ...
>
> **Response:** We thank the reviewer for pointing out these related works and have included discussions on them in our revised manuscript. Further, we clarify that among these works,  [1] is the only theoretical work and focuses on the analysis of MoE modeling, but does not address continual learning. Moreover, this work also does not have theoretical results on non-linear DNNs. The other three works are all empirical works related with CL ([2]), MoE ([3]), and MoE for CL ([4]), and they do not have any theoretical results. While these works are valuable, our study distinguishes itself by offering the first theoretical insights into the dynamics of MoE models in continual learning. We have cited and discussed these works where appropriate, ensuring proper acknowledgment of their contributions.

---

> ### Author Response · Authors · 2024-11-22
> **Response to Reviewer nnJv (Part 2)**
>
> **Q4.** [Model complexity] In Line 76-78, it states the MoE enhances performance over the single expert case, which seems trivial. Is the condition, such as with the same model complexity, missing in the statement?
>
> **Response:** Thanks for this great question! Under the same model complexity,  the learning performance of a vanilla system (without MoE) remains significantly inferior to that of the MoE, which can be observed from  the explicit expressions for the single expert case in Proposition 4. To elaborate, in the vanilla system, both forgetting and generalization error stem primarily from model gaps between tasks, and increasing model complexity fails to mitigate these losses over the training rounds $T$. In contrast, the MoE system ensures that, from round $T_1$ onward, enabled by correct routing, each expert specializes in tasks within the same cluster. This specialization significantly reduces the forgetting and generalization errors caused by task model gaps, as detailed in Theorems 1 and 2.
>
> We also clarify that our result shows far more than the simple fact that “the MoE significantly enhances the learning performance over the single expert case”. Instead, as stated in Line 75, we “provide **explicit expressions** of the expected forgetting and generalization error to **characterize the benefit** of MoE on the performance of CL”. By comparing our derived expressions for MoE models (Theorems 1 and 2) with those for the single expert case (Proposition 4), we reveal how system parameters (e.g., number of experts, task similarities) and the training algorithm contribute to the MoE's advantages compared to the single expert case. Such insights, presented in Lines 79-83, provide valuable theoretical guidance for designing MoE systems, which are non-trivial.
>
> **Q5.** [Inversion matrix] In Line 221-224, it seems the update of model parameterizes use the inversion matrix, which brings more computational overhead over gradient-based methods.
>
> **Response:** We appreciate the reviewer pointing this out. While Eq. (5) uses the inversion matrix to derive the optimal solution to the optimization problem (4), this is primarily for theoretical analysis. In practice, we can adopt gradient-based methods to update $w_t^{(m_t)}$. Specifically, $y-X_t^\top w_{t-1}^{(m_t)}$ is the corresponding gradient, and $X_t(X_t^\top X_t)^{-1}$ can be replaced with a learning rate $\eta$. Our theoretical results remain valid under gradient-based updates, as the gradient $y-X_t^\top w_{t-1}^{(m_t)}$ converges to zero with correct routing in the MoE system. Additionally, our real-data experiments employ gradient-based methods to train DNNs, verifying that the theoretical insights extend to practical scenarios.
>
>
> **Q6.** [Early termination] In Line 293-294, it seems the early termination achieves stable convergence, but the motivation is also related to overfitting for alleviating imbalanced loads. So does the early termination improve balanced loads?
>
> **Response:** This is an excellent question. To clarify, reducing overfitting for alleviating imbalance loads is the primary motivation behind our multi-objective training loss design (as noted in Key Design I, Line 232). To ensure clarity, we have revised Line 231 of the manuscript to explicitly state this.
>
> As pointed out by the reviewer, the motivation for early termination lies in stabilizing the convergence of the MoE system and ensuring balanced loads across tasks (as described in Lines 263-264). Specifically, as analyzed in Proposition 3, after early termination of the gating network training, there are $|M_{n_t}|$ experts specializing in task $n_t$. Then for any subsequent task arrival $n_t$, the router selects an expert from expert set $M_{n_t}$ with equal probability of $\frac{1}{|M_{n_t}|}$, ensuring balanced loads across experts. We have empirically validated this in Figures 4 and 5 of Appendix A.4, which demonstrate how early termination affects load balancing and average learning performance, respectively.

---

> > ### Comment · Reviewer_nnJV · 2024-11-22
> >
> > Thanks for the response. After reading the rebuttal, I have updated the review and rating. Overall, the revised manuscript well claried the scope and well polished some statements.

---

> > > ### Author Response · Authors · 2024-11-22
> > >
> > > We would like to express our sincere thanks to the reviewer for re-evaluating our work. We will continue to improve our manuscript by incorporating your valuable insights.

---

> ### Author Response · Authors · 2024-11-22
> **Response to Reviewer nnJv (Part 3)**
>
> **Q7.** [Statement and experiments] In Line 532, “the first theoretical analysis of MoE” is overstated considering the existing work [1]. And I am also wondering how the developed strategy works in more complicated experiments.
>
> **Response:** We clarify that our statement in Line 532 specifically refers to “our work is the first theoretical analysis of MoE and its impact on learning performance **in continual learning**”, which aligns with the title, the scope, and the former statements of this work. While [1] provides a theoretical analysis of MoE modeling, it does not address continual learning, nor does it analyze MoE's impact on performance in such settings. Therefore, our claim accurately reflects the novelty and focus of our contributions.
>
> Despite the limited time, we made every effort to expand our experiments to address your constructive comments, incorporating more complex datasets such as CIFAR-10, CIFAR-100, and Tiny ImageNet. In Section 6, we replaced the MNIST experiments with CIFAR-10 experiments, and additional experimental details are provided in Appendices A.3-A.6 of the revised manuscript. In these supplementary experiments, we evaluated not only forgetting and generalization errors but also the test accuracy, confirming that the MoE model significantly outperforms the single model in continual learning. Furthermore, the key insights from our theoretical results remain valid, such as early termination ensuring stable convergence with balanced expert loads and the observation that increasing the number of experts does not always improve learning performance.
>
>
> **Finally, if our response resolves your concerns to a satisfactory level, we wonder if the reviewer could kindly consider raising the score of your evaluation. Certainly, we are more than happy to address any further questions that you may have during the discussion period. We thank the reviewer again for the helpful comments and suggestions for our work.**

---

### Official Review · Reviewer_WfDL · 2024-11-03

**Soundness:** 3
**Presentation:** 3
**Contribution:** 2
**Rating:** 6
**Confidence:** 4

**Summary:**

The paper "Theory on Mixture-of-Experts in Continual Learning" analyzes the Mixture-of-Experts (MoE) model's effectiveness in addressing catastrophic forgetting in continual learning (CL). It establishes that MoE can enhance learning by diversifying expert specialization through a gating network, which routes tasks efficiently. The study provides theoretical insights into expected forgetting and generalization error, demonstrating that MoE outperforms single-expert models, especially with diverse data distributions. However, it notes that adding more experts requires additional training rounds before achieving convergence. Empirical results support the theoretical claims, extending the findings to deep neural networks (DNNs) for practical algorithms

**Strengths:**

1) Theoretical Foundations: The paper provides a comprehensive theoretical analysis of MoE in the context of continual learning, establishing clear benefits over single-expert models through explicit expressions for expected forgetting and generalization error.

2) Load Balancing: The model ensures balanced utilization of experts, which can lead to improved generalization performance as it reduces the risk of overloading any single expert with too many tasks.

3) Empirical Validation: Experiments conducted on both synthetic and real datasets support the theoretical findings, demonstrating that MoE can effectively improve learning performance across diverse scenarios. It is interesting to note that even with a higher number of experts than tasks, the model might not perform well.

**Weaknesses:**

1) Validity of Proposition (4): The model gap term ∑n≠n′∥wn−wn′∥2\sum_{n \neq n'} \|w_n - w_{n'}\|^2∑n=n′​∥wn​−wn′​∥2 only considers the Euclidean distance between weights. This may not fully capture the complex relationships between tasks. In practice, tasks could overlap in non-trivial ways (e.g., in feature space or output space), and simple weight differences do not reflect true "task divergence" accurately.

2) Limited Experiments: Though the main contribution is to present the theoretical analysis of forgetting and generalization error for MoE in CL, the main objective of the model is to reduce catastrophic forgetting. Without presenting enough empirical evaluation in terms of accuracy in continual learning settings for benchmark datasets like CIFAR10/100, ImageNet is not enough. I would encourage the authors to extend the simulation to the more complex dataset. For example, SEED [1] uses a mixture of expert networks and selects a single expert to finetune downstream tasks. If the authors could provide details of this work by forgetting a generalization error it would provide better judgment of where the current state-of-the-art MoE methods for CL stands in terms of forgetting and generalization error and strengthen the contribution of the work.

[1] Rypeść, Grzegorz, et al. "Divide and not forget: Ensemble of selectively trained experts in Continual Learning." arXiv preprint arXiv:2401.10191 (2024).

**Questions:**

1) Line 56-57 “One learning task arrives in each round and its dataset is generated with ground truth randomly drawn from a shared pool encompassing N unknown linear model”. What is the intuition behind generating the ground truth from the linear model?

2) It might make sense if the number of experts is less than the number of tasks But in Line Line 69 if M > N implies the number of experts more than the number of tasks. This is not continual learning if you are training task-specific experts. Furthermore, in Section 4, you have done the main study with more experts than tasks (Line 296).

3) In Figure 2, on which datasets the experiment is performed?

4) For MNIST which DNN model is used? As from the setup number of task N = 3, how did the 10 classes were split?


## After Reading Comments

I raised my score to 6 as an experiment on MNIST and CIFAR10 is not enough.

Again, thank you for your contributions.

---

> ### Author Response · Authors · 2024-11-22
> **Response to Reviewer WfDL (Part 1)**
>
> Thank you for your thorough reviews and constructive comments. We provide our responses to your comments below and have made major revisions in our revised manuscript. To enhance clarity, we have highlighted the revised text in blue for easy identification.
>
> **Q1.** [Task correlations] Validity of Proposition (4): The model gap term … only considers the Euclidean distance between weights. This may not fully capture the complex relationships between tasks. In practice, tasks could overlap in non-trivial ways (e.g., in feature space or output space), and simple weight differences do not reflect true "task divergence" accurately.
>
> **Response:** We appreciate the reviewer’s insightful observation. Capturing task correlations is indeed a complex and open problem without a universally accepted metric. From a theoretical perspective, we followed widely accepted practices in prior works (e.g., Evron et al., (2022,2023); Gunasekar et al., (2018); Lin et al., (2023)), using Euclidean distance as it is both popular and effective in analyzing task relationships.
>
> We also welcome suggestions for alternative metrics and would be happy to explore them in future work. Importantly, while the choice of correlation metric may alter the specific forms of our derived expressions for forgetting and generalization errors, we expect that it does not affect the underlying insights. For instance, as shown in Theorem 1, forgetting and generalization errors primarily arise from task model gaps due to incorrect routing during the expert exploration phase ($t < T_1$). After $T_1$, consistent and correct routing ensures each expert specializes in tasks within the same cluster, eliminating additional losses that are related with task correlations. These essential dynamics of CL would still hold to a large extent.
>
>
> **Q2.** [Limited Experiments] ... Without presenting enough empirical evaluation in terms of accuracy in continual learning settings for benchmark datasets like CIFAR10/100, ImageNet is not enough. I would encourage the authors to extend the simulation to the more complex dataset ...
>
> **Response:** Thank you for your constructive suggestion. Despite the limited time, we made every effort to expand our experiments to include more complex datasets such as CIFAR-10, CIFAR-100, and Tiny ImageNet. In Section 6, we replaced the MNIST experiments with CIFAR-10 experiments, and additional experimental details are provided in Appendices A.3-A.6 of the revised manuscript. In these supplementary experiments, we evaluated not only forgetting and generalization errors but also the test accuracy, confirming that the MoE model significantly outperforms the single model in continual learning. Furthermore, the key insights from our theoretical results remain valid, such as early termination ensuring stable convergence with balanced expert loads and the observation that increasing the number of experts does not always improve learning performance.
>
> From a theoretical perspective, we expect that the exact expressions of forgetting and generalization errors for [1] are likely to differ in form from those presented in our Theorems 1 and 2. However, we expect the fundamental nature of the terms in these expressions to remain consistent, providing similar core insights. For example, under SEED in [1], we still anticipate that, after sufficient rounds of fine-tuning, each expert would eventually specialize in distinct sets of tasks aligned with their respective distributions. In this case, forgetting and generalization errors would still mainly arise from task model gaps. This presents an interesting avenue for future exploration.
>
> **Q3.** [Model setup] Line 56-57 “One learning task arrives in each round and its dataset is generated with ground truth randomly drawn from a shared pool encompassing N unknown linear model”. What is the intuition behind generating the ground truth from the linear model?
>
> **Response:** This setup is standard in the existing theoretical literature (e.g., Chen et al., (2022); Evron et al., (2022); Lin et al., (2023); Huang et al., (2024)) to generate a sequence of tasks in CL. Randomly sampled ground truth can capture the diverse nature of practical tasks and also serve as a benchmark for theoretically evaluating the performance of the estimated model. The flexibility of this setup also lies in not restricting $N$, making it applicable to a wide range of real-world scenarios. By adopting this commonly used assumption, we ensure consistency with existing works and maintain the generality of our theoretical framework.

---

> ### Author Response · Authors · 2024-11-22
> **Response to Reviewer WfDL (Part 2)**
>
> **Q4.** [Less experts than tasks] It might make sense if the number of experts is less than the number of tasks But in Line 69 if M > N implies the number of experts more than the number of tasks. This is not continual learning if you are training task-specific experts. Furthermore, in Section 4, you have done the main study with more experts than tasks (Line 296).
>
> **Response:** Thank you for raising this important clarification. We call the reviewer’s attention that the more general $M<N$ scenario has been comprehensively addressed in our work (in Appendices C, E, F, and G, where we provide full versions of Propositions 1-3). Additionally, in the main body of the manuscript, we included Theorem 2 specifically for the $M<N$ case, emphasizing its significance. Our presentation focuses on the $M>N$ case to facilitate better understanding of the core insights of the theory. In our revised manuscript, we have further highlighted the results for the $M<N$ scenario following each lemma and proposition in Section 4.
>
> We also clarify that the fact that $M>N$ does not imply that our MoE system is limited to only $N$ training rounds. Instead, our system operates over $T$ rounds, where tasks continually arrive for training. In this case, multiple experts may specialize in the same task.
>
>
> **Q5.** [Figure 2] In Figure 2, on which datasets the experiment is performed?
>
> **Response:** Figure 2 is operated on synthetic data. The detailed data generation process is described in Appendix A for complete transparency and reproducibility.
>
> **Q6.** [DNN model] For MNIST which DNN model is used? As from the setup number of task N = 3, how did the 10 classes were split?
>
> **Response:** The details of DNN model are included in Appendix A.4 due to the page limitation:
> “We use a five-layer neural network, consisting of two convolutional layers and three fully connected layers. ReLU activation is applied to the first four layers, while Sigmoid is used for the final layer. The first convolutional layer is followed by a 2D max-pooling operation with a stride of 2. In our experiments with the MNIST dataset, we defined three distinct tasks: identifying whether a given image depicts the number 1, 4, or 7. We chose these numbers because they exhibit relatively distinct features compared to other numbers in the dataset. For each task arrival $t \in [T]$, we first randomly determine the task type (e.g., recognizing the number 1). We then randomly select 100 samples from the dataset, filtering for samples corresponding to the task type (e.g., only images of the number 1). As a result, different tasks indeed have different data distributions and distinct feature signals after the filtering process.”
> Please note that we have also followed your constructive suggestion to extend our experiments to more complex networks on CIFAR-10, CIFAR-100, and Tiny ImageNet datasets.
>
>
> **Finally, if our response resolves your concerns to a satisfactory level, we wonder if the reviewer could kindly consider raising the score of your evaluation. Certainly, we are more than happy to address any further questions that you may have during the discussion period. We thank the reviewer again for the helpful comments and suggestions for our work.**

---

> ### Comment · Reviewer_WfDL · 2024-11-25
>
> After reading all the comments on my concerns, I raised my score. Indeed, the work is great. I had minor concerns about whether this theoretical foundation would hold if the model encounters a large number of tasks with a drastic shift in the distribution.

---

> > ### Author Response · Authors · 2024-11-25
> >
> > We sincerely thank the reviewer for re-evaluating our work and for raising new concerns. Below, we provide our responses to address your comments:
> >
> > **[CIFAR-100 and Tiny ImageNet]** Actually, we have conducted extensive experiments on CIFAR-100 and Tiny ImageNet datasets. Due to space limitations, the details of these experiments are presented in Appendices A.5 and A.6 of our revised manuscript. We have also emphasized this inclusion in Lines 486-487 of Section 6 in the main text.
> >
> > **[Large number of distinct tasks]** If there is a large number of tasks with drastic shift in the distribution, our theoretical results still hold. Specifically,
> >
> > 1. Exploration phase: As outlined in Proposition 1, all experts still undergo an exploration phase where they are exposed to diverse tasks. By the end of the exploration phase, each expert will still specialize in a cluster of similar tasks, while tasks across different clusters can be very different from each other due to the drastic shifts in data distributions.
> >
> > 2. Load balancing: Concurrently, the gating network still adaptively updates its parameters to balance expert workloads, as analyzed in Proposition 2.
> >
> > 3. Explicit expressions: Finally, our explicit expressions of expected forgetting and generalization error in Theorems 1 and 2 continue to hold. Note that the loss mainly arises from the wrong task routing duing the exploration phase. While a large number of tasks may prolong the exploration phase and slightly increase the loss, both forgetting and generalization error remain bounded after this phase, thanks to the correct routing of tasks.
> >
> > We thank the reviewer again for the constructive feedback. We hope our responses have addressed your concerns satisfactorily. Please do not hesitate to reach out with any further questions or comments, and we will be happy to address them.

---

### Official Review · Reviewer_DWT2 · 2024-11-04

**Soundness:** 3
**Presentation:** 2
**Contribution:** 3
**Rating:** 8
**Confidence:** 3

**Summary:**

This paper provides a theoretical study of Mixture-of-Experts (MoE) models for Continual Learning (CL). Specifically, it examines the CL of linear regression tasks in an overparameterized regime. The benefit of this regime is that each task has multiple possible solutions, increasing the likelihood of expert specialization and transfer.

This paper implements a one-layer MoE architecture with the following design choices:

- **Experts**: Each expert is implemented as a linear model.
- **Router**: A top-1 router is implemented as a linear model.
- **Router Training**: The router is trained using gradient descent.
- **Expert Parameters**: Expert parameters are found through a closed-form solution
- **Training regime**: routing is performed at a per-task level, at each new trainign iteration a new task is sampled from a limited pool of tasks.

Authors propose two key design choices to facilitate learning in their MoE:

- Training loss: in addition to the standard load balancing loss (eq. 7), they propose to use a **locality loss** (is a novel contribution afaik), that facilitates routing of the similar tasks to the same experts
- Early termination: intuitively, once an expert has specialized sufficiently (expected to be the case after T_1 updates), further updates of the routing parameters can result in instabilities. This is achieved by terminating training of the router if the current expert's routing "dominates sufficiently" the other experts' routing.

Authors address the following setting:
- at each update step t, a feature matrix X is randomly sampled following the data generating process described in Definition 1
- among the s_t examples in the feature matrix, there is one example that contains the feature signal
- in the addressed setting, identical tasks can reoccur, as tasks are sampled independently in each update step

Authors derive the following properties of the MoE with previously mentioned design choices:
- the router routes experts primarily based on feature signal, and all experts can be clustered into experts sets, with specialized experts in each expert set
- therefore, under the proposed algorithm (due to the locality loss and termination criteria), experts will likely specialize on certain task clusters after sufficient training round T_1 minimizing the effect of forgetting
- if no termination criteria is applied, proposition 2 states that the specialization will break at round t_2 (the gap between any two experts is predicted to be the same)
- if early termination is applied, then experts within the same expert set are chosen uniformly after T2 updates

The authors derive upper bounds on forgetting and generalization error for the MoE, showcasing that **both are reduced compared to a monolithic single-expert model**. When there are more tasks than experts, forgetting is bounded due to the router’s tendency to route similar tasks to the same experts.

Overall, the intuition that MoE models, with correct routing and module specialization, can address the challenges of CL is natural and compelling. This paper effectively demonstrates how this intuition can be implemented and validated in a controlled, toy setting. The findings can offer a useful starting point for scaling these ideas to settings more relevant for practical AI applications, a very relevant work in this context is [2].

[2] Rypeść, Grzegorz, et al. "Divide and not forget: Ensemble of selectively trained experts in Continual Learning." arXiv preprint arXiv:2401.10191 (2024).

**Strengths:**

**Originality**:
- the idea that forgetting in MoE can be mitigated solely through specialized experts and correct routing is not entirely new, e.g. see [1]
- this paper is original in its theoretical contribution (to the best of my knowledge), providing proofs and bounds on Cl metrics with MoEs
- the proposed locality loss and load balancing loss provide clear mechanisms for task clustering and specialization (even though load balancing loss is not a contribution of this work)

**Quality**:
I think the quality of this work is decent, due to the rigorous theoretical work and relevant experiments. Even though the focus of the work is mainly on the theoretical side, further validation on more complex datasets would be interesting and benefit the credibility of the paper's contributions.

**Clarity**: while the theoretical analysis appears to be rigorous and well presented, I think the presentation would benefit a lot from incorporating more **intuitive explanations**. E.g. authors could explicitly state the idea that forgetting may be prevented solely through correct routing. Also the MNIST experiment's design is somewhat hard to follow.

**Significance**:
Despite the small scale of the experiments, I think this paper opers some interesting directions for future research, mainly among the lines of scaling these ideas to larger systems.

Overall, I appreciate how this paper demonstrates how under proper specialization and routing CL can be addressed with modular solutions like MoEs.

[1] Ostapenko, Oleksiy, et al. "From IID to the Independent Mechanisms assumption in continual learning." AAAI Bridge Program on Continual Causality. PMLR, 2023.

**Weaknesses:**

- the scale of the experiments is small, while one has to uknowledge that the contributions are mainly theoretical
- I would appreciate more intuitive lingo and explanations
- the current implementation is essentially on the one extreme of the parameter sharing trade-off where no transfer happens between tasks?
- it is not exactly clear how these ideas can be extended to large scale MoE with multiple expert layers and per-token routing, where correct routing as well as expert specialization is not guaranteed

**Questions:**

-  is expert specialization necessary for CL with MoEs?
- are modern-day large-scale AI systems operating in an overparameterized regime?
- for load balancing loss, why not use entropy maximization? What are the benefits of the proposed load balancing loss compared to entropy maximization?
- is cross-expert and cross-task transfer possible in the proposed system design?

---

> ### Author Response · Authors · 2024-11-22
> **Response to Reviewer DWT2 (Part 1)**
>
> Thank you for your thorough reviews and constructive comments. We provide our responses to your comments below and have made major revisions in our revised manuscript. To enhance clarity, we have highlighted the revised text in blue for easy identification.
>
> **Q1.** [Experiment scale] the scale of the experiments is small, while one has to acknowledge that the contributions are mainly theoretical
>
> **Response:** Despite the limited time, we made every effort to expand our experiments to address your constructive comments, incorporating more complex datasets such as CIFAR-10, CIFAR-100, and Tiny ImageNet. In Section 6, we replaced the MNIST experiments with CIFAR-10 experiments, and additional experimental details are provided in Appendices A.3-A.6 of the revised manuscript. In these supplementary experiments, we evaluated not only forgetting and generalization errors but also the test accuracy, confirming that the MoE model significantly outperforms the single model in continual learning. Furthermore, the key insights from our theoretical results remain valid, such as early termination ensuring stable convergence with balanced expert loads and the observation that increasing the number of experts does not always improve learning performance.
>
>
> **Q2.** [Intuition explanations] I would appreciate more intuitive lingo and explanations
>
> **Response:** We have incorporated more intuitive explanations to clarify the impact of correct routing on forgetting in our revised manuscript. For instance, following the derivation of the expected forgetting expression in Theorem 1, we provide insights by analyzing the result. In Lines 405-410, we explain: ”However, as stated in Proposition 1, once the expert models converge at $t=T_1$, training on newly arriving tasks with correct routing no longer causes forgetting of previous tasks.  Consequently, for $t\in\{T_1+1,\cdots, T\}$, $\mathbb{E}[F_t]$ decreases with $t$ and converges to zero as $T\rightarrow \infty$. This result highlights that, unlike the oscillatory forgetting observed in Eq. (17) for a single expert, the MoE model effectively minimizes expected forgetting in CL through its correct routing mechanism.” (on page 8)
>
> For the details of the MNIST experiments, we included them in Appendix A.4 of our manuscript due to the page limitation. For example, the task setups are:
>
> “We define the ground truth pool as $\mathcal{W}={(1), (4), (7)}$, representing $N=3$ tasks for recognizing the numbers 1, 4, and 7, respectively. The experiment spans $T=60$ training rounds. Before the experiments in Figure 4, we randomly generate the task arrival sequence $[n_t]_{t\in[T]}$, where each $n_t$ is drawn from ${(1),(4),(7)}$ with equal probability $\frac{1}{3}$. We then conduct two experiments (with and without termination) using the same task arrival order. For each task $t \in [T]$, we randomly select its type (e.g., task $(1)$ for recognizing the number 1) and 100 corresponding samples, ensuring that tasks have distinct distributions and features.”
> We have also included the experiment details of the CIFAR-10, CIFAR-100, and Tiny ImageNet datasets in Appendices A.3, A.5, and A.6, respectively.
>
> **Q3.**[Cross-task transfer] the current implementation is essentially on the one extreme of the parameter sharing trade-off where no transfer happens between tasks?
>
> **Response:** We appreciate the reviewer’s observation. Our approach indeed facilitates knowledge transfer among tasks. Specifically, each expert in the MoE system will be stabilized to handle tasks within the same cluster, ensuring that the knowledge transfer occurs primarily among these similar tasks. Consequently, each task will not only achieve smaller generalization error by leveraging the positive knowledge transfer within the cluster, but also suffer from less forgetting without the interference from dissimilar tasks from other clusters (as characterized in Theorems 1 and 2). Additionally, as we analyzed in Proposition 4, knowledge transfer across clusters introduces severe forgetting for a single-expert case especially when tasks are very dissimilar and interfere with each other, undermining its specialization. The MoE model here can balance knowledge transfer and task specialization while preventing negative interference across task clusters.

---

> ### Author Response · Authors · 2024-11-22
> **Response to Reviewer DWT2 (Part 2)**
>
> **Q4.** [Large scale MoE] it is not exactly clear how these ideas can be extended to large scale MoE with multiple expert layers and per-token routing, where correct routing as well as expert specialization is not guaranteed
>
> **Response:** Thank you for highlighting this important consideration. In large-scale MoE architectures with multiple expert layers, achieving perfect expert specialization can be highly challenging due to the independent gating networks at each layer, which lack cross-layer dependencies. While this may lead to suboptimal routing and degraded learning performance compared to correct routing, we expect that our theoretical insights can still carry over. For instance, as demonstrated in Proposition 1, an exploration phase shall still exist for experts across all MoE layers to explore tasks. Concurrently, the gating networks at each layer will adaptively update their parameters to balance expert workloads, as analyzed in Proposition 2. Additionally, even in the absence of guaranteed correct routing, the gating mechanism will still seek to assign tasks to experts with minimal generalization error, maintaining a degree of specialization. Thank you for this suggestion, we are excited to further explore this more complicated expert setting in our future work.
>
>
> **Q5.** [Expert specialization] Is expert specialization necessary for CL with MoEs?
>
> **Response:** This is a good question. Our study indicates that MoE tends to achieve a certain level of expert specialization for CL. This can be seen by the expressions of the expected forgetting in Theorems 1 and 2, which indicate that a certain level of specialization for each expert can reduce catastrophic forgetting after the system convergence. However, achieving the optimal balance between specialization and knowledge transfer is non-trivial. Excessive specialization may hinder effective transfer across tasks, while insufficient specialization may increase forgetting. This trade-off is fundamental and requires further study to optimize system performance. We appreciate the reviewer for highlighting this point, and we consider it an important direction for future research.
>
>
> **Q6.** [Overparameterized regime] Are modern-day large-scale AI systems operating in an overparameterized regime?
>
> **Response:** Yes, modern-day large-scale AI systems, such as large language models (LLMs) and vision transformers (ViTs), usually operate in an overparameterized regime. This overparameterization enables these models to achieve impressive generalization capabilities while accommodating diverse data distributions and complex tasks.
>
>
> **Q7.** [Load balancing loss] For load balancing loss, why not use entropy maximization? What are the benefits of the proposed load balancing loss compared to entropy maximization?
>
> **Response:** We followed most of the existing MoE studies (e.g., Fedus et al. (2022); Shazeer et al. (2016); Li et al. (2024)) to define this standard load balancing loss in Eq. (7). Compared to entropy maximization, the standard load balancing loss not only controls the average selection probability $P_t^{(m)}$ but also the usage frequency $f_t^{(m)}$ for each expert $m$ since $t=0$. Additionally, it explicitly identifies and penalizes load imbalances, enabling a more predictable and uniform expert load distribution. These benefits facilitate efficient analysis of the load balancing mechanism across different learning phases during MoE training.
>
> Both our adopted load balancing loss and entropy maximization serve as auxiliary losses to promote balanced expert utilization. Hence, our theoretical insights remain applicable to systems employing entropy maximization.
>
>
> **Q8.** [Cross-expert transfer] Is cross-expert and cross-task transfer possible in the proposed system design?
>
> **Response:** Cross-task transfer is an inherent feature of our proposed system, as highlighted in our response to Q3. Cross-expert transfer, however, is indirect in our design. Each expert’s knowledge influences the routing strategy in Eq. (1), which subsequently affects the training dynamics of other experts. To further enhance cross-expert transfer, we consider allowing experts within the same expert set $\mathcal{M}_n$ to share their knowledge after system convergence. Since experts in the same set specialize in the same task cluster, this sharing would not increase catastrophic forgetting. Instead, it could improve overall system performance and robustness. We appreciate the reviewer for raising this valuable point, which inspires further extensions of our work.
>
> We thank the reviewer again for the helpful comments and suggestions for our work.

---

> > ### Comment · Reviewer_DWT2 · 2024-11-24
> > **Thank you for detailed response.**
> >
> > I thank the authors for their detailed response. I will keep my high score.

---

### Comment · Area_Chair_RxFZ · 2024-11-24

Dear Reviewers,

This is a gentle reminder that the authors have submitted their rebuttal, and the discussion period will conclude on November 26th AoE. To ensure a constructive and meaningful discussion, we kindly ask that you review the rebuttal as soon as possible and verify if your questions and comments have been adequately addressed.

We greatly appreciate your time, effort, and thoughtful contributions to this process.

Best regards,
AC

---

### Meta-Review · Area_Chair_RxFZ · 2024-12-16

**Metareview:**

This paper provides a theoretical study of Mixture-of-Experts (MoE) models for Continual Learning (CL). Specifically, it examines the CL of linear regression tasks in an over-parameterized regime. The benefit of this regime is that each task has multiple possible solutions, increasing the likelihood of expert specialization and transfer.

**Additional Comments On Reviewer Discussion:**

All reviewers agree in accepting this work and I will follow their recommendation.

---

### Decision · Program_Chairs · 2025-01-22

Accept (Spotlight)